# Multi-Facet Clustering Variational Autoencoders

**Fabian Falck** [*,1,5]   **Haoting Zhang** [*,2,5]   **Matthew Willetts** [3,6]
**George Nicholson** [1]   **Christopher Yau** [4,5,6]   **Chris Holmes** [1,5,6]
[1]University of Oxford  [2]University of Cambridge  [3]University College London
[4]University of Manchester  [5]Health Data Research UK  [6]The Alan Turing Institute
`fabian.falck@stats.ox.ac.uk, hz381@cl.cam.ac.uk, mwilletts@turing.ac.uk,`
`george.nicholson@stats.ox.ac.uk, cyau@turing.ac.uk, cholmes@stats.ox.ac.uk`

## Abstract

Work in deep clustering focuses on finding a *single* partition of data. However, high-dimensional data, such as images, typically feature *multiple* interesting characteristics one could cluster over. For example, images of objects against a background could be clustered over the shape of the object and separately by the colour of the background. In this paper, we introduce *Multi-Facet Clustering Variational Autoencoders (MFCVAE)*, a novel class of variational autoencoders with a hierarchy of latent variables, each with a Mixture-of-Gaussians prior, that learns multiple clusterings simultaneously, and is trained fully unsupervised and end-to-end. MFC-VAE uses a progressively-trained ladder architecture which leads to highly stable performance. We provide novel theoretical results for optimising the ELBO analytically with respect to the categorical variational posterior distribution, correcting earlier influential theoretical work. On image benchmarks, we demonstrate that our approach separates out and clusters over different aspects of the data in a disentangled manner. We also show other advantages of our model: the compositionality of its latent space and that it provides controlled generation of samples.

## 1 Introduction

*Clustering* is the task of finding structure by partitioning samples in a finite, unlabeled dataset according to statistical or geometric notions of similarity [1, 2, 3].For example, we might group items along axes of empirical variation in the data, or maximise internal homogeneity and external separation of items within and between clusters with respect to a specified distance metric. The choice of similarity measure and how one consequently validates clustering quality is fundamentally a subjective one: it depends on what is useful for a particular task [2, 4]. In this work, we are interested in uncovering abstract, latent characteristics/facets/aspects/levels of the data to understand and characterise the data-generative process. We further assume a fully exploratory, unsupervised setting without prior knowledge on the data, which could be exploited while fitting the clustering algorithm, and in particular without given ground-truth partitions at training time.

When being faced with high-dimensional data such as images, speech or electronic health records, items typically have more than one abstract characteristic. Consider the example of the MNIST dataset [5]: MNIST images possess at least two such characteristics: The digit class, which might impose the largest amount of statistical variation, and the style of the digit (e.g. stroke width). This naturally raises a question: By which characteristic is a clustering algorithm supposed to partition the data? In MNIST, both digit class and (the sub-categories of) style would be perfectly reasonable candidates to answer this question. In our exploratory setting described above, there is not one "correct" partition of the data.

---

[*]Equal contribution.

35th Conference on Neural Information Processing Systems (NeurIPS 2021).

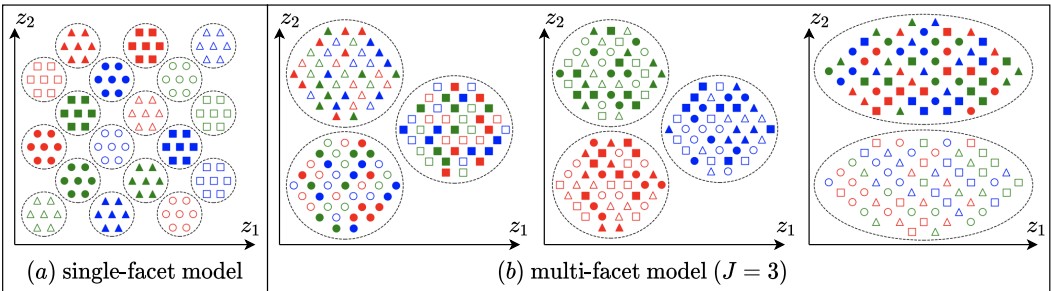

$(a)$ single-facet model $\qquad$ $(b)$ multi-facet model $(J = 3)$

Figure 1: Latent space of a (a) single-facet model and a (b) multi-facet model ($J = 3$) with two dimensions ($z_1$, $z_2$) per facet. Both models perfectly separate the abstract characteristics of the data. However, the multi-facet model disentangles them into three sensible partitions (one per facet) and its required clusters scale linearly as opposed to exponentially w.r.t. the number of aspects in the data.

Deep learning based clustering algorithms, so-called *deep clustering*, were particularly successful in recent years in dealing with high-dimensional data by compressing the inputs into a lower-dimensional latent space in which clustering is computationally tractable [6, 7]. However, almost all of these deep clustering algorithms find only a *single* partition of the data, typically the one corresponding to the given class label in a supervised dataset [8, 9, 10, 11, 12, 13, 14, 15]. When evaluating their model, said approaches validate clustering performance by treating the one supervision label (e.g. digit class in the case of MNIST) as the de-facto "ground truth clustering". We argue that restricting our view to a single facet $C_1$ rather than all or at least multiple facets $(C_1, C_2, \ldots, C_J)$ is an arbitrary, incomplete choice of formulating the problem of clustering a high-dimensional dataset.

To this end, we propose *Multi-Facet Clustering Variational Autoencoders (MFCVAE)*, a principled, probabilistic model which finds multiple characteristics of the data simultaneously through its multiple Mixtures-of-Gaussians (MoG) prior structure. Our contributions are as follows: (a) *Multi-Facet Clustering Variational Autoencoders (MFCVAE)*, a novel class of probabilistic deep learning models for unsupervised, multi-facet clustering in high-dimensional data that can be optimised end-to-end. (b) Novel theoretical results for the optimisation of the corresponding ELBO, correcting and extending an influential, related paper for the single-facet case. (c) Demonstrating MFCVAE's stable empirical performance in terms of multi-facet clustering of various levels of abstraction, compositionality of facets, generative, unsupervised classification, and diversity of generation.

## 2 Multi-facet clustering

High-dimensional data are inherently structured according to a number of abstract characteristics, and in an exploratory setting, it is clear that arbitrarily clustering by one of them is insufficient. However, the question remains whether these multiple facets should also be explicitly *represented* by the model. In particular, one might argue that a single partition could be used to represent all cross-combinations[2] of facets $\mathcal{C} = C_1 \times C_2 \times \cdots \times C_J$ where $C_j = \{1, 2, \ldots, K_j\}$, as in Fig. 1 (a). In this work, we explain that explicitly representing and clustering by multiple facets, as we do in MFCVAE and illustrated in Fig. 1 (b), has the following four properties that are especially desirable in an unsupervised learning setting:

**(a) Discovering a multi-facet structure.** We adopt a probabilistically principled, unsupervised approach, specifying an independent, multiple Mixtures of Gaussians (MoG) prior on the latent space. This induces a disentangled representation across facets, meaning that in addition to examples assigned to certain clusters being homogeneous, the facets (and their corresponding clusters) represent different abstract characteristics of the data (such as digit class or digit style). Because of this multi-facet structure, the total number of clusters required to represent a given multi-partition structure of the data scales *linearly* w.r.t. the number of data characteristics. In comparison, the number of clusters required in a single-facet model scales *exponentially* (see Fig. 1, and Appendix A for details).

---

[2]Note that in practice, not all cross-combinations of facets might be present. For example, in a dataset like MNIST, one might not observe 'right-tilted zeros', even though we observe 'right-tilted' digits and 'zeros'.

**(b) Compositionality of facets.** A multi-facet model has a compositional advantage: different levels of abstraction of the data are represented in separate latent variables. As we will show, this allows qualitatively diverse characteristics to be meaningfully combined.

**(c) Generative, unsupervised classification.** Our method joins a myriad of single-facet clustering models in being able to accurately identify known class structures given by the label in standard supervised image datasets. However, in contrast to previous work, we are also able find interesting characteristics in other facets with homogeneous clusters. We stress that while we compare generative classification performance against other models to demonstrate statistical competitiveness, this task is *not* the main motivation for our fully unsupervised model.

**(d) Diversity of generated samples.** In a generative sense, the structure of the latent space allows us to compose new, synthetic examples by a set of $J$ pairs of (continuous, discrete) latent variables. We can in particular intervene on each facet separately. This yields a rich set of options and fine-grained control for interventions and the diversity of generated examples.

We illustrate these four properties in our experiments in Section 4.

## 3  Multi-Facet Clustering Variational Autoencoders

Our model comprises $J$ latent facets, each learning its own unique clustering of samples via a Mixture-of-Gaussians (MoG) distribution:

$$c_j \sim \text{Cat}(\boldsymbol{\pi}_j), \quad \mathbf{z}_j \mid c_j \sim \mathcal{N}(\boldsymbol{\mu}_{c_j}, \boldsymbol{\Sigma}_{c_j}) \quad (1)$$

where $\boldsymbol{\pi}_j$ is the $j$th facet's $K_j$-dimensional vector of mixing weights, and $(\boldsymbol{\mu}_{c_j}, \boldsymbol{\Sigma}_{c_j})$ are the mean and covariance of the $c_j$th mixture component in facet $j$ ($\boldsymbol{\Sigma}_{c_j}$ can be either diagonal or full).

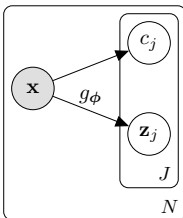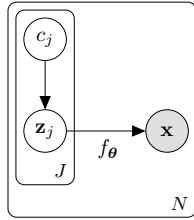

Figure 2: Graphical model of MFCVAE. [Left] Variational posterior, $q_\phi(\vec{\mathbf{z}}, \mathbf{c}|\mathbf{x})$. [Right] Generative model, $p_\theta(\mathbf{x}, \vec{\mathbf{z}}, \mathbf{c})$.

The multi-facet generative model (Fig. 2 [Right]) is thus structured as

$$p_\theta(\mathbf{x}, \vec{\mathbf{z}}, \mathbf{c}) = p_\theta(\mathbf{x}|\vec{\mathbf{z}})p_\theta(\vec{\mathbf{z}}|\mathbf{c})p_\theta(\mathbf{c}) = p_\theta(\mathbf{x}|\vec{\mathbf{z}})\prod_{j=1}^{J} p_\theta(\mathbf{z}_j|c_j)p_\theta(c_j), \quad (2)$$

where $\mathbf{c} = \{c_1, c_2, ..., c_J\}$, $\vec{\mathbf{z}} = \{\mathbf{z}_1, \mathbf{z}_2, ..., \mathbf{z}_J\}$, and $p_\theta(\mathbf{x}|\vec{\mathbf{z}})$ is a user-defined likelihood model, for example a product of Bernoulli or Gaussian distributions, which is parameterised with a deep neural network $f(\vec{\mathbf{z}}; \theta)$. Importantly, this structure in Eq. (2) encodes prior independence across facets, i.e. $p_\theta(\vec{\mathbf{z}}, \mathbf{c}) = \prod_j p_\theta(\mathbf{z}_j, c_j)$, thereby encouraging facets to learn clusterings that span distinct subspaces of $\vec{\mathbf{z}}$. The overall marginal prior $p_\theta(\vec{\mathbf{z}})$ can be interpreted as a product of independent MoGs.

### 3.1  VaDE tricks

To train this model, we wish to optimise the evidence lower bound (ELBO) of the data marginal likelihood using an amortised variational posterior $q_\phi(\vec{\mathbf{z}}, \mathbf{c}|\mathbf{x})$ (Fig. 2 [Left]), parameterised by a neural network $g(\mathbf{x}; \phi)$, within which we will perform Monte Carlo (MC) estimation where necessary to approximate expectations

$$\log p(\mathcal{D}) \geq \mathcal{L}(\mathcal{D}; \theta, \phi) = \mathbb{E}_{\mathbf{x} \sim \mathcal{D}}\left[\mathbb{E}_{q_\phi(\vec{\mathbf{z}}, \mathbf{c}|\mathbf{x})}[\log \frac{p_\theta(\mathbf{x}, \vec{\mathbf{z}}, \mathbf{c})}{q_\phi(\vec{\mathbf{z}}, \mathbf{c}|\mathbf{x})}]\right]. \quad (3)$$

What should we choose for $q_\phi(\vec{\mathbf{z}}, \mathbf{c}|\mathbf{x})$? Training deep generative models with discrete latent variables can be challenging, as reparameterisation tricks so far developed, such as the Gumbel-Softmax trick [16, 17], necessarily introduce bias into the optimisation, and become unstable when a discrete latent variable has a high cardinality. Our setting where we have multiple discrete latent variables is even more challenging. First, the bias from using the Gumbel-Softmax trick compounds when there is a hierarchy of dependent latent variables, leading to poor optimisation [18]. Second, we cannot necessarily avail ourselves of advances in obtaining good estimators for discrete latent variables as either they do not carry over to the hierarchical case [19], or are restricted to binary latent variables

[20]. Third, we wish for light-weight optimisation, avoiding the introduction of additional neural networks whenever possible as this simplifies both training and neural specification.

Thus, we sidestep these problems, bias from relaxations of discrete variables *and* the downsides of additional amortised-posterior neural networks for the discrete latent variables, by developing the hierarchical version of the *VaDE trick*. This trick was first developed for clustering VAEs with a *single* Gaussian mixture in the generative model [10]. Informally, the idea (for a single-facet model) is to define a Bayes-optimal posterior for the discrete latent variable using the responsibilities of the constituent components of the mixture model; these responsibilities are calculated using samples taken from the amortised posterior for the continuous latent variable.

Estimating the ELBO for models of this form does not require us to take MC samples from discrete distributions—the data likelihood is conditioned only on the continuous latent variable $\vec{z}$, which we sample using the reparameterization trick [21], and the posterior for $\vec{z}$ is conditioned only on $\mathbf{x}$. Thus, when calculating the ELBO, we can cheaply marginalise out discrete latent variables where needed. In other words, we do not have to perform multiple forward passes through the decoder as neither it nor the $\vec{z}$ samples we feed it depend on $c$.

As it is fundamental to our method, we now briefly recapitulate the original VaDE trick for VAEs with a single latent mixture (correcting a misapprehension in the original form of this idea) and will then cover our hierarchical extension[3].

**Single-Facet VaDE Trick:** Consider a single facet model, so the generative model is $p_\theta(\mathbf{x}, \mathbf{z}, c) = p_\theta(\mathbf{x}|\mathbf{z})p_\theta(\mathbf{z}|c)p_\theta(c)$. Introduce a posterior $q_\phi(\mathbf{z}, c|\mathbf{x}) = q_\phi(\mathbf{z}|\mathbf{x})q_\phi(c|\mathbf{x})$ where $q_\phi(\mathbf{z}|\mathbf{x})$ is a multivariate Gaussian with diagonal covariance. The ELBO for this model for one datapoint is

$$\mathcal{L}(\mathbf{x}; \theta, \phi) = \mathbb{E}_{q_\phi(\mathbf{z}, c|\mathbf{x})}[\log \frac{p_\theta(\mathbf{x}|\mathbf{z})p_\theta(\mathbf{z}|c)p_\theta(c)}{q_\phi(\mathbf{z}|\mathbf{x})q_\phi(c|\mathbf{x})}] = \mathbb{E}_{q_\phi(\mathbf{z}, c|\mathbf{x})}[\log \frac{p_\theta(\mathbf{x}|\mathbf{z})p_\theta(\mathbf{z})p_\theta(c|\mathbf{z})}{q_\phi(\mathbf{z}|\mathbf{x})q_\phi(c|\mathbf{x})}], \quad (4)$$

where we have chosen to rewrite the generative model factorisation, $p_\theta(\mathbf{z}) = \sum_c p_\theta(\mathbf{z}|c)p_\theta(c)$ is the marginal mixture of Gaussians, and $p_\theta(c|\mathbf{z}) = p_\theta(\mathbf{z}|c)p_\theta(c)/p_\theta(\mathbf{z})$ is the Bayesian posterior for $c$.

Expanding out the ELBO, we get

$$\mathcal{L}(\mathbf{x}; \theta, \phi) = \mathbb{E}_{q_\phi(\mathbf{z}|\mathbf{x})} \log p_\theta(\mathbf{x}|\mathbf{z}) - \text{KL}\left[q_\phi(\mathbf{z}|\mathbf{x})||p_\theta(\mathbf{z})\right] - \mathbb{E}_{q_\phi(\mathbf{z}|\mathbf{x})} \text{KL}\left[q_\phi(c|\mathbf{x})||p_\theta(c|\mathbf{z})\right]. \quad (5)$$

We can *define* $q_\phi(c|\mathbf{x})$ such that $\mathbb{E}_{q_\phi(\mathbf{z}|\mathbf{x})} \text{KL}\left[q_\phi(c|\mathbf{x})||p_\theta(c|\mathbf{z})\right]$ is *minimal*, by construction, which is the case if we choose $q_\phi(c|\mathbf{x}) \propto \exp\left(\mathbb{E}_{q_\phi(\mathbf{z}|\mathbf{x})} \log p_\theta(c|\mathbf{z})\right)$ as we will show in Theorem 1. This means that we can simply use samples from the posterior for $\mathbf{z}$ to define the posterior for $c$, using Bayes' rule within the latent mixture model.

**Remark:** We note, however, that in the original description of this idea in [10], it was claimed that $\mathbb{E}_{q_\phi(\mathbf{z}|\mathbf{x})} \text{KL}\left[q_\phi(c|\mathbf{x})||p_\theta(c|\mathbf{z})\right]$ could, in general, be set to *zero*, which is not the case. Rather, this KL can be minimised, in general, to a *non-zero* value. We discuss this misapprehension in more detail and why the empirical results in [10] are still valid in Appendix B.1.1.

**Theorem 1.** *(Single-Facet VaDE Trick) For any probability distribution $q_\phi(\mathbf{z}|\mathbf{x})$, the distribution $q_\phi(c|\mathbf{x})$ that minimises $\mathbb{E}_{q_\phi(\mathbf{z}|\mathbf{x})} \text{KL}\left[q_\phi(c|\mathbf{x})||p_\theta(c|\mathbf{z})\right]$ in (5) is*

$$\underset{q_\phi(c|\mathbf{x})}{\operatorname{argmin}} \mathbb{E}_{q_\phi(\mathbf{z}|\mathbf{x})} \text{KL}\left[q_\phi(c|\mathbf{x})||p_\theta(c|\mathbf{z})\right] = \boldsymbol{\pi}(c|q_\phi(\mathbf{z}|\mathbf{x})) \quad (6)$$

*with the minimum value attained being*

$$\underset{q_\phi(c|\mathbf{x})}{\min} \mathbb{E}_{q_\phi(\mathbf{z}|\mathbf{x})} \text{KL}\left[q_\phi(c|\mathbf{x})||p_\theta(c|\mathbf{z})\right] = -\log Z(q_\phi(\mathbf{z}|\mathbf{x})) \quad (7)$$

$$\textit{where} \qquad \boldsymbol{\pi}(c|q_\phi(\mathbf{z}|\mathbf{x})) := \frac{\exp\left(\mathbb{E}_{q_\phi(\mathbf{z}|\mathbf{x})} \log p(c|\mathbf{z})\right)}{Z(q_\phi(\mathbf{z}|\mathbf{x}))} \textit{ for } c = 1, \dots, K \quad (8)$$

$$Z(q_\phi(\mathbf{z}|\mathbf{x})) := \sum_{c=1}^{K} \exp\left(\mathbb{E}_{q_\phi(\mathbf{z}|\mathbf{x})} \log p(c|\mathbf{z})\right) . \quad (9)$$

*Proof: See Appendix B.1.* □

---

[3]We note that the original VaDE paper, besides the misapprehension discussed in Section 3.1 and Appendix B.1.1, proposed a highly complex training algorithm with various pre-training heuristics which we significantly simplified while maintaining or increasing performance (details in Appendix D.5).

**Multi-facet VaDE Trick:** In this work, we consider the case of having $J$ facets, each with its own pair of variables $(\mathbf{z}_j, c_j)$. Perhaps surprisingly, we do *not* have to make a mean-field assumption *between* the $J$ facets for $\mathbf{c}$ once we have made one for $\vec{\mathbf{z}}$. In other words, once we have chosen that $q_\phi(\vec{\mathbf{z}}, \mathbf{c}|\mathbf{x}) = q_\phi(\mathbf{c}|\mathbf{x}) \prod_{j=1}^{J} q_\phi(\mathbf{z}_j|\mathbf{x})$, where $q_\phi(\mathbf{z}_j|\mathbf{x})$ is defined to be a multivariate Gaussian with diagonal covariance for each $j$, the optimal $q_\phi(\mathbf{c}|\mathbf{x})$ similarly factorises[4]. We formalise this:

**Theorem 2.** *(Multi-Facet VaDE Trick for factorized $q_\phi(\vec{\mathbf{z}}|\mathbf{x})$, $p(\vec{\mathbf{z}}, \mathbf{c})$) For any factorized probability distribution $q_\phi(\vec{\mathbf{z}}|\mathbf{x}) = \prod_j q_\phi(\mathbf{z}_j|\mathbf{x})$, the distribution $q_\phi(\mathbf{c}|\mathbf{x})$ that minimises $\mathbb{E}_{q_\phi(\vec{\mathbf{z}}|\mathbf{x})} \mathrm{KL}\left[q_\phi(\mathbf{c}|\mathbf{x})||p_\theta(\mathbf{c}|\vec{\mathbf{z}})\right]$ under factorized prior $p(\vec{\mathbf{z}}, \mathbf{c}) = \prod_j p(\mathbf{z}_j, c_j)$ of (2) is*

$$\underset{q_\phi(\mathbf{c}|\mathbf{x})}{\mathrm{argmin}}\, \mathbb{E}_{q_\phi(\vec{\mathbf{z}}|\mathbf{x})} \mathrm{KL}\left[q_\phi(\mathbf{c}|\mathbf{x})||p_\theta(\mathbf{c}|\vec{\mathbf{z}})\right] = \prod_j \boldsymbol{\pi}_j(c_j|q_\phi(\mathbf{z}_j|\mathbf{x})) \tag{10}$$

*where the minimum value is attained at*

$$\min_{q_\phi(\mathbf{c}|\mathbf{x})} \mathbb{E}_{q_\phi(\vec{\mathbf{z}}|\mathbf{x})} \mathrm{KL}\left[q_\phi(\mathbf{c}|\mathbf{x})||p_\theta(\mathbf{c}|\vec{\mathbf{z}})\right] = -\sum_j \log Z_j(q_\phi(\mathbf{z}_j|\mathbf{x})) \tag{11}$$

$$\text{where} \qquad \boldsymbol{\pi}_j(c_j|q_\phi(\mathbf{z}_j|\mathbf{x})) := \frac{\exp(\mathbb{E}_{q_\phi(\mathbf{z}_j|\mathbf{x})} \log p_\theta(c_j|\mathbf{z}_j))}{Z_j(q_\phi(\mathbf{z}_j|\mathbf{x}))}, \text{for } c_j = 1, \ldots, K_j \tag{12}$$

$$Z_j(q_\phi(\mathbf{z}_j|\mathbf{x})) := \sum_{c_j=1}^{K_j} \exp(\mathbb{E}_{q_\phi(\mathbf{z}_j|\mathbf{x})} \log p_\theta(c_j|\mathbf{z}_j)). \tag{13}$$

*Proof:  See Appendix B.2.* □

Note that we use Eq. (12) as the probability distribution of assigning input $\mathbf{x}$ to clusters of facet $j$.

Armed with these theoretical results, we can now write the ELBO for our model, with the optimal posterior for $\mathbf{c}$, in a form that trivially admits stochastic estimation and does not necessitate extra recognition networks for $\mathbf{c}$,

$$\mathcal{L}^{\mathrm{MFCVAE}}(\mathcal{D}; \theta, \phi) = \mathbb{E}_{\mathbf{x} \sim \mathcal{D}} \Bigg[ \mathbb{E}_{q_\phi(\vec{\mathbf{z}}|\mathbf{x})} \log p_\theta(\mathbf{x}|\vec{\mathbf{z}})$$
$$- \sum_{j=1}^{J} \left[ \mathbb{E}_{q_\phi(c_j|\mathbf{x})} \mathrm{KL}(q_\phi(\mathbf{z}_j|\mathbf{x})||p_\theta(\mathbf{z}_j|c_j)) + \mathrm{KL}(q_\phi(c_j|\mathbf{x})||p(c_j)) \right] \Bigg] \tag{14}$$

where the optimal $q_\phi(c_j|\mathbf{x})$ is given by Eq. (12) for each $j$.

To obtain the posterior distributions for $\mathbf{c}$, we take MC samples from $q_\phi(\vec{\mathbf{z}}|\mathbf{x})$ and use these to construct the posterior as in Eq. (12). We found one MC sample ($L = 1$; for each facet and for each $\mathbf{x}$) to be sufficient. We derive the complete MC estimator which we use as the loss function of our model and ablations on two alternative forms in Appendix C.

### 3.2  Neural implementation and training algorithm

It is worth pausing here to consider what neural architecture best suits our desire for learning multiple disentangled facets, and then further how we can best train our model to robustly elicit from it well-separated facets. In the introduction, we discussed the different plausible ways to cluster high-dimensional data, such as in MNIST digits by stroke thickness and class identity. These different aspects intuitively correspond to different levels of abstraction about the image. It is thus natural that these levels would be best captured by different depths of the neural networks in each amortised posterior. These ideas have motivated the use of *ladder networks* in deep generative models that aim to learn different facets of the input data into different layers of latent variables. Here, we take inspiration from *Variational Ladder Autoencoders (VLAEs)* [22]: A VLAE architecture has a deterministic "backbone" in both the recognition and generative model. The different layers of latent variables branch out from these at different depths along. This inductive bias naturally leads to stratification and does so without having to bear the computational cost of training a completely separate encoder (say) for each layer. Here, we use this ladder architecture for MFCVAE, as illustrated in Fig. 3, and refer to Appendix D.2 for further implementation details.

---

[4]We also provide the VaDE trick for the general form of the posterior for $\vec{\mathbf{z}}$, i.e. without assuming the factorisation $q_\phi(\vec{\mathbf{z}}|\mathbf{x}) = \prod_{j=1}^{J} q_\phi(\mathbf{z}_j|\mathbf{x})$, in Appendix B.3.

Further, we found *progressive training* [23], previously shown to help VLAEs learn layer-by-layer disentangled representations, to be of great use in making each facet consistently represent the same aspects of data. The general idea of progressive training is to start with training a single facet (typically the one corresponding to the deepest recognition and generative neural networks) for a certain number of epochs, and progressively and smoothly loop in the other facets one after the other. We discuss the details of our progressive training schedule in Appendix D.3. We find that both the VLAE architecture and pro-

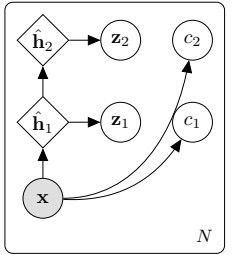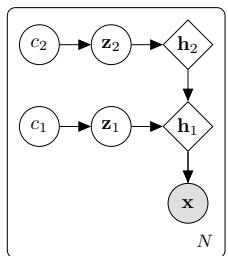

Figure 3: Ladder-MFCVAE architecture. [Left] Variational posterior. [Right] Generative model.

gressive training are jointly important to stabilise training and get robust qualitative and quantitative results as we show in Appendix E.1.

# 4 Experiments

In the following, we demonstrate the usefulness of our model and its prior structure in four experimental analyses: (a) discovering a multi-facet structure (b) compositionality of latent facets (c) generative, unsupervised classification, and (d) diversity of generated samples from our model. We train our model on three image datasets: MNIST [5], 3DShapes (two configurations) [24] and SVHN [25]. We refer to Appendices D and E for experimental details and further results. We also provide our code implementing MFCVAE, using *PyTorch Distributions* [26], and reproducing our results at https://github.com/FabianFalck/mfcvae.

## 4.1 Discovering a multi-facet structure

We start by demonstrating that our model can discover a multi-facet structure in data. Fig. 4 visualises input examples representative of clusters in a two-facet ($J = 2$) model. For each facet $j$, input examples $x$ with latent variable $z_j$ are assigned to latent cluster $c_j = \text{argmax}_{c_j} \pi_j(c_j|q_\phi(\mathbf{z}_j|\mathbf{x}))$ according to Eq. (12). Surprisingly, we find that we can represent the two most striking data characteristics—digit class and style (mostly in the form of stroke width, e.g. 'bold', 'thin') in MNIST, object shape and floor colour in 3DShapes (configuration 1), and digit class and background colour in SVHN—in two separate facets of the data. In each facet, clusters are homogeneous w.r.t. a value from the represented characteristic. When comparing our results on MNIST with LTVAE [27], the model closest to ours in its attempt to learn a clustered latent space of multiple facets, LTVAE struggles to separate data characteristics into separate facets (c.f. [27] Fig. 5; in particular, both facets learn digit class, i.e. this characteristic is not properly disentangled between facets), whereas MFCVAE better isolates the two.

To quantitatively assess the degree of disentanglement in the learned multi-facet structure of our model, we perform a set of supervised experiments. For each dataset, we formulate three classification tasks, for which we use latent embeddings $\mathbf{z}_1$, $\mathbf{z}_2$ and $\vec{\mathbf{z}}$, respectively, sampled from their corresponding amortised posterior, as inputs, and the label present in the dataset (e.g. digit class in MNIST) as the target. For each task and dataset, we train (on the training inputs) a multi-layer perceptron of one hidden layer with 100 hidden units and a ReLU activation, and an output layer followed by a softmax activation, which are the default hyperparameters in the Python package sklearn. Table 1 shows test accuracy of these experiments. We find that the supervised classifiers predict the supervised label with high accuracy when presented with latent embeddings which we found to cluster the abstract characteristic corresponding to this label, or with the concatenation of both latent embeddings. However, when presented with latent embeddings corresponding to the "non-label" facet, the classifier should—if facets are strongly disentangled—not be presented with useful information to learn the supervised mapping, and this is indeed what we find, observing significantly worse performance. This demonstrates the multi-facet structure of the latent space, which learns separate abstract characteristics of the data.

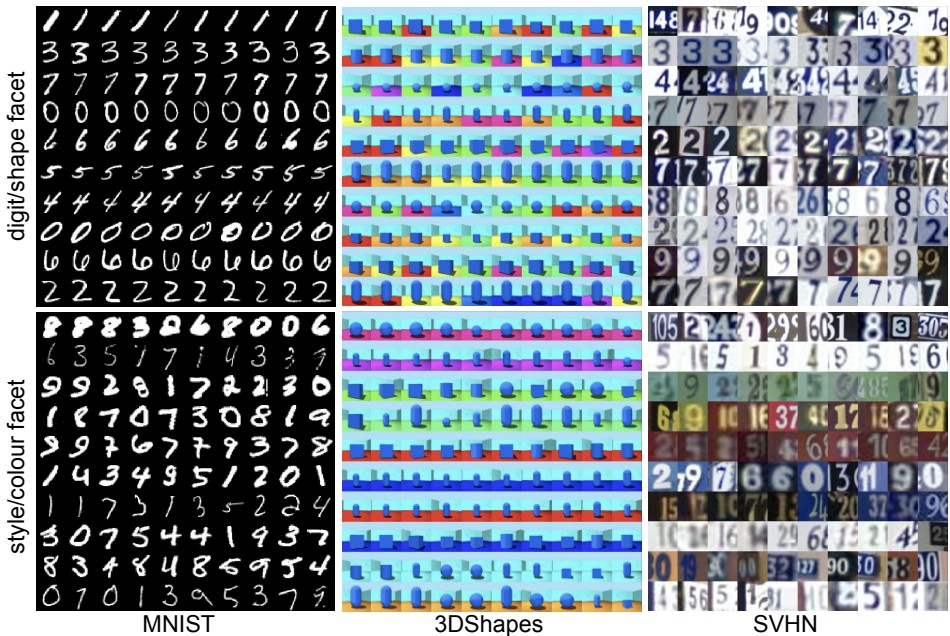

Figure 4: Input examples for clusters of MFCVAE with two-facets ($J = 2$) trained on MNIST, 3DShapes and SVHN. Clusters (rows) in each facet $j$ are sorted in decreasing order by the average assignment probability of test inputs over each cluster. Inputs (columns) are sorted in decreasing order by their assignment probability $\max_{c_j} \boldsymbol{\pi}_j(c_j|q_\phi(\mathbf{z}_j|\mathbf{x}))$. We visualise the first 10 clusters and inputs from the test set (see Appendix E.3 for all clusters).

Table 1: Supervised classification experiment to assess the disentanglement of MFCVAE's multi-facet structure on all three datasets. Values report test accuracy in %. Error bars are the sample standard deviation across 3 runs.

|  | MNIST | 3DShapes config. 1 | | 3DShapes config. 2 | | SVHN |
|---|---|---|---|---|---|---|
|  | digit class | object shape | floor colour | object shape | wall colour | digit class |
| $\mathbf{z}_1$ | 17.34 (0.24) | 95.00 (0.45) | 20.00 (0.68) | 98.26 (0.16) | 73.40 (1.48) | 69.46 (0.36) |
| $\mathbf{z}_2$ | 94.95 (0.04) | 32.43 (1.38) | 100.00 (0.00) | 24.41 (1.34) | 100.00 (0.00) | 22.30 (0.16) |
| $\vec{\mathbf{z}}$ | 95.27 (0.07) | 95.18 (0.42) | 100.00 (0.00) | 98.19 (0.30) | 99.97 (0.06) | 70.39 (0.29) |

## 4.2 Compositionality of latent facets

A unique advantage of the prior structure of MFCVAE compared to other unsupervised generative models, say a VAE with an isotropic Gaussian prior, is that it allows different abstract characteristics to be composed in the separated latent space. Here, we show how this enables interventions on a per-facet basis, illustrated with a two-facet model where style/colour is learned in one facet and digit/shape is learned in the other facet. Let us have two inputs $\mathbf{x}^{(1)}$ and $\mathbf{x}^{(2)}$ assigned to two different style clusters according to Eq. (12) (and two different digit clusters). For both inputs, we obtain their latent representation $\tilde{\mathbf{z}}_j$ as the modes of $q_\phi(\mathbf{z}_j|\mathbf{x})$, respectively. Now, we swap the style/colour facet's representation, i.e. $\tilde{\mathbf{z}}_1$ of both inputs for MNIST, and $\tilde{\mathbf{z}}_2$ of both inputs for 3DShapes and SVHN, and pass these together with their unchanged digit/shape representation ($\tilde{\mathbf{z}}_2$ for MNIST and $\tilde{\mathbf{z}}_1$ for 3DShapes and SVHN) through the decoder $f(\vec{\mathbf{z}}; \theta)$ to get reconstructions $\hat{\mathbf{x}}^{(1)} = f(\{\tilde{\mathbf{z}}_1^{(1)}, \tilde{\mathbf{z}}_2^{(2)}\}; \theta)$ and $\hat{\mathbf{x}}^{(2)} = f(\{\tilde{\mathbf{z}}_1^{(2)}, \tilde{\mathbf{z}}_2^{(1)}\}; \theta)$ which we visualise in Fig. 5 (see Appendix E.4 for a more rigorous explanation of this swapping procedure).

Surprisingly, by construction of this intervention in our multi-facet model, we observe reconstructions that "swap" their style/background colour, yet in most cases preserve their digit/shape. This intervention is successful across a wide set of clusters on MNIST and 3DShapes. It works less so on

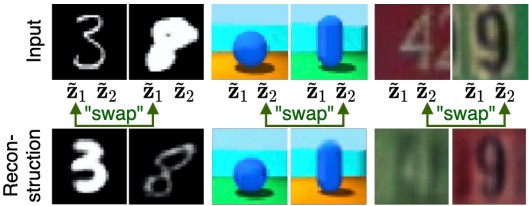

Figure 5: Reconstructions of two input examples when swapping their latent style/colour.

Table 2: Unsupervised clustering accuracy (%) of single-facet (SF) and multi-facet (MF), generative (G) and non-generative (NG) models on the test set. Error bars (if available) are the sample standard deviation across multiple runs. Results marked with $^{\eta}$ do not provide error bars.

| Method | MNIST | SVHN |
|---|---|---|
| DEC ([8]; SF; NG) | 84.3 $^{\eta}$ | 11.9 (0.4) |
| VaDE ([10]; MLP; SF; G) | 94.46 $^{\eta}$; 89.09 (3.32) | 27.03 (1.53) |
| VaDE ([10]; conv.; SF; G) | 92.65 (1.14) | 30.80 (1.99) |
| IMSAT ([11]; SF; NG) | 98.4 (0.4) | 57.3 (3.9) |
| ACOL-GAR ([15]; SF; NG) | 98.32 (0.08) | 76.80 (1.30) |
| VLAC ([28]; MF; G) | - | 37.8 (2.2) |
| LTVAE ([27]; MF; G) | 86.3 | - |
| MFCVAE (ours; MF; G) | 92.02 (3.18) | 56.25 (0.93) |

SVHN where we hypothesise that this is due to the much more diverse dataset and (consequently) the model reaching a lower fit (see Section 4.3). We show further examples including failure cases in Appendix E.4 which show that our model learns a multi-facet structure allowing complex inventions.

### 4.3 Generative, unsupervised classification

Recall our fully unsupervised, exploratory setting of clustering where the goal is to identify and characterise multiple meaningful latent structures *de novo*. In practice, we have no ground-truth data partition—if labels were available, the task would be better formulated as a supervised classification in the first place. That said, it is often reasonable to assume that the class label in a supervised dataset represents a semantically meaningful latent structure that contributes to observed variation in the data. Indeed, this assumption underlies the common approach for benchmarking clustering models on labelled data: the class label is hidden during training; afterwards it is revealed as a pseudo ground-truth partitioning of the data for assessing clustering "accuracy". MFCVAE aims to capture multiple latent structures and can be deployed as a multi-facet generative classifier, as distinct from standard single-facet discriminative classifiers [1, p.30]. But we emphasise that high classification accuracy is attained as a by-product, and is *not* our core goal—we do not explicitly target label accuracy, nor does high label accuracy necessarily correspond to the "best" multi-facet clustering.

Following earlier work, in Table 2, we report classification performance on MNIST and SVHN in terms of *unsupervised clustering accuracy* on the test set, which intuitively measures homogeneity w.r.t. a set of ground-truth clusters in each facet (see Appendix E.5 for a formal definition). We compare our method against commonly used single-facet (SF) and multi-facet (MF), generative (G) and non-generative (NG) deep clustering approaches (we use results as reported) of both deterministic and probabilistic nature. We report the mean and standard deviation (if available) of accuracy over $T$ runs with different random seeds, where $T = 10$ for MFCVAE. For VaDE [10], we report results from the original paper, and our two implementations, one with a multi-layer perceptron encoder and decoder architecture, one using convolutional layers. Models marked with $^{\eta}$ explicitly state that they instead report the best result obtained from $R$ restarts with different random seeds (DEC: $R = 20$, VaDE: $R = 10$). Both of these types of reporting in previous work—not providing error bars over several runs and picking the best run (while not providing error bars)—ignore stability of the model w.r.t. initialisation. We further discuss this issue and the importance of stability in deep clustering approaches in Appendix E.1.

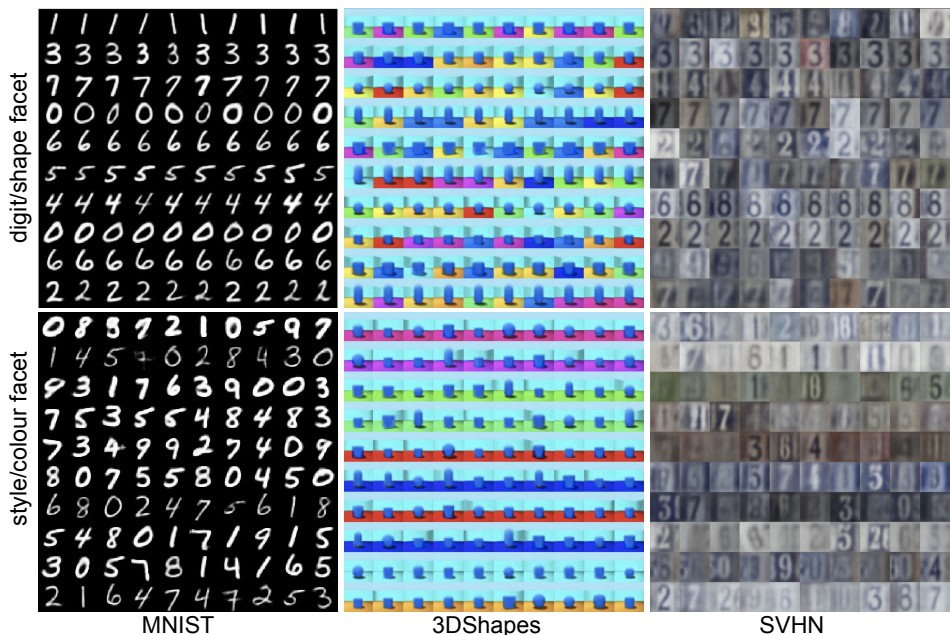

Figure 6: Synthetic samples generated from MFCVAE with two facets ($J = 2$) trained on MNIST, 3DShapes, and SVHN. For each cluster $c_j$ in facet $j$, $\mathbf{z}_j$ is sampled from $p(\mathbf{z}_j|c_j)$ and $\mathbf{z}_{j'}$ is sampled from $p(\mathbf{z}_{j'})$ for the other facet $j' \neq j$. Each row corresponds to 10 random samples from a cluster. Clusters (rows) are sorted and selected (and are from the same trained model) as in Fig. 4 (see Appendix E.6 for visualisation of all clusters and comparison with LTVAE).

MFCVAE is able to recover the assumed ground-truth clustering stably. It achieves competitive performance compared to other probabilistic deep clustering models, but is clearly outperformed by ACOL-GAR on SVHN, a single-facet, non-generative and deterministic model which does not possess three of the four properties demonstrated in Sections 4.1, 4.4) and 4.2). Besides the results presented in the table, we also note that MFCVAE performs strongly on 3DShapes, obtaining $99.46\% \pm 1.10\%$ for floor colour and $88.47\% \pm 1.82\%$ for object shape on configuration 1, and $100.00\% \pm 0.00\%$ for wall colour and $90.05\% \pm 2.65\%$ for object shape on configuration 2. Lastly, it is worth noting that we report classification performance for the same hyperparameter configurations and training runs of our model that are used in all experimental sections and in particular for Fig. 4, 5 and 6, i.e. our trained model has a pronounced multi-facet characteristic. In contrast, while it is somewhat unclear, LTVAE seems to report its clustering performance when trained with only a single facet, not when performing multi-facet clustering [27].

## 4.4  Diversity of generated samples

We lastly show that MFCVAE enables diverse generation of synthetic examples for each given cluster in the different facets, as a downstream task in addition to clustering. To obtain synthetic examples for a cluster $c_j$ in facet $j$, we sample $\mathbf{z}_j$ from $p(\mathbf{z}_j|c_j)$, and sample $\mathbf{z}_{j'}$ from $p(\mathbf{z}_{j'})$ for all other facets $j' \neq j$. We then take the modes of $p_\theta(\mathbf{x}|\vec{\mathbf{z}})$ where $\vec{\mathbf{z}} = (\mathbf{z}_1, \ldots, \mathbf{z}_j, \ldots, \mathbf{z}_J)$ as the generated images. Fig. 6 shows synthetic examples generated from the models ($J = 2$) trained on MNIST, 3DShapes and SVHN.

For all three datasets, we observe synthetic samples that are homogeneous w.r.t. the characteristic value (e.g. 'red background') of a cluster in the chosen facet (as we sample this continuous latent variable from the conditional distribution), but heterogeneous and diverse w.r.t. all other facets (as we sample all other continuous latent variables from their marginal distribution). For example, on MNIST, when fixing a cluster in the digit facet, we observe generated samples that have the same digit class (e.g. all '1'), but are diverse in style (e.g. different 'stroke width'). Conversely, when fixing a cluster in the style facet, we get samples homogeneous in style, but heterogeneous in digit class. Likewise, on 3DShapes, fixing a cluster in the wall colour facet produces generations diverse in shape,

but having the same wall color, and conversely when fixing the shape facet. Besides, in all clusters, generated samples are diverse w.r.t. other factors of variation on 3DShapes, such as orientation and scale. On SVHN, while less strong than in Fig. 4, these patterns extend here to the two facets style (background colour is particularly distinct) and digit class. These results are consistent with and underline the observed disentanglement of facets that we found in our previous experimental analyses. We also compare sample generation performance between MFCVAE and LTVAE and assess the diversity of generations quantitatively in Appendix E.6.

## 5 Related work

Within the deep generative framework, various deep clustering models have been proposed. VaDE [10] is the most important prior work, a probabilistic model which has been highly influential in deep clustering. Related approaches, GM-VAEs [29] and GM-DGMs [30, 31, 32], have similar overall performance and explicitly represent the discrete clustering latent variable during training. Non-parametric approaches include DLDPMMs [30], and HDP-VAEs [33]. Further, many non-generative methods for clustering have been proposed that use neural components [8, 9, 14, 11, 15, 29, 12]. All these approaches, however, propose single-facet models.

Hierarchical VAEs can both be a way to learn more powerful models [34, 35, 36, 37, 38], but can also enable to separate out representations where each layer of latent variables represents a different aspect of the data. Variational Ladder Autoencoders (VLAEs) [22] aim to do the latter: to learn independent sets of latent variables, each representing some part of the data; but each group of latent variables within this set has a $\mathcal{N}(\mathbf{0}, \mathbf{1})$ prior, so it does not perform clustering. Recently, progressive training for VLAEs has been proposed [23] which sharpens the separation between layers. Here, we also mention disentanglement methods [39, 40, 41, 42] which likewise attempt to find separated latent variables. However, rather than discovering facets through the prior and a hierarchical structure, these techniques attempt to find statistically-independent representations via regularisation, leading the loss to deviate from the ELBO. Unfortunately, these methods require lucky selection of hyperparameters to work [43, 44], and do not provide a clustered latent space.

Learning multiple clusterings simultaneously has been studied in the case of very low dimensional datasets [45, 46, 47, 48] under the names *alternative clusterings* and *non-redundant clustering*. However, when it comes to clustering high-dimensional data like images, approaches are rare. The recently proposed LTVAE [27] aims to perform this task, proposing a variational autoencoder with a latent tree model prior for a set of continuous latent variables $\vec{z}$, of which each $\mathbf{z}_j$ has a GMM prior. The neural components are trained via stochastic gradient ascent under the ELBO; this is interleaved with a heuristic (hill-climbing) search algorithm to grow or prune the tree structure and message-passing to learn its nodes' GMM parameters of the current structure of the tree prior in a manner reminiscent of SVAEs [49], rendering the entire training algorithm *not* end-to-end differentiable (in contrast to MFCVAE). LTVAE learns multiple clusterings over the data, however, lacks a proper disentanglement of facets, as discussed in Section 4.1.

## 6 Conclusion

We introduced Multi-Facet Clustering Variational Autoencoders (MFCVAE), a novel class of probabilistic deep learning models for unsupervised, multi-partition clustering in high-dimensional data which is end-to-end differentiable. We provided novel theoretical results for optimising its ELBO, correcting and extending an influential related paper for the single-facet case. We demonstrated MFCVAE's empirical performance in terms of multi-facet clustering of various levels of abstraction, and the usefulness of its prior structure for composing, classifying and generating samples, achieving state-of-the-art performance among deep probabilistic multi-facet models.

An important limitation of our work shared with many other deep clustering algorithms is the lack of a procedure to find good hyperparameters through a metric known at training time. Future work should explore: MFCVAE with $J > 2$; automatic tuning of hyperparameters $J$ and $K_j$; application to large-scale datasets of other modalities; and regularising the model facet-wise to further enforce disentangled representations in the latent space [50]. While we successfully stabilised model training, further work will be key to harness the full potential of deep clustering models.

## Acknowledgments and Disclosure of Funding

FF and HZ acknowledge the receipt of studentship awards from the Health Data Research UK-The Alan Turing Institute Wellcome PhD Programme in Health Data Science (Grant Ref: 218529/Z/19/Z). HZ acknowledges the receipt of Wellcome Cambridge Trust Scholarship. MW is grateful for the support of UCL Computer Science and The Alan Turing Institute. GN acknowledges support from the Medical Research Council Programme Leaders award MC_UP_A390_1107. CY is funded by a UKRI Turing AI Fellowship (Ref: EP/V023233/1). CH acknowledges support from the Medical Research Council Programme Leaders award MC_UP_A390_1107, The Alan Turing Institute, Health Data Research, U.K., and the U.K. Engineering and Physical Sciences Research Council through the Bayes4Health programme grant.

The authors report no competing interests.

We thank Tomas Lazauskas, Jim Madge and Oscar Giles from the Alan Turing Institute's Research Engineering team for their help and support. We thank Adam Huffman, Jonathan Diprose, Geoffrey Ferrari and Colin Freeman from the Biomedical Research Computing team at the University of Oxford for their help and support. We thank Angela Wood and Ben Cairns for their support and useful discussions.

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
