# A $J$ Independent Mixture of Gaussians prior on $z$

Let $p(\boldsymbol{z}_j)$ be the marginal distribution of $\boldsymbol{z}_j$ as follows

$$p(\boldsymbol{z}_j) = \sum_{c_j=1}^{K_j} p(c_j, \boldsymbol{z}_j) \tag{15}$$

$$= \sum_{c_j=1}^{K_j} p(c_j) p(\boldsymbol{z}_j | c_j) \tag{16}$$

$$= \sum_{c_j=1}^{K_j} p(c_j) \mathcal{N}(\boldsymbol{z}_j | \boldsymbol{\mu}_{c_j}, \boldsymbol{\Sigma}_{c_j}) \tag{17}$$

where $p(c_j)$ is a categorical distribution. Thus, $p(\boldsymbol{z}_j)$ is a Mixture-of-Gaussians (MoG).

Let us now derive $p(\vec{\boldsymbol{z}})$, the marginal distribution of $\vec{\boldsymbol{z}}$, as follows

$$p(\vec{\boldsymbol{z}}) = p(\boldsymbol{z}_1, \boldsymbol{z}_2, \ldots, \boldsymbol{z}_J) \tag{18}$$

$$= \prod_{j=1}^{J} p(\boldsymbol{z}_j) \tag{19}$$

$$= \prod_{j=1}^{J} \sum_{c_j=1}^{K_j} p(c_j) \mathcal{N}(\boldsymbol{z}_j | \boldsymbol{\mu}_{c_j}, \boldsymbol{\Sigma}_{c_j}) \tag{20}$$

where Eq. (19) follows from the indepdendence assumption of facets, and Eq. (20) uses Eq. (17). The resulting marginal of $\vec{\boldsymbol{z}}$ is our prior of $J$ independent MoGs.

**Linear (rather than exponential) complexity of number of clusters.** Besides its representational advantages, the multi-facet prior structure features a computational advantage: Given multiple known partitions of a dataset, the total number of clusters over all facets required to represent these partitions scales *linearly* w.r.t. the number of such partitions. In comparison, the number of clusters required in a single-facet model suffers from combinatorial explosion and scales *exponentially*.

To understand this, let us consider a hypothetical multi-partition image dataset of (rather standardised) hotel rooms which features $J$ facets $C_1, C_2, \ldots, C_J$ with $K_j$ possible discrete values for each characteristic, for example, the colour of the bed sheets, walls, interiors, whether a phone is present or not, the view of the room (beach, forest, city, . . . ). We now attempt to find reasonable clusters in this dataset. In principle, a single-partition model could learn all cross-combinations $C_1 \times C_2 \times \cdots \times C_J$. In general, this requires to learn "at least" $\mathcal{O}(\prod_{j=1}^{J} K_j)$ latent clusters [5]. Compare this with a multi-partition model such as MFCVAE. Here, we need to learn "at least" $\mathcal{O}(\sum_{j=1}^{J} K_j)$. If $K_j = K$ is equally large for all facets, the number of latent clusters to learn is $\mathcal{O}(K^J)$ for a single-partition model and $\mathcal{O}(K \cdot J)$ for a multi-partition model.

# B VaDE Trick Proofs

## B.1 Single-Facet VaDE Trick

*Proof.* (Theorem 1: Single-Facet VaDE Trick)

$$\mathbb{E}_{q_\phi(\mathbf{z}|\mathbf{x})} \mathrm{KL}\left[q_\phi(c|\mathbf{x}) || p_\theta(c|\mathbf{z})\right] = \mathbb{E}_{q_\phi(\mathbf{z}|\mathbf{x})} \mathbb{E}_{q_\phi(c|\mathbf{x})} \log \frac{q_\phi(c|\mathbf{x})}{p_\theta(c|\mathbf{z})} \tag{21}$$

$$= \mathbb{E}_{q_\phi(c|\mathbf{x})} \log \frac{q_\phi(c|\mathbf{x})}{\exp(\mathbb{E}_{q_\phi(\mathbf{z}|\mathbf{x})} \log p_\theta(c|\mathbf{z}))} \tag{22}$$

$$= \mathrm{KL}\left[q_\phi(c|\mathbf{x}) || \boldsymbol{\pi}(c|q_\phi(\mathbf{z}|\mathbf{x}))\right] - \log Z(q_\phi(\mathbf{z}|\mathbf{x})) \tag{23}$$

---

[5]This assumes that every cluster in the latent space corresponds to exactly one cross-combination of the data facets. Empirically, we find that for statistical reasons ("having more shots"), it can be desirable to have more latent clusters per facet than values possible for each facet.

which is minimised w.r.t. $q_\phi(c|\mathbf{x})$ by setting the KL term to zero by $q_\phi(c|\mathbf{x}) = \boldsymbol{\pi}(c|q_\phi(\mathbf{z}|\mathbf{x}))$, where

$$\boldsymbol{\pi}(c|q_\phi(\mathbf{z}|\mathbf{x})) := \frac{\exp\left(\mathbb{E}_{q_\phi(\mathbf{z}|\mathbf{x})}\log p(c|\mathbf{z})\right)}{Z(q_\phi(\mathbf{z}|\mathbf{x}))} \quad \text{for } c = 1, \ldots, K \tag{24}$$

$$Z(q_\phi(\mathbf{z}|\mathbf{x})) := \sum_{c=1}^{K} \exp\left(\mathbb{E}_{q_\phi(\mathbf{z}|\mathbf{x})}\log p(c|\mathbf{z})\right) \ . \tag{25}$$

as required. Here, $Z(q_\phi(\mathbf{z}|\mathbf{x}))$ is the appropriate normalization constant for Eq. (24) to define a probability mass function. $\qquad\square$

### B.1.1 Misapprehension in original statement

In the original paper, [10], they reach Eq. (5):

$$\mathcal{L}(\mathbf{x}; \theta, \phi) = \mathbb{E}_{q_\phi(\mathbf{z}|\mathbf{x})}\log p_\theta(\mathbf{x}|\mathbf{z}) - \mathrm{KL}\left[q_\phi(\mathbf{z}|\mathbf{x})||p_\theta(\mathbf{z})\right] - \underbrace{\mathbb{E}_{q_\phi(\mathbf{z}|\mathbf{x})}\mathrm{KL}\left[q_\phi(c|\mathbf{x})||p_\theta(c|\mathbf{z})\right]}_{\text{(A)}} \ .$$

The claim made (appendix A of [10]) is that $q(c|\mathbf{x}) = \mathbb{E}_{q_\phi(\mathbf{z}'|\mathbf{x})}p(c|\mathbf{z}')$ makes the final term, $\text{(A)} = \mathbb{E}_{q_\phi(\mathbf{z}|\mathbf{x})}\mathrm{KL}\left[q_\phi(c|\mathbf{x})||p_\theta(c|\mathbf{z})\right]$, equal to zero. Substituting this form for $q_\phi(c|\mathbf{x})$ in Eq. (26), we get:

$$
\begin{aligned}
\text{(A)} &= \mathbb{E}_{q_\phi(\mathbf{z}|\mathbf{x})}\mathrm{KL}\left[q_\phi(c|\mathbf{x})||p_\theta(c|\mathbf{z})\right] \\
&= \int \mathrm{d}\mathbf{z}\, q_\phi(\mathbf{z}|\mathbf{x})\sum_{c=1}^{K} q_\phi(c|\mathbf{x})\log\frac{q_\phi(c|\mathbf{x})}{p_\theta(c|\mathbf{z})} \\
&= \int \mathrm{d}\mathbf{z}\, q_\phi(\mathbf{z}|\mathbf{x})\sum_{c=1}^{K} \mathbb{E}_{q_\phi(\mathbf{z}'|\mathbf{x})}p(c|\mathbf{z}')\log\frac{\mathbb{E}_{q_\phi(\mathbf{z}''|\mathbf{x})}p(c|\mathbf{z}'')}{p_\theta(c|\mathbf{z})} \\
&= \int \mathrm{d}\mathbf{z}\, q_\phi(\mathbf{z}|\mathbf{x})\sum_{c=1}^{K} \left(\int \mathrm{d}\mathbf{z}'\, q_\phi(\mathbf{z}'|\mathbf{x})p(c|\mathbf{z}')\right)\log\frac{\int \mathrm{d}\mathbf{z}''\, q_\phi(\mathbf{z}''|\mathbf{x})p(c|\mathbf{z}'')}{p_\theta(c|\mathbf{z})} \\
&= \sum_{c=1}^{K} \underbrace{\left(\int \mathrm{d}\mathbf{z}'\, q_\phi(\mathbf{z}'|\mathbf{x})p(c|\mathbf{z}')\right)}_{\neq 0}\underbrace{\left[\log\int \mathrm{d}\mathbf{z}''\, q_\phi(\mathbf{z}''|\mathbf{x})p(c|\mathbf{z}'') - \int \mathrm{d}\mathbf{z}\, q_\phi(\mathbf{z}|\mathbf{x})\log p_\theta(c|\mathbf{z})\right]}_{\overset{?}{=}0}
\end{aligned}
\tag{26}
$$

$$\tag{27}$$

$$\overset{?}{=} 0$$

In the above derivations, we use $\mathbf{z}$, $\mathbf{z}'$ and $\mathbf{z}''$ to mark separate occurrences of the variable $\mathbf{z}$ in different integrals. The first term in Eq. (27) is strictly positive. To satisfy the claim, the second term in Eq. (27) would have to be equal to zero for all $c \in \{1, \ldots, K\}$, which in general does not hold. We note that in the original codebase for [10], training is not done using the form of the ELBO as above, Eq. (5) with $\text{(A)} = 0$, instead, using the general form where all terms are calculated.

**Monte Carlo Sampling of the VaDE-Trick objective:** These results and analysis raise the natural question: how is it that this misapprehension has lasted? Perhaps this is because of the following lucky accident when performing MC sampling.

If one substitutes the optimal forms of Theorem 1 back into the initial $\mathcal{L}$ and then estimates the resulting objective using a *single* MC sample from $\mathbf{z}$, then the resulting objective *looks* like is an estimator of Eq (5) with the final term set to zero, that is:

$$\mathcal{L}(\mathbf{x}; \theta, \phi) \overset{?}{=} \mathbb{E}_{q_\phi(\mathbf{z}|\mathbf{x})}\log p_\theta(\mathbf{x}|\mathbf{z}) - \mathrm{KL}\left[q_\phi(\mathbf{z}|\mathbf{x})||p_\theta(\mathbf{z})\right] \ .$$

Equivalently, in reverse, taking a *single* MC sample for $\mathbf{z}$ and using the above misapprehension as the training objective results in the same estimator as one gets from taking one MC sample for the true objective.

Let us push through the former of these, constructing the objective and MC estimator for the correct optimal objective:

$$\mathcal{L}(\mathbf{x}; \theta, \phi) = \mathbb{E}_{q_\phi(\mathbf{z}|\mathbf{x})q_\phi(c|\mathbf{x})} \left[ \log \frac{p_\theta(\mathbf{x}|\mathbf{z})p_\theta(\mathbf{z}, c)}{q_\phi(\mathbf{z}|\mathbf{x})q_\phi(c|\mathbf{x})} \right] \tag{28}$$

$$= \mathbb{E}_{q_\phi(\mathbf{z}|\mathbf{x})}[\log p_\theta(\mathbf{x}|\mathbf{z}) - \log q_\phi(\mathbf{z}|\mathbf{x})] - \mathbb{E}_{q_\phi(\mathbf{z}|\mathbf{x})} \mathbb{E}_{q_\phi(c|\mathbf{x})} \log \frac{q_\phi(c|\mathbf{x})}{p_\theta(\mathbf{z}, c)} \tag{29}$$

$$= \mathbb{E}_{q_\phi(\mathbf{z}|\mathbf{x})}[\log p_\theta(\mathbf{x}|\mathbf{z}) - \log q_\phi(\mathbf{z}|\mathbf{x})] - \mathbb{E}_{q_\phi(c|\mathbf{x})} \log \frac{q_\phi(c|\mathbf{x})}{\exp(\mathbb{E}_{q_\phi(\mathbf{z}|\mathbf{x})} \log p_\theta(\mathbf{z}, c))} \tag{30}$$

$$= \mathbb{E}_{q_\phi(\mathbf{z}|\mathbf{x})}[\log p_\theta(\mathbf{x}|\mathbf{z}) - \log q_\phi(\mathbf{z}|\mathbf{x})]$$
$$- \mathrm{KL}\left[q_\phi(c|\mathbf{x})||\pi(c|q_\phi(\mathbf{z}|\mathbf{x}))\right] + \log \vec{Z}(q_\phi(\mathbf{z}|\mathbf{x})) \tag{31}$$

where

$$\boldsymbol{\pi}(c|q_\phi(\mathbf{z}|\mathbf{x})) := \frac{\exp(\mathbb{E}_{q_\phi(\mathbf{z}|\mathbf{x})} \log p_\theta(\mathbf{z}, c))}{\vec{Z}(q_\phi(\mathbf{z}|\mathbf{x}))} \quad \text{for } c \in \mathcal{C} \tag{32}$$

$$\vec{Z}(q_\phi(\mathbf{z}|\mathbf{x})) := \sum_{c \in \mathcal{C}} \exp(\mathbb{E}_{q_\phi(\mathbf{z}|\mathbf{x})} \log p_\theta(\mathbf{z}, c)) . \tag{33}$$

Setting $q_\phi(c|\mathbf{x}) = \boldsymbol{\pi}(c|q_\phi(\mathbf{z}|\mathbf{x}))$ and substituting $\mathbf{z}^{(l)}$ for $l = 1, \ldots, L$ Monte Carlo samples from $q_\phi(\mathbf{z}|\mathbf{x})$:

$$\mathcal{L}(\mathbf{x}; \theta, \phi) \approx \frac{1}{L} \sum_{l=1}^{L} \log p_\theta(\mathbf{x}|\mathbf{z}^{(l)}) - \log q_\phi(\mathbf{z}^{(l)}|\mathbf{x})$$
$$+ \log \sum_{c \in \mathcal{C}} \exp\left( \frac{1}{L} \sum_{l=1}^{L} \log p_\theta(\mathbf{z}^{(l)}, c) \right) \tag{34}$$

which reduces for $L = 1$ to

$$\mathcal{L}(\mathbf{x}; \theta, \phi) \approx \log p_\theta(\mathbf{x}|\mathbf{z}^{(1)}) - \log q_\phi(\mathbf{z}^{(1)}|\mathbf{x}) + \log p_\theta(\mathbf{z}^{(1)}). \tag{35}$$

This ***appears*** to be a MC estimator for

$$\mathcal{L}(\mathbf{x}; \theta, \phi) = \mathbb{E}_{q_\phi(\mathbf{z}|\mathbf{x})} \log p_\theta(\mathbf{x}|\mathbf{z}) - \mathrm{KL}\left[q_\phi(\mathbf{z}|\mathbf{x})||p_\theta(\mathbf{z})\right]. \tag{36}$$

This appearance is purely because the $\log \sum \exp \sum \log$ in Eq (34) luckily simplifies when $L = 1$. Because of this lucky coincidence, all empirical results in [10] are valid.

## B.2 Multi-Facet VaDE Trick (factorised distribution)

*Proof.* (Theorem 2: Multi-Facet VaDE Trick for factorised distribution $q_\phi(\vec{\mathbf{z}}|\mathbf{x}) = \prod_j q_\phi(\mathbf{z}_j|\mathbf{x})$)

$$\mathbb{E}_{q_\phi(\vec{\mathbf{z}}|\mathbf{x})} \mathrm{KL}\left[q_\phi(\mathbf{c}|\mathbf{x})||p_\theta(\mathbf{c}|\vec{\mathbf{z}})\right] = \mathbb{E}_{q_\phi(\vec{\mathbf{z}}|\mathbf{x})} \mathbb{E}_{q_\phi(\mathbf{c}|\mathbf{x})} \log \frac{q_\phi(\mathbf{c}|\mathbf{x})}{p_\theta(\mathbf{c}|\vec{\mathbf{z}})} \tag{37}$$

$$= \mathbb{E}_{q_\phi(\mathbf{c}|\mathbf{x})} \log \frac{q_\phi(\mathbf{c}|\mathbf{x})}{\exp(\mathbb{E}_{q_\phi(\vec{\mathbf{z}}|\mathbf{x})} \log p_\theta(\mathbf{c}|\vec{\mathbf{z}}))} \tag{38}$$

$$= \mathbb{E}_{q_\phi(\mathbf{c}|\mathbf{x})} \log \frac{q_\phi(\mathbf{c}|\mathbf{x})}{\exp(\sum_j \mathbb{E}_{q_\phi(\mathbf{z}_j|\mathbf{x})} \log p_\theta(c_j|\mathbf{z}_j))} \tag{39}$$

$$= \mathbb{E}_{q_\phi(\mathbf{c}|\mathbf{x})} \log \frac{q_\phi(\mathbf{c}|\mathbf{x})}{\prod_j \exp(\mathbb{E}_{q_\phi(\mathbf{z}_j|\mathbf{x})} \log p_\theta(c_j|\mathbf{z}_j))} \tag{40}$$

$$= \mathrm{KL}\left[ q_\phi(\mathbf{c}|\mathbf{x})|| \prod_j \boldsymbol{\pi}_j(c_j|q_\phi(\mathbf{z}_j|\mathbf{x})) \right] - \sum_j \log Z_j(q_\phi(\mathbf{z}_j|\mathbf{x})) \tag{41}$$

which is minimised w.r.t. $q_\phi(c|\mathbf{x})$ by setting the KL term to zero by $q_\phi(c|\mathbf{x}) = \prod_j \boldsymbol{\pi}_j(c_j|q_\phi(\mathbf{z}_j|\mathbf{x}))$, where

$$\boldsymbol{\pi}_j(c_j|q_\phi(\mathbf{z}_j|\mathbf{x})) := \frac{\exp(\mathbb{E}_{q_\phi(\mathbf{z}_j|\mathbf{x})} \log p_\theta(c_j|\mathbf{z}_j))}{Z_j(q_\phi(\mathbf{z}_j|\mathbf{x}))} \text{ for } c_j = 1, \dots, K_j \tag{42}$$

$$Z_j(q_\phi(\mathbf{z}_j|\mathbf{x})) := \sum_{c_j=1}^{K_j} \exp(\mathbb{E}_{q_\phi(\mathbf{z}_j|\mathbf{x})} \log p_\theta(c_j|\mathbf{z}_j)) . \tag{43}$$

where $Z_j(q_\phi(\mathbf{z}_j|\mathbf{x}))$ is a normalisation constant for $\boldsymbol{\pi}_j(c_j|q_\phi(\mathbf{z}_j|\mathbf{x}))$, and we have used the relations

$$q_\phi(\vec{\mathbf{z}}|\mathbf{x}) = \prod_j q_\phi(\mathbf{z}_j|\mathbf{x}) \tag{44}$$

$$p_\theta(\mathbf{c}|\vec{\mathbf{z}}) = \prod_j p_\theta(c_j|\mathbf{z}_j) \tag{45}$$

as required. □

## B.3 Multi-Facet VaDE Trick (general distribution)

While we use a posterior over $\vec{\mathbf{z}}$ that factorises between facets, $q_\phi(\vec{\mathbf{z}}|\mathbf{x}) = \prod_{j=1}^{J} q_\phi(\mathbf{z}_j|\mathbf{x})$, there is the question as to whether one can use a VaDE trick in the case where the posterior for $\vec{\mathbf{z}}$ has a general factorisation (e.g. an autoregressive factorisation over facets). An example would be $q_\phi(\vec{\mathbf{z}}|\mathbf{x}) = \prod_{j=1}^{J} q_\phi(\mathbf{z}_j|\mathbf{z}_{<j}, \mathbf{x})$, the posterior factorisation used in many hierarchical VAEs [34, 37, 38]. We answer this question in the affirmative:

**Theorem 3.** *(Multi-Facet VaDE Trick for general $q_\phi(\vec{\mathbf{z}}|\mathbf{x})$, $p(\vec{\mathbf{z}}, \mathbf{c})$) For any probability distribution $q_\phi(\vec{\mathbf{z}}|\mathbf{x})$, the distribution $q_\phi(\mathbf{c}|\mathbf{x})$ that minimises $\mathbb{E}_{q_\phi(\vec{\mathbf{z}}|\mathbf{x})} \mathrm{KL}\left[q_\phi(\mathbf{c}|\mathbf{x})||p_\theta(\mathbf{c}|\vec{\mathbf{z}})\right]$ is*

$$\underset{q_\phi(\mathbf{c}|\mathbf{x})}{\operatorname{argmin}} \mathbb{E}_{q_\phi(\vec{\mathbf{z}}|\mathbf{x})} \mathrm{KL}\left[q_\phi(\mathbf{c}|\mathbf{x})||p_\theta(\mathbf{c}|\vec{\mathbf{z}})\right] = \vec{\pi}(\mathbf{c}|q_\phi(\vec{\mathbf{z}}|\mathbf{x})) \tag{46}$$

*where the minimum value is attained at*

$$\underset{q_\phi(\mathbf{c}|\mathbf{x})}{\operatorname{min}} \mathbb{E}_{q_\phi(\vec{\mathbf{z}}|\mathbf{x})} \mathrm{KL}\left[q_\phi(\mathbf{c}|\mathbf{x})||p_\theta(\mathbf{c}|\vec{\mathbf{z}})\right] = -\log \vec{Z}(q_\phi(\vec{\mathbf{z}}|\mathbf{x})) \tag{47}$$

*where*

$$\vec{\pi}(\mathbf{c}|q_\phi(\vec{\mathbf{z}}|\mathbf{x})) := \frac{\exp(\mathbb{E}_{q_\phi(\vec{\mathbf{z}}|\mathbf{x})} \log p_\theta(\mathbf{c}|\vec{\mathbf{z}}))}{\vec{Z}(q_\phi(\vec{\mathbf{z}}|\mathbf{x}))} \text{ for } \mathbf{c} \in \mathcal{C} \tag{48}$$

$$\vec{Z}(q_\phi(\vec{\mathbf{z}}|\mathbf{x})) := \sum_{\mathbf{c} \in \mathcal{C}} \exp(\mathbb{E}_{q_\phi(\vec{\mathbf{z}}|\mathbf{x})} \log p_\theta(\mathbf{c}|\vec{\mathbf{z}})) . \tag{49}$$

*Proof.*

$$\mathbb{E}_{q_\phi(\vec{\mathbf{z}}|\mathbf{x})} \mathrm{KL}\left[q_\phi(\mathbf{c}|\mathbf{x})||p_\theta(\mathbf{c}|\vec{\mathbf{z}})\right] = \mathbb{E}_{q_\phi(\vec{\mathbf{z}}|\mathbf{x})} \mathbb{E}_{q_\phi(\mathbf{c}|\mathbf{x})} \log \frac{q_\phi(\mathbf{c}|\mathbf{x})}{p_\theta(\mathbf{c}|\vec{\mathbf{z}})} \tag{50}$$

$$= \mathbb{E}_{q_\phi(\mathbf{c}|\mathbf{x})} \log \frac{q_\phi(\mathbf{c}|\mathbf{x})}{\exp(\mathbb{E}_{q_\phi(\vec{\mathbf{z}}|\mathbf{x})} \log p_\theta(\mathbf{c}|\vec{\mathbf{z}}))} \tag{51}$$

$$= \mathrm{KL}\left[q_\phi(\mathbf{c}|\mathbf{x})||\vec{\pi}(\mathbf{c}|q_\phi(\vec{\mathbf{z}}|\mathbf{x}))\right] - \log \vec{Z}(q_\phi(\vec{\mathbf{z}}|\mathbf{x})) \tag{52}$$

which is minimised by setting the KL term to zero as required. □

# C Monte Carlo estimator of Evidence Lower Bound

## C.1 Primary form

We start the derivation from the ELBO in Eq. (4):

$$\mathcal{L}(\mathbf{x}; \theta, \phi) = \mathbb{E}_{q_\phi(\vec{\mathbf{z}}|\mathbf{x})q_\phi(\mathbf{c}|\mathbf{x})}[\log \frac{p_\theta(\mathbf{x}|\vec{\mathbf{z}})p_\theta(\vec{\mathbf{z}}, \mathbf{c})}{q_\phi(\vec{\mathbf{z}}|\mathbf{x})q_\phi(\mathbf{c}|\mathbf{x})}] \tag{53}$$

$$= \mathbb{E}_{q_\phi(\vec{\mathbf{z}}|\mathbf{x})q_\phi(\mathbf{c}|\mathbf{x})}[\log \frac{p_\theta(\mathbf{x}|\vec{\mathbf{z}})p_\theta(\vec{\mathbf{z}}|\mathbf{c})p_\theta(\mathbf{c})}{q_\phi(\vec{\mathbf{z}}|\mathbf{x})q_\phi(\mathbf{c}|\mathbf{x})}] \tag{54}$$

$$= \mathbb{E}_{q_\phi(\vec{\mathbf{z}}|\mathbf{x})}[\log p_\theta(\mathbf{x}|\vec{\mathbf{z}})] + \mathbb{E}_{q_\phi(\vec{\mathbf{z}}|\mathbf{x})q_\phi(\mathbf{c}|\mathbf{x})}[\log p_\theta(\vec{\mathbf{z}} \mid \mathbf{c})]$$
$$+ \mathbb{E}_{q_\phi(\mathbf{c}|\mathbf{x})}[\log p_\theta(\mathbf{c})] - \mathbb{E}_{q_\phi(\vec{\mathbf{z}}|\mathbf{x})}[\log q_\phi(\vec{\mathbf{z}}|\mathbf{x})] - \mathbb{E}_{q_\phi(\mathbf{c}|\mathbf{x})}[\log q_\phi(\mathbf{c}|\mathbf{x})] \tag{55}$$

Next, we note that Theorem 2 has the optimal value of $q(\mathbf{c}|\mathbf{x})$ taking the factorised form

$$q(\mathbf{c}|\mathbf{x}) = \prod_{j=1}^{J} q(c_j|\mathbf{x}). \tag{56}$$

Combining this with the factorised prior introduced in Eq. (2), the loss can then be simplified and approximated as

$$\mathcal{L}(\mathbf{x}; \theta, \phi) \approx \mathbb{E}_{q_\phi(\vec{\mathbf{z}}|\mathbf{x})}[\log p_\theta(\mathbf{x}|\vec{\mathbf{z}})] + \sum_{j=1}^{J} \mathbb{E}_{q_\phi(\mathbf{z}_j|\mathbf{x})q_\phi(c_j|\mathbf{x})}[\log p_\theta(\mathbf{z}_j \mid c_j)]$$
$$+ \sum_{j=1}^{J} \mathbb{E}_{q_\phi(c_j|\mathbf{x})}[\log p_\theta(c_j)] - \sum_{j=1}^{J} \mathbb{E}_{q_\phi(\mathbf{z}_j|\mathbf{x})}[\log q_\phi(\mathbf{z}_j|\mathbf{x})] - \sum_{j=1}^{J} \mathbb{E}_{q_\phi(c_j|\mathbf{x})}[\log q_\phi(c_j|\mathbf{x})] \tag{57}$$

Then we approximate the ELBO using MC estimation by drawing samples from $q_\phi(\vec{\mathbf{z}}|\mathbf{x})$:

$$\tilde{\mathcal{L}}(\mathbf{x}; \theta, \phi) = \frac{1}{L} \sum_{l=1}^{L} \log p_\theta(\mathbf{x}|\vec{\mathbf{z}}^{(l)}) + \frac{1}{L} \sum_{l=1}^{L} \sum_{j=1}^{J} \sum_{c_j=1}^{K_j} q_\phi(c_j|\mathbf{x}) \log p_\theta(\mathbf{z}_j^{(l)}|c_j)$$
$$+ \sum_{j=1}^{J} \sum_{c_j=1}^{K_j} q_\phi(c_j|\mathbf{x}) \log p_\theta(c_j) - \frac{1}{L} \sum_{l=1}^{L} \sum_{j=1}^{J} \log q_\phi(\mathbf{z}_j^{(l)}|\mathbf{x}) - \sum_{j=1}^{J} \sum_{c_j=1}^{K_j} q_\phi(c_j|\mathbf{x}) \log q_\phi(c_j|\mathbf{x}) \tag{58}$$

where the optimal value of $q_\phi(c_j|\mathbf{x})$ is obtained from Theorem 2:

$$q_\phi(c_j|\mathbf{x}) := \frac{\exp\left[\frac{1}{L} \sum_{l=1}^{L} \log p_\theta(c_j|\mathbf{z}_j^{(l)})\right]}{Z_j(q_\phi(\mathbf{z}_j|\mathbf{x}))} \quad \text{for } c_j = 1, \dots, K_j \tag{59}$$

$$Z_j(q_\phi(\mathbf{z}_j|\mathbf{x})) := \sum_{c_j=1}^{K_j} \exp\left[\frac{1}{L} \sum_{l=1}^{L} \log p_\theta(c_j|\mathbf{z}_j^{(l)})\right] . \tag{60}$$

which reduces to

$$q_\phi(c_j|\mathbf{x}) = p_\theta(c_j|\mathbf{z}_j^{(1)}) \text{ for } j = 1, \dots, J \tag{61}$$

when $L = 1$.

As a result, we obtain the loss for $L = 1$:

$$\tilde{\mathcal{L}}(\mathbf{x}; \theta, \phi) = \log p_\theta(\mathbf{x}|\vec{\mathbf{z}}^{(1)}) + \sum_{j=1}^{J} \sum_{c_j=1}^{K_j} p_\theta(c_j|\mathbf{z}_j^{(1)})\left(\log p_\theta(\mathbf{z}_j^{(1)}|c_j) + \log p_\theta(c_j)\right)$$
$$- \sum_{j=1}^{J} \log q_\phi(\mathbf{z}_j^{(1)}|\mathbf{x}) - \sum_{j=1}^{J} \sum_{c_j=1}^{K_j} p_\theta(c_j|\mathbf{z}_j^{(1)}) \log p_\theta(c_j|\mathbf{z}_j^{(1)}) \tag{62}$$

## C.2 Alternate form

Again, we start the derivation of an MC estimator from Eq. (4):

$$\mathcal{L}(\mathbf{x}; \theta, \phi) = \mathbb{E}_{q_\phi(\vec{\mathbf{z}}|\mathbf{x})q_\phi(\mathbf{c}|\mathbf{x})}[\log \frac{p_\theta(\mathbf{x}|\vec{\mathbf{z}})p_\theta(\vec{\mathbf{z}}, \mathbf{c})}{q_\phi(\vec{\mathbf{z}}|\mathbf{x})q_\phi(\mathbf{c}|\mathbf{x})}] \tag{63}$$

$$= \mathbb{E}_{q_\phi(\vec{\mathbf{z}}|\mathbf{x})q_\phi(\mathbf{c}|\mathbf{x})}[\log \frac{p_\theta(\mathbf{x}|\vec{\mathbf{z}})p_\theta(\vec{\mathbf{z}})p_\theta(\mathbf{c}|\vec{\mathbf{z}})}{q_\phi(\vec{\mathbf{z}}|\mathbf{x})q_\phi(\mathbf{c}|\mathbf{x})}] \tag{64}$$

$$= \mathbb{E}_{q_\phi(\vec{\mathbf{z}}|\mathbf{x})} \left[ \log p_\theta(\mathbf{x}|\vec{\mathbf{z}}) - \log q_\phi(\vec{\mathbf{z}}|\mathbf{x}) + \log p_\theta(\vec{\mathbf{z}}) \right]$$
$$+ \mathbb{E}_{q_\phi(\mathbf{c}|\mathbf{x})} \left[ \mathbb{E}_{q_\phi(\vec{\mathbf{z}}|\mathbf{x})} \log p_\theta(\mathbf{c} \mid \vec{\mathbf{z}}) - \log q_\phi(\mathbf{c}|\mathbf{x}) \right] \tag{65}$$

where an alternative factorisation of $p_\theta(\vec{\mathbf{z}}, \mathbf{c})$ is used in Eq. (64), resulting in a different, but equivalent formulation of the ELBO in Eq. (55).

Next, we draw MC samples from $q_\phi(\vec{\mathbf{z}}|\mathbf{x})$:

$$\tilde{\mathcal{L}}(\mathbf{x}; \theta, \phi) = \frac{1}{L} \sum_{l=1}^{L} \left[ \log p_\theta(\mathbf{x}|\vec{\mathbf{z}}^{(l)}) - \log q_\phi(\vec{\mathbf{z}}^{(l)}|\mathbf{x}) + \log p_\theta(\vec{\mathbf{z}}^{(l)}) \right] \tag{66}$$

$$+ \sum_{\mathbf{c} \in \mathcal{C}} \left\{ q_\phi(\mathbf{c}|\mathbf{x}) \left[ \frac{1}{L} \sum_{l=1}^{L} \log p_\theta(\mathbf{c} \mid \vec{\mathbf{z}}^{(l)}) - \log q_\phi(\mathbf{c}|\mathbf{x}) \right] \right\} \tag{67}$$

where the optimal value of $q_\phi(\mathbf{c}|\mathbf{x})$ when $L = 1$ is similarly obtained from Theorem 2:

$$q_\phi(\mathbf{c}|\mathbf{x}) = \prod_{j=1}^{J} q_\phi(c_j|\mathbf{x}) \tag{68}$$

$$q_\phi(c_j|\mathbf{x}) = p_\theta(c_j|\mathbf{z}_j^{(1)}) \text{ for } j = 1, \ldots, J \tag{69}$$

In this case, the term in (67) evaluates to zero, because from Theorem 2

$$\log p_\theta(\mathbf{c} \mid \vec{\mathbf{z}}^{(1)}) = \sum_{j=1}^{J} \log p_\theta(c_j \mid \mathbf{z}_j^{(1)}) = \sum_{j=1}^{J} \log q_\phi(c_j|\mathbf{x}) = \log q_\phi(\mathbf{c}|\mathbf{x}). \tag{70}$$

Consequently, we obtain the loss for $L = 1$:

$$\tilde{\mathcal{L}}(\mathbf{x}; \theta, \phi) = \log p_\theta(\mathbf{x}|\vec{\mathbf{z}}^{(1)}) - \log q_\phi(\vec{\mathbf{z}}^{(1)}|\mathbf{x}) + \log p_\theta(\vec{\mathbf{z}}^{(1)}) \tag{71}$$

where

$$q_\phi(\vec{\mathbf{z}}^{(1)}|\mathbf{x}) = \prod_{j=1}^{J} q_\phi(\mathbf{z}_j^{(1)}|\mathbf{x}) \tag{72}$$

$$p_\theta(\vec{\mathbf{z}}^{(1)}) = \prod_{j=1}^{J} p_\theta(\mathbf{z}_j^{(1)}) = \prod_{j=1}^{J} \sum_{c_j=1}^{K_j} p_\theta(\mathbf{z}_j^{(1)}|c_j)p_\theta(c_j) \tag{73}$$

## C.3 Empirical comparison of primary and alternate form

Here, we empirically compare the primary and alternate form with five and three terms, respectively. Each loss comes from a different factorization of $p_\theta(\vec{\mathbf{z}}, \mathbf{c})$ as we show above, but are equivalent.

We verified in our implementation that both losses yield the exact same loss values on the same mini-batch, but gradients computed during optimisation are different as both losses have non-overlapping terms and consequently convergence behavior may differ during training. We are interested in whether these differences are substantial. In particular, [10] used a 5-term loss function (similar to the primary loss in Eq. (62), even though a 3-term loss function (similar to the alternate loss) could also be obtained and is arguably more compact.

We investigate this question with the following experimental setup, which is close to the one in [10]: On MNIST, we conduct 10 training runs of our model with varying random seeds for both the primary and alternate form of the loss. We use the following hyperparameters with a shared architecture and refer to Appendix D for a more detailed understanding on these configurations:

- Number of facets: $J = 1$
- Batch size: 512
- Learning rate: 0.002
- Dimension of $\mathbf{z}$: 10
- Number of $c$ (number of clusters): 50
- Covariance structure of $p_\theta(\mathbf{z}|c)$: diagonal
- Output dimensions for layers in $g(\mathbf{x}; \phi)$: $[500, 500, 2000]$
- Output dimensions for layers in $f(\mathbf{z}; \theta)$: $[2000, 500, 500]$

In Fig. 7, we show unsupervised clustering accuracy on the test set over training epochs for the 10 runs and the primary (left) and alternate (right) form of the loss. Our results indicate that there is no significant difference in performance between the two loss forms. We decide to use the primary form in all our experiments going forward as it is simpler for our implementation, e.g. when combined with progressive training, and is also more intuitively following the generative process of our model.

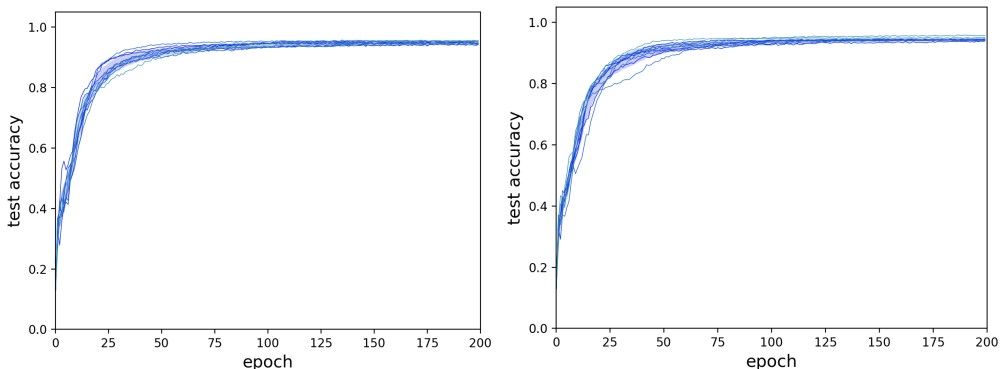

Figure 7: Unsupervised clustering accuracy on the test set for 10 runs, comparing the primary (left) and alternate (right) loss form. Each run is illustrated by one curve. The blue shade is bounded by the mean accuracy plus and minus one standard deviation across the ten runs.

# D  Experimental details

We provide our code implementing MFCVAE, using *PyTorch Distributions* [26], together with detailed instructions setup, training and evaluating our model, as well as reproducing the main results of this paper via shell scripts at https://github.com/FabianFalck/mfcvae.

### D.1  Datasets and preprocessing

Throughout our experiments, we use three datasets: MNIST [5], 3DShapes [24], and SVHN [25]. In the following, we briefly introduce these datasets, particularly focusing on their abstract characteristics which might be separated out by a multi-facet clustering model, as well as ethical considerations with regards to their collection. For MNIST and SVHN, we use their implementations as PyTorch Dataset classes as part of the torchvision package to process [26]. For 3DShapes, we provide a custom PyTorch Dataset class which contains several preprocessing steps (detailed below) and the selection of arbitrary combinations of factors.

**MNIST.** The MNIST database [5] consists of grey-scale (almost binary) handwritten digits from 10 classes ('0' to '9'). There are 60,000 training examples and 10,000 test examples. The handwritten digits are written by 500 writers (250 writers for training and test set, respectively), introducing a large variation in terms of style of these characters. The most prominent characteristics of MNIST are 1) the digit class, given as a supervised label 2) stroke width (e.g. 'bold', 'thin', ...) 3) the slant

of the digits (e.g. 'right-tilted', 'left-tilted', 'upright', ...). During preprocessing, we transform the images to a 0 to 1 scale using min-max scaling.

To the best of our knowledge, the dataset is highly curated and cropped to individual digits, so that we can exclude offensive content or important personally identifiable information in these images. However, we note that as the images are handwritten, there is a possibility that they can be linked to these individuals.

MNIST "was constructed from NIST's Special Database 3 [SD-3] and Special Database 1 [SD-1]" [5]. To the best of our knowledge, SD-3 and SD-1 are no longer available for download (see `https://www.nist.gov/srd/shop/special-database-catalog`), as opposed to other Special Databases. We thus cannot comment on whether and if so in what form consent was obtained from subjects providing the handwritten digits.

**3DShapes.** The 3DShapes dataset [24] consists of images of three-dimensional shapes in front of a background, generated from six independent ground truth latent factors. These latent factors are floor colour (10 values), wall colour (10 values), object colour (10 values), scale (8 values), shape (4 values), and orientation (15 values). Since all ground truth latent factors are discrete, the nature of this dataset makes it particularly suited for a multi-facet clustering task.

The dataset is preprocessed as follows: We transform each factor's values to a scale of integers between 0 and the number of values of that factor minus one. Then, to be consistent with the SVHN dataset, we resize the original $64 \times 64$ images to the size $32 \times 32$ using bilinear interpolation. Lastly, we transform the images to a 0 to 1 scale using min-max scaling.

From this 3DShapes dataset, we extract the following 2 configurations which are used during our experiments (note that other configurations can be easily created using our provided Dataset class):

- *Configuration 1* (4,800 images): 10 values for floor colour, 1 value for wall colour, 1 value for object colour, 8 values for scale, 4 values for shape, 15 values for orientation
- *Configuration 2* (4,800 images): 1 value for floor colour, 10 values for wall colour, 1 value for object colour, 8 values for scale, 4 values for shape, 15 values for orientation

3DShapes is a simulated dataset. The dataset was generated using the QUery Networks Mujocu environment [51].

**SVHN.** SVHN [25] is a real-world image dataset of cropped digits obtained of house numbers in Google Street View images. We focus on the 73,257 training digits and the 26,032 test digits available in the `torchvision` package in PyTorch. SVHN is similar to MNIST in the sense that it is a labelled digit dataset, however, was collected with the aim of being significantly more complex and diverse: As the images were extracted from random Google Street View images in various countries, for example they have varying backgrounds, different number of digits per image (the central digit is used as the label), varying resolutions, and different digit styles, rendering them a challenging dataset for supervised and unsupervised learning tasks, and an interesting test bed for multi-facet clustering, as for some of these characteristics, it might be possible to separate them out. During preprocessing, we transform the images to a 0 to 1 scale using min-max scaling.

### D.2   Neural architectures and Variational Ladder Autoencoder

In the following, we define two architectures which we implemented in our experiments:

- A Variational Ladder Autoencoder (VLAE) architecture, as defined in [22] and illustrated in Fig. 8.
- A shared encoder and decoder architecture (we refer to it as "shared architecture" in the following), illustrated in Fig. 9.

**VLAE architecture.**   The VLAE architecture consists of an encoder (recognition model) and a decoder (generative model) which are symmetric to each other. Both encoder and decoder have a set of backbone layers $b_j^{\text{enc}}$ and $b_j^{\text{dec}}$, respectively, which share parameters across the layers of latent variables, and thus naturally build a hierarchy of abstractions. From and into these backbones, a set of rung layers $r_j^{\text{enc}}$ and $r_j^{\text{dec}}$ emerge which parameterise the latent variables $\vec{z}$ (encoder) and process their samples towards reconstructions (decoder).

Formally, following the notation in [22], we define the recognition model as

$$\hat{\mathbf{h}}_j = \mathbf{b}_j^{\text{enc}}(\hat{\mathbf{h}}_{j-1}) \tag{74}$$

$$\mathbf{z}_j \sim \mathcal{N}\left(\mathbf{z}_j; r_{j,\mu}^{\text{enc}}(\hat{\mathbf{h}}_j), r_{j,\sigma^2}^{\text{enc}}(\hat{\mathbf{h}}_j)\right) \tag{75}$$

where $j = 1, \ldots, J$; $b_j^{\text{enc}}$ and $r_j^{\text{enc}}$ are neural networks, $r_{j,\mu}^{\text{enc}}$ and $r_{j,\sigma^2}^{\text{enc}}$ refer to the elements of the output vector of $r_j$ corresponding to the mean and variance of the parameterised Gaussian distribution $q(\mathbf{z}_j \mid \mathbf{x})$ with diagonal covariance matrix, and $h_0 \equiv \mathbf{x}$.

For each $j$, $q(c_j \mid \mathbf{x})$ is not directly parameterised by neural networks, but instead computed by Theorem 2 as described in Section 3.1.

We define the generative model as

$$c_j \sim p(c_j), \text{ for } j = 1, \ldots, J \tag{76}$$

$$\mathbf{z}_j \sim p(\mathbf{z}_j|c_j), \text{ for } j = 1, \ldots, J \tag{77}$$

$$\tilde{\mathbf{z}}_J = b_J^{\text{dec}} \circ r_J^{\text{dec}}(\mathbf{z}_J) \tag{78}$$

$$\tilde{\mathbf{z}}_j = b_j^{\text{dec}}\left(\left[\tilde{\mathbf{z}}_{j+1}, r_j^{\text{dec}}(\mathbf{z}_j)\right]\right), \text{ for } j = 1, \ldots, J-1 \tag{79}$$

$$\mathbf{x} \sim u(\mathbf{x}; \tilde{\mathbf{z}}_1) \tag{80}$$

where $b_j^{\text{enc}}$ and $r_j^{\text{enc}}$ are neural networks, $[\cdot, \cdot]$ denotes concatenation of two vectors, and $u(\mathbf{x})$ is the likelihood model of $\mathbf{x}$.

We refer to Appendix D.4 for the exact implementation of all neural networks $b_j^{\text{enc}}$, $r_j^{\text{enc}}$, $b_j^{\text{dec}}$, and $r_j^{\text{dec}}$ and the likelihood model $u(\cdot)$ for each of the three datasets.

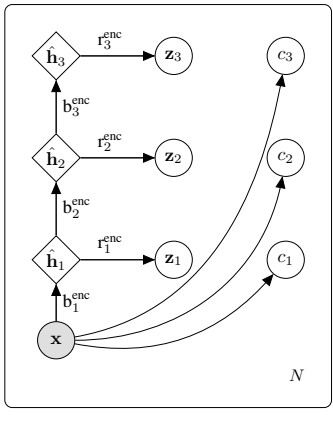
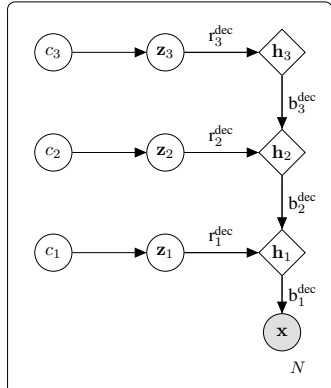

|  (a) Recognition model  |  (b) Generative Model  |

Figure 8: Ladder-MFCVAE architecture with $J = 3$ as an example. (a) The recognition model and (b) generative model. Each *labelled* arrow corresponds to a neural network. The posterior for each $c_j$ is defined using the multi-facet VaDE trick.

**Shared architecture.** We use the shared architecture as a simple comparison to test our hypothesis that a VLAE helps stabilise training. In the shared architecture, each facet has an equal depth of neural networks and shares the parameters. The encoder and decoder are both fully shared, except for the last hidden layers in both.

More precisely, the recognition model is defined as

$$\hat{\mathbf{h}} = s^{\text{enc}}(\mathbf{x}) \tag{81}$$

$$\mathbf{z}_j \sim \mathcal{N}\left(\mathbf{z}_j; t_{j,\mu}^{\text{enc}}(\hat{\mathbf{h}}), t_{j,\sigma^2}^{\text{enc}}(\hat{\mathbf{h}})\right) \tag{82}$$

where $s^{\text{enc}}$ and each $t_j^{\text{enc}}$ are neural networks, and $t_{j,\mu}^{\text{dec}}$ and $t_{j,\sigma^2}^{\text{dec}}$ again refer to those elements of the output vector of $t_j$ corresponding to the mean and variance of the parameterised Gaussian distribution $q(\mathbf{z}_j \mid \mathbf{x})$ with diagonal covariance matrix. $q(c_j \mid \mathbf{x})$ is computed by Theorem 2 as described in Section 3.1.

The generative model is defined as

$$c_j \sim p(c_j), \text{ for } j = 1, \ldots, J \tag{83}$$

$$\mathbf{z}_j \sim p(\mathbf{z}_j | c_j), \text{ for } j = 1, \ldots, J \tag{84}$$

$$\mathbf{h} = t^{\text{dec}}([\mathbf{z}_1, \ldots, \mathbf{z}_J]) \tag{85}$$

$$\tilde{\mathbf{x}} = s^{\text{dec}}(\mathbf{h}) \tag{86}$$

$$\mathbf{x} \sim u(\tilde{\mathbf{x}}) \tag{87}$$

where $t^{\text{dec}}$ and $s^{\text{dec}}$ are neural networks, $[\cdot, \ldots, \cdot]$ refers to vector concatenation, and $u(\cdot)$ is the likelihood model of $\mathbf{x}$.

Again, we refer to Appendix D.4 for the exact implementation of all neural networks $s^{\text{enc}}$, $t_j^{\text{enc}}$, $s^{\text{dec}}$, and $t^{\text{dec}}$, as well as for the likelihood model $u(\cdot)$ for each of the three datasets.

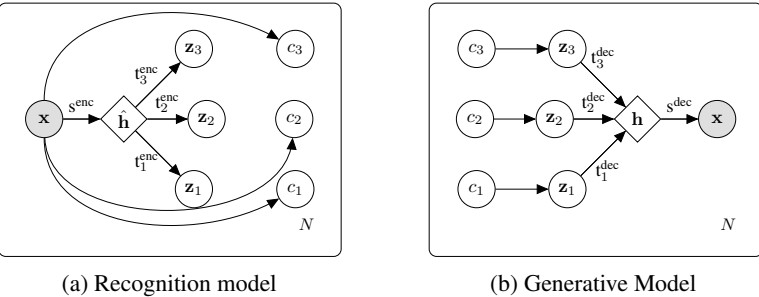

(a) Recognition model          (b) Generative Model

Figure 9: Shared encoder and decoder MFCVAE architecture with $J = 3$ as an example. (a) The recognition model and (b) generative model. Each labelled arrow corresponds to a neural network. The posterior for each $c_j$ is defined using the multi-facet VaDE trick.

### D.3 Progressive training algorithm

We use a progressive training algorithm to train our VLAE architectures. We strongly base its implementation on [23] and refer to this source for a more complete introduction, but will point out differences to this formulation below.

The idea of progressive training is to start with training a single facet (typically the one of highest depth in the VLAE architecture), and then progressively loop in the other facets one after the other in a smooth manner. To formalise this, we define a progressive step $s = 1, 2, \ldots, J - 1$, where in step $s$, facets $J - s + 1$ to $J$ (both including) are contributing to the network (and might be currently looped in), and $\alpha_j$, the fade-in coefficient of layer $j$. $\alpha_j$ linearly increases from 0.0 to 1.0 during the first 15,000 (for MNIST and SVHN) or 2,000 (for 3DShapes) batches of a progressive step (except for $s = 1$), is 0.0 if the facet has not yet been looped in, and is 1.0, otherwise. [23] used 5,000, but we increased this number for MNIST and SVHN to have a smoother loop-in of facets.

In contrast to the formulation in [23] which excludes from the model latent facets that are not looped in yet in a certain progressive step, in our formulation, all latent facets are part of the model throughout all progressive steps, yet do not contribute to the KL-divergences or the reconstruction term in Eq. (14). We achieve this by applying the fade-in coefficient only to the decoder rungs, not the encoder rungs (compare Eq. (9) in [23], where both the encoder and decoder rungs are faded in); and to weigh the KL-divergences in $\mathbf{z}_j$ and $c_j$, as before. In other words, this is similar to the implementation of progressive training in [23] with $\alpha_j = 1.0$ for the encoder rungs throughout all progressive steps, and the regular, smoothly increasing $\alpha_j$ value for the decoder rungs. Precisely, to implement the progressive training algorithm, we amend Eq. (79) as follows:

$$\tilde{\mathbf{z}}_j = b_j^{\text{dec}} \left( [\tilde{\mathbf{z}}_{j+1}, \alpha_j r_j^{\text{dec}}(\mathbf{z}_j)] \right) \quad , \text{ for } j = 1, \ldots, J, \tag{88}$$

$$\mathcal{L}^{\text{MFCVAE}}(\mathcal{D}; \theta, \phi) = \mathbb{E}_{\mathbf{x} \sim \mathcal{D}} \Bigg[ \mathbb{E}_{q_\phi(\vec{\mathbf{z}}|\mathbf{x})} \log p_\theta(\mathbf{x}|\vec{\mathbf{z}})$$

$$- \sum_{j=1}^{J} \alpha_j \left[ \mathbb{E}_{q_\phi(c_j|\mathbf{x})} \text{KL}(q_\phi(\mathbf{z}_j|x)||p_\theta(\mathbf{z}_j|c_j)) + \text{KL}(q_\phi(c_j|\mathbf{x})||p(c_j)) \right] \Bigg] \tag{89}$$

where

$$\alpha_j = 1.0, \quad \text{for } j = (J - s + 1), \dots, J \tag{90}$$

$$\alpha_{J-s} \in [0, 1] \quad \text{(looped in)} \tag{91}$$

$$\alpha_j = 0.0, \quad \text{for all } j = 1, \dots, (J - s - 1) \tag{92}$$

and all other equations of the VLAE remain unchanged. Thus, when $\alpha_j = 0.0$, the gradient w.r.t. any parameters in $b_j^{\text{enc}}, r_j^{\text{enc}}, b_j^{\text{dec}}, r_j^{\text{dec}}$, as well as the parameters of the priors $p(c_j)$ and $p(\mathbf{z}_j)$ are 0, and we achieve the same effect as if those components would not be part of the model.

Lastly, while we have tested "pretraining" the latent facets which are not looped in yet through a KL-regularisation terms in $\mathbf{z}_j$ and $c_j$ (see Eq. (10) in [23]), we could not see a beneficial effect on stability of model training in our model. As this would add complexity to the training algorithm, we do not pursue this type of pretraining here.

### D.4 Implementation details and hyperparameters

**New assets.** We publish the following new assets accompanying this paper:

- *Code*: We provide our source code with detailed instructions on setup, reproducing our results via shell scripts, training, evaluation in a README file at `https://github.com/FabianFalck/mfcvae`. We follow the code templates and NeurIPS guidelines for code submissions. Our code is provided under MIT license. The initial implementation of our model are inspired by the codebase of VaDE in `https://github.com/eelxpeng/UnsupervisedDeepLearning-Pytorch/blob/master/udlp/clustering/vade.py`, as well as the official VLAE implementation ([22] and `https://github.com/ermongroup/Variational-Ladder-Autoencoder`), and the official ProVLAE implementation ([23] and `https://github.com/Zhiyuan1991/proVLAE/blob/master/model_ladder_progress.py`). Our convolutional VLAE architecture is largely based on the neural architecture for the CelebA dataset in the ProVLAE codebase mentioned above.
- *Pretrained models*: We further provide pretrained models with the hyperparameters reported in this section as part of the folder `pretrained_models/`.

**Existing assets used.** Our work uses the following Python software packages with accompanying licenses (if known): PyTorch [26] (in particular the PyTorch Distributions and Torchvision packages; custom license), Numpy [52] (BSD 3-Clause License), Weights&Biases [53] (MIT License), Matplotlib [54] (PSF License), Seaborn [55] (BSD 3-Clause License), Pickle [56] (N/A), H5Py [57] (BSD 3-Clause License), OpenCV 2 [58] (Apache License), Scikit-learn [59] (BSD 3-Clause License), boilr (`https://github.com/addtt/boiler-pytorch`) (MIT License). Regarding data assets used, we refer to Section D.1.

**Data splits.** For all three datasets, we split data into training and test dataset (no validation dataset used). For MNIST and SVHN, we use the standard data splits as provided with these datasets and in the TorchVision PyTorch package. For 3DShapes, in both configurations, we use 80% for training and the remaining 20% for testing. Here, we sample the images uniformly at random and without replacement. We also refer to Appendix D.1 for a more detailed discussion on preprocessing of these datasets.

**Likelihood models.** For MNIST data, we define its likelihood $p(\mathbf{x}|\vec{\mathbf{z}})$ as a product of independent Bernoulli likelihoods, where each dimension is Bernoulli-distributed with respect to some learnt parameter and independent of other dimensions. Bernoulli likelihood is a reasonable assumption, because most pixels in MNIST images have values close or equal to 0 and 1.

For 3DShapes and SVHN data, we define their likelihood $p(\mathbf{x}|\vec{\mathbf{z}})$ to be a product of independent Gaussian likelihoods, where each dimension is Gaussian-distributed with its mean learnt as a parameter and its variance fixed as a hyperparameter.

**Other design choices.** Apart from the neural architectures, hyperparameters and likelihood models of MFCVAE, there were other design choices which were made on a per-dataset basis:

- *Covariance structure of the Gaussian $p(\mathbf{z}_j|c_j)$ for each $j$ and $c_j$*: The covariance matrices can be set to either diagonal or full. In this paper, diagonal covariance is found to be sufficient for MNIST. Full covariance is chosen for 3DShapes and SVHN as it results in a stronger disentanglement of facets.

- *Whether to fix $\boldsymbol{\pi}_j$ or train them as parameters*: In order to encourage clusters to have similar sizes in each facet, one option is to fix $\boldsymbol{\pi}_j$ to be $1/K_j$ componentwise for each facet $j$. We fix $\boldsymbol{\pi}_j$ in models trained on 3DShapes and SVHN.
- *Activation functions:* To avoid vanishing gradients and encourage a more stable training, we tested three activation functions, in particular, ReLU, leaky ReLU and ELU. For MNIST, we found ReLU to be sufficient. For 3DShapes and SVHN, where convolutional neural networks are involved, we sometimes observed vanishing gradients in training runs, which is why we used the ELU activation function where we no longer observed this problem (leaky ReLU likewise worked, but we chose ELU for consistency with previous VLAE implementations mentioned above).

**Hyperparameter tuning.** We performed several large exploratory hyperparameter sweeps over wide grids of possible hyperparameter values, looking at the qualitative improvement in facet disentanglement and, where available, the training accuracy of supervised labels (often only one label of the two facets of interest available). In these exploratory hyperparameter sweeps, we observed the following hyperparameter patterns that generalise across all datasets:

We noticed that results are stable w.r.t. to a large set of hyperparameters and ranges of possible values. In particular, this applies to batch size, learning rate, the number of batches used during fade-in, and to some degree the number of clusters in both facets. However, we noticed that some hyperparameters must be set be rather carefully to achieve strong disentanglement between facets. In particular, we find that the style/colour facet's latent dimension must be rather precisely set to a narrow range of values yielding strong disentanglement of facets: For MNIST, $\dim(\mathbf{z}_1)$ has to be around 5. For SVHN, $\dim(\mathbf{z}_2)$ has to be around 5. For 3DShapes, $\dim(\mathbf{z}_2)$ has to be around 2.

**Hyperparameters of the reported results.** In the following, we report the hyperparameters for training our models reported and presented in Section 4 (note that the hyperparameters for the models trained on the two different 3DShapes configurations are the same). In Table 3, we report chosen values of scalar hyperparameters. For full details on each of these hyperparameters, we refer to our code and in particular the help message of the respective command line arguments in the training script.

Table 3: Scalar hyperparameters and design choices for our three model configurations on MNIST, 3DShapes and SVHN, with results presented in Section 4 and Appendix E.

|  | MNIST | 3DShapes | SVHN |
| --- | --- | --- | --- |
| Batch size | 512 | 150 | 150 |
| Learning rate | 0.0005 | 0.0003 | 0.0005 |
| Latent dimension of the first facet, $\dim(\mathbf{z}_1)$ | 5 | 20 | 22 |
| Number of clusters in the first facet, $\dim(c_1)$ | 25 | 60 | 200 |
| Latent dimension of the second facet, $\dim(\mathbf{z}_2)$ | 5 | 2 | 7 |
| Number of clusters in the second facet, $\dim(c_2)$ | 25 | 20 | 50 |
| Number of training batches for each fade-in | 15000 | 2000 | 15000 |
| Likelihood model for $p(\mathbf{x}|\vec{\mathbf{z}})$ | Bernoulli | Gaussian | Gaussian |
| Standard deviation of $p(\mathbf{x}|\vec{\mathbf{z}})$ componentwise (if Gaussian) | N/A | 0.6 | 0.3 |
| Data dependent initialisation for $g(\mathbf{x};\phi)$ and $f(\vec{\mathbf{z}};\theta)$ | No | Yes | Yes |
| Covariance structure of $p(\mathbf{z}_j|c_j)$ | diagonal | full | full |
| Fix $\boldsymbol{\pi}_j$ | No | Yes | Yes |
| Diagonal entries during initialisation for covariance of $p(\mathbf{z}_j|c_j)$ | 0.01 | 0.01 | 0.01 |
| Activation function | ReLU | ELU | ELU |

**Neural architectures.** Following up Appendix D.2, we here provide the detailed initialisation of hidden layers of our neural architectures.

We first discuss the fully-connected ladder architecture which we use to train MFCVAE on MNIST, with results presented in Section 4 and Appendix E. We initialise the VLAE architecture as detailed in Table 4.

Next, we describe the convolutional ladder architecture which we use to train MFCVAE on 3DShapes and SVHN, with results presented in Section 4 and Appendix E. We initialise the VLAE architecture as detailed in Table 5.

Table 4: Details of the fully-connected ladder architecture for our model trained on MNIST and reported in Section 4 and Appendix E.

| Recognition Network | Generative Network |
| --- | --- |
| $b_1^{enc}$: $\dim(\mathbf{x}) \times 500$ linear layer | $r_2^{enc}$: $\dim(\mathbf{z}_2) \times 2000$ linear layer |
| ReLU activation | ReLU activation |
| $r_1^{enc}$: $500 \times (2 \cdot \dim(\mathbf{z}_1))$ linear layer | $b_2^{enc}$: $2000 \times 500$ linear layer |
| | ReLU activation |
| $b_2^{enc}$: $500 \times 2000$ linear layer | $r_1^{enc}$: $\dim(\mathbf{z}_1) \times 500$ linear layer |
| ReLU activation | ReLU activation |
| $r_2^{enc}$: $2000 \times (2 \cdot \dim(\mathbf{z}_2))$ linear layer | $b_1^{enc}$: $500 \times \dim(\mathbf{x})$ linear layer |
| | Sigmoid activation |

Table 5: Details of the convolutional ladder architecture trained on 3DShapes and SVHN. Conv2d is the 2D convolutional operation, and ConvTranspose2d is the 2D transposed convolutional operation. We implement both operations using the `torch.nn` package in PyTorch. For both operations, the four numbers represent output channels, input channels, kernel size (height) and kernel size (width) respectively $(C_{out}, C_{in}, H, W)$. Convolutional operations marked with (*) have stride 1 to ensure valid dimensions. All remaining convolutional operations have stride 2. For experimental results of models from this architecture, see Section 4 and Appendix E.

| Recognition Network | Generative Network |
| --- | --- |
| $b_1^{enc}$: $64 \times \dim(\mathbf{x}) \times 4 \times 4$ Conv2d | $r_2^{enc}$: $\dim(\mathbf{z}_2) \times 16384$ linear layer |
| ELU activation, batch norm | ELU activation, batch norm |
| $r_1^{enc}$: $64 \times 64 \times 4 \times 4$ Conv2d | $b_2^{enc}$: $128 \times 256 \times 4 \times 4$ ConvTranspose2d |
| ELU activation, batch norm | ELU activation, batch norm |
| $64 \times 64 \times 4 \times 4$ Conv2d (*) | $64 \times 128 \times 4 \times 4$ ConvTranspose2d (*) |
| ELU activation, batch norm | ELU activation, batch norm |
| $1024 \times (2 \cdot \dim(\mathbf{z}_1))$ linear layer | $r_1^{enc}$: $\dim(\mathbf{z}_1) \times 16384$ linear layer |
| $b_2^{enc}$: $128 \times 64 \times 4 \times 4$ Conv2d | ELU activation, batch norm |
| ELU activation, batch norm | $b_1^{enc}$: $\dim(\mathbf{x}) \times 128 \times 4 \times 4$ ConvTranspose2d |
| $r_2^{enc}$: $128 \times 128 \times 4 \times 4$ Conv2d | ELU activation, batch norm |
| ELU activation, batch norm | Sigmoid activation (only for SVHN) |
| $256 \times 128 \times 4 \times 4$ Conv2d | |
| ELU activation, batch norm | |
| $3136 \times (2 \cdot \dim(\mathbf{z}_2))$ linear layer | |

Lastly, in Table 6, we provide details on the shared architecture for MNIST training, with its results presented in Appendix E.1.

Table 6: Details of the shared architecture trained on MNIST. For its results, see Appendix E.1.

| Recognition Network | Generative Network |
| --- | --- |
| $s^{enc}$: $\dim(\mathbf{x}) \times 500$ linear layer | $t_1^{dec}$: $\dim(\mathbf{z}_1) \times 2000$ linear layer |
| ReLU activation | ReLU activation |
| $500 \times 2000$ linear layer | $t_2^{dec}$: $\dim(\mathbf{z}_2) \times 2000$ linear layer |
| ReLU activation | ReLU activation |
| $t_1^{enc}$: $2000 \times (2 \cdot \dim(\mathbf{z}_1))$ linear layer | $s^{dec}$: $2000 \times 500$ linear layer |
| | ReLU activation |
| $t_2^{enc}$: $2000 \times (2 \cdot \dim(\mathbf{z}_2))$ linear layer | $500 \times \dim(\mathbf{x})$ linear layer |
| | Sigmoid activation |

**Initialisation.** In MFCVAE, all parameters in the deep neural networks $g(\mathbf{x}; \phi)$ and $f(\vec{\mathbf{z}}; \theta)$ are initialised using either Glorot normal initialisation [60] for MNIST, and using a data-dependent

initialisation method [61] for 3DShapes and SVHN. The idea of the data-dependent initialisation is to set the parameters in the deep neural network such that all layers in the network are encouraged to train at roughly the same rate, with the aim of avoiding vanishing or exploding gradients. Data-dependent initialisation is particularly useful for convolutional neural networks. Therefore, we use it as a starting point for model training of 3DShapes and SVHN datasets, where convolutional neural networks are used.

For the parameters of the MoGs, we initialise them facet-wise as follows:

- Mixing weights $\boldsymbol{\pi}_j$ are initialised to be $1/K_j$ component-wise.
- For each $k_j \in \{1, ..., K_j\}$, means $\boldsymbol{\mu}_{j,k_j}$ of the Gaussians are initialised with the means on an MoG (implemented with the package `sklearn.mixture.GaussianMixture`) fitted on a dataset consisting of latent observations $\mathbf{z}_j$ sampled from $q(\mathbf{z}_j|\mathbf{x})$, where $\mathbf{x}$ are all batches from the corresponding training dataset of MFCVAE. Note that encoder parameterising $q(\mathbf{z}_j|\mathbf{x})$ is not trained at this point. The aim is to encourage a smoother and faster learning of the multiple MoG prior by starting from an MoG fitted to the initial state instead of one initialised at random.
- Covariance matrices $\Sigma_{j,k_j}$ are not initialised by the outputs from the fitted MoG above. Instead, a fixed value is assigned to all diagonal entries of the covariance matrices. The fixed value is the same across all clusters in all facets, which is a hyperparameter set to be much larger than the output variances from the trained fitted MoGs. This initialisation is favoured because at the start of the training, the MoGs do not contain much useful information as they were fitted on latent observations obtained from a randomly initialised model. An overly small variance at the start of the training could result in the model being stuck in a local optimum prematurely.

We note that we found these prior initialisations to be reasonable choices, but have not extensively explored alternatives.

**Potential negative societal impacts.** Our work is mainly of theoretical and methodological nature, thus, we do not have a direct application of our model on which it could cause immediate negative societal impacts. Since we provide a general clustering algorithm, MFCVAE can be used in malicious or potentially unethical ways for any clustering task at hand, suited particularly for high-dimensional data. Our model does not account for fairness of clusters, which should be taken particular care of when dealing with data from human subjects. As our model is a generative model by nature, we mention the possibility to abuse our model for the generation of deepfakes for disinformation. Further, we have not investigated the vulnerability of our model to aversarial attacks, which might cause a significant security problem when applied in front-end applications and tasks.

**Compute resources.** We had access to two GPU clusters: One internal cluster with 12 Nvidia GeForce GTX 1080 graphic cards each with 8GB of memory that was shared with many other users (access for 5 months ongoing), and one Microsoft Azure cluster with initially two, later four Nvidia Tesla M60 each with 8GB of memory that was used only by the authors (access for approximately 4 months).

To train one model on each of the 4 dataset (configurations) on the Azure cluster detailed above, it takes approximately 31 min for MNIST, 36 min for 3DShapes (in both configurations), and 5h 54 min for SVHN. Since we performed a seed sweep over 10 runs on each of the 4 dataset (configurations), the total computational time to reproduce the main results in this paper is (31 min + 2 · 36 min + 354 min) · 10 = 4570 min ≈ 76 hours of GPU time.

### D.5  Differences between VaDE and ($J = 1$) MFCVAE

In the following, we describe the (pre-)training algorithm of VaDE [10] and compare it with the training of MFCVAE with one facet ($J = 1$). Throughout, VaDE uses a symmetric, shared encoder and decoder architecture (see Appendix D.2). VaDE has the following two differences compared to MFCVAE ($J = 1$):

- *Stacked Denoising Autoencoder (SAE)* [62] pretraining: VaDE uses a two-stage SAE deterministic pretraining algorithm which is in detail described in [8] to find good initialisations for the parameters $\theta$ of the decoder and $\phi$ of the encoder. During the first stage, denoising autoencoders, which are two-layer neural networks of symmetric shapes, are trained using a least-squares reconstruction loss (i.e. deterministically as a plain autoencoder). In every iteration of this first stage of the pretraining routine, the outermost layers (at the front of the encoder and the back of

the decoder), which have been trained in previous iterations, are frozen, and the next denoising autoencoder towards the centre of the architecture is trained. Then, in the second stage of training, the entire architecture, which has been trained in this sequential fashion, is fine-tuned, again using a deterministic reconstruction loss. For a detailed description of this pretraining algorithm, we refer to [8].

Once the pretraining routine is complete, VaDE uses the weights $\theta$ and $\phi$ obtained as the initialisation of regular VaDE training, maximizing the ELBO with Monte Carlo sampling.— We note that MFCVAE does not require any SAE pretraining. Instead, we either initialise these weights randomly (MNIST) or using data-dependent initialization (3DShapes and SVHN; details see Appendix D.4).

- VaDE restricts the covariance matrices $\Sigma_c$ of the conditional Gaussian distributions $p(\mathbf{z} \mid c) = \mathcal{N}(\mu_c, \Sigma_c)$ to be *diagonal*, i.e. each $p(\mathbf{z} \mid c)$ is a product of $\dim(\mathbf{z})$ independent univariate Gaussian distributions. In contrast, MFCVAE allows $\Sigma_c$ to be *full* and enables MFCVAE to express more complex (facet-wise) dependencies in the prior.

The SAE pretraining routine adds significant complexity to the training algorithm which MFCVAE does not require in order to obtain comparable performance. Once the encoder and decoder are initialised, both VaDE and MFCVAE use a Gaussian-Mixture model to initialise the prior $p(\mathbf{z})$ and its parameters $\pi, \mu_c$ and $\Sigma_c$, fitted with an EM-algorithm. We note that in the single-facet case, training MFCVAE simplifies to only one progressive step, i.e. the main training stage of VaDE is equivalent to that of MFCVAE (but with different initialisation).

# E  Additional experimental results

## E.1  On the stability of training

In this appendix, we analyse the stability of MFCVAE with respect to different neural architectures and discuss the stability of deep clustering models in this context.

A natural starting point for a neural architecture of MFCVAE is a shared encoder and decoder architecture (as detailed in Appendix D.2), which was previously used in VaDE [10] and other deep clustering models. When using this architecture, we observe a high variation between runs which only vary in their (partly) random initialisation (determined by the random seed; see Fig. 10 [Top left]). However, when being lucky, drawing the right lottery ticket, this architecture can yield excellent disentanglement of facets, just like our progressively trained VLAE architecture can do (but in a stable manner). We visualise input examples assigned to clusters from such a lucky run (which is not part of Fig. 10 [Top left]) in Fig. 11. We point out that this run is cherry-picked from over 100 runs with different random initialisation. We could not produce stable results with a shared encoder and decoder architecture, neither for $J = 1$ nor $J > 1$. Given these stability issues of deep clustering models, it is not only crucial to address them (which we do next), but this even more highlights the importance of providing error bars, and that picking the best run of many (as has been common practice among several deep clustering papers) is particularly here not acceptable.

To overcome these stability issues, we used a combination of a progressive training algorithm and a VLAE architecture. We found that only using a VLAE architecture (Fig. 10 [Top right]) significantly improves the performance of disentangling facets (here only measured in terms of accuracy w.r.t. the supervised label), but is not sufficient to fully stabilise the runs over different random seeds. Only by additionally using a progressive training schedule (Fig. 10 [Bottom]), we achieve very good disentanglement of facets and at the same time stable performance. While we here report this for one configuration of hyperparameters only and on MNIST, we made this observation throughout all datasets and in diverse hyperparameter settings.

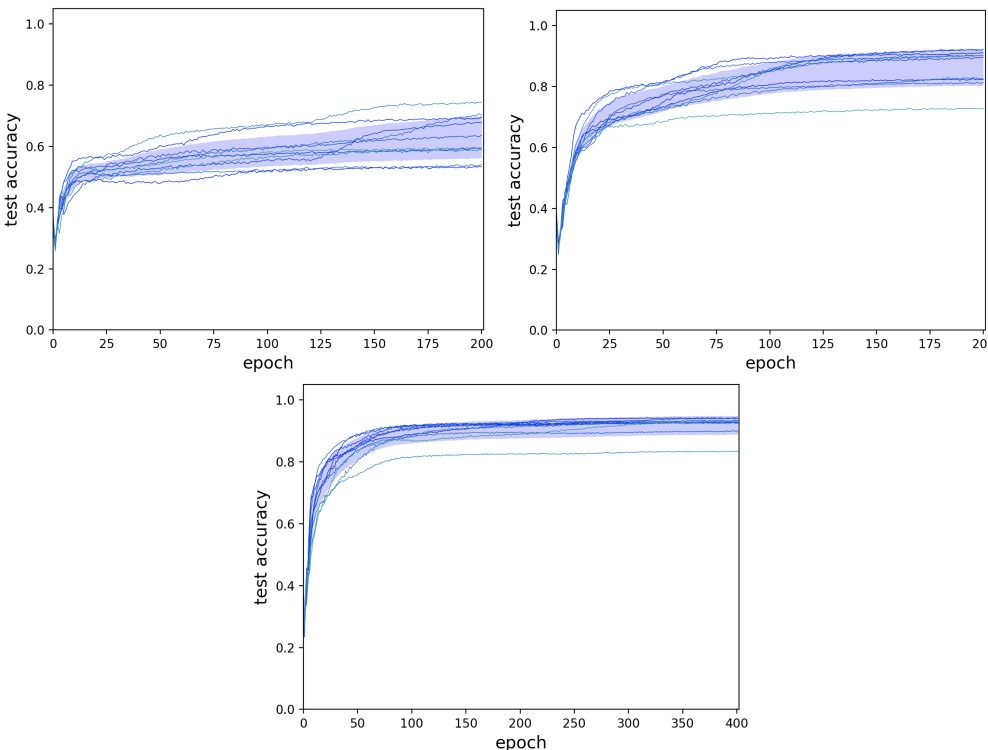

Figure 10: Test accuracy over training epochs for models trained on MNIST for three different architectures. Ten runs are performed for each architecture. Each run is illustrated by one curve. The blue shade is bounded by the mean accuracy plus and minus one standard deviation across the ten runs. [Top left] Shared architecture [Top right] VLAE architecture *without* progressive training schedule [Bottom] VLAE architecture *with* progressive training schedule

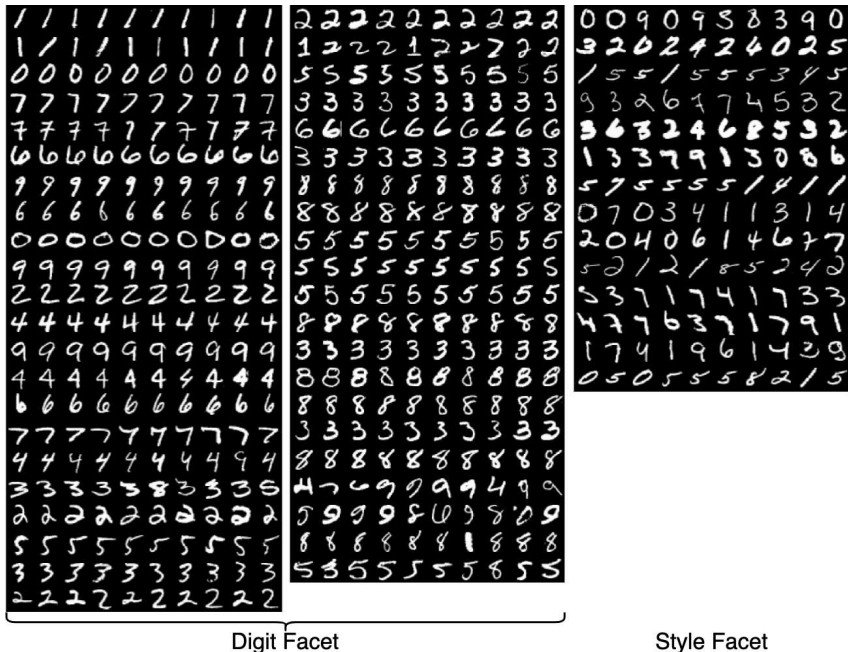

Digit Facet                    Style Facet

Figure 11: Input examples of a cherry-picked MFCVAE model with a shared architecture, with a lucky lottery ticket drawn as the random initialisation, and two-facets ($J = 2$), trained on MNIST. Sorting is performed in the same way as in Fig. 4.

### E.2 Generalisation between training and test set

An important question in an exploratory setting is to what degree clustering results generalise from a training to a test set. In supervised machine learning, it is common to see a generalisation gap: Performance of a model is generally better on the training set than on the test set, often because the model overfits on the training set, and the goal is to minimise this gap, while actually being interested in test set performance. Perhaps surprisingly, we observe that MFCVAE has a negligible generalisation gap, i.e. typically performs almost equally well on the training and test set.

To analyse this, we use the exact experimental setup of our main results as detailed in Appendix D.4 with $J = 2$ facets, training on MNIST. Fig. 12 shows the unsupervised clustering accuracy over training epochs, evaluated both on the training set (left) and the test set (right). When evaluating unsupervised clustering accuracy after the model is fully trained, a mean training accuracy of $91.85\% \pm 3.09\%$ is achieved across ten runs, which is slightly lower than the mean test accuracy of $92.02\% \pm 3.02\%$ presented in Section 4.3. Thus, while we observe small differences between performance on training and test set, also when considering individual runs, these are not significant. In summary, MFCVAE generalises well between training and test set.

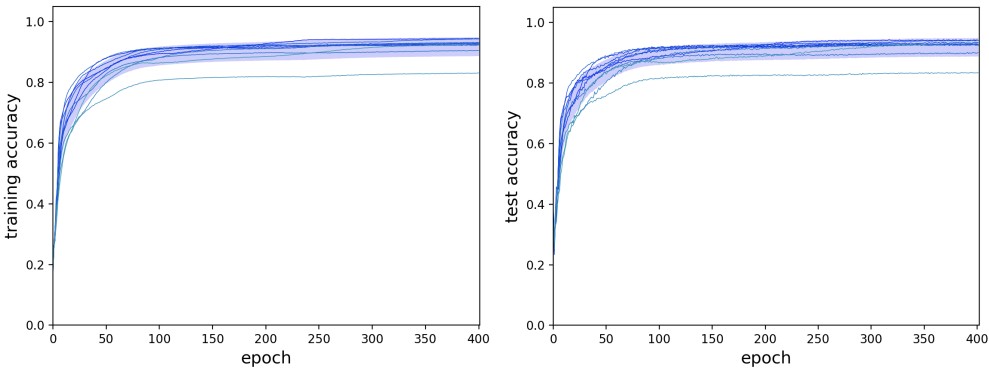

Figure 12: Unsupervised clustering accuracy over training epochs for MFCVAE trained on (the training set of) MNIST as detailed in Appendix D.4, and evaluated on the training set [Left] and the test set [Right], respectively. Ten runs are performed, with each run being illustrated by one curve on the left and right, respectively. The blue shade is bounded by the mean accuracy plus and minus one standard deviation across the ten runs.

### E.3 Discovering a multi-facet structure

This appendix provides the complete results of Section 4.1. As before, we visualise input examples from the test set for clusters of MFCVAE with two-facets ($J = 2$) trained on MNIST, 3DShapes and SVHN. Here, we show all clusters of our results in Fig. 4, visualised in Figs. 13 to 16. In all figures, inputs (columns) are sorted in decreasing order by their assignment probability $\max_{c_j} \boldsymbol{\pi}_j(c_j | q_\phi(\mathbf{z}_j | \mathbf{x}))$.

In particular, in Fig. 13, we directly compare our model to the results shown in LTVAE [27], the model closest to ours in its attempt to learn a clustered latent space with multiple disentangled facets. As can be seen, LTVAE struggles to separate data characteristics into separate facets (see Fig. 13 (b)). In particular, LTVAE learns digit class in both facets, i.e. this characteristic is not properly disentangled between facets. In comparison, MFCVAE better isolates the two characteristics, and does not learn digit class in the style facet.

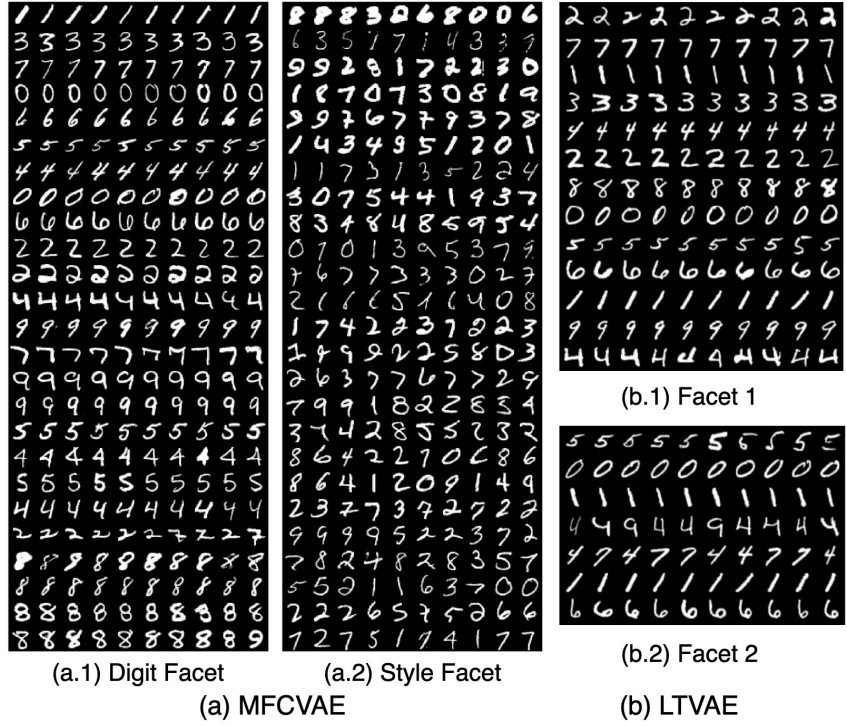

(a.1) Digit Facet  (a.2) Style Facet  (b.1) Facet 1

(a) MFCVAE     (b.2) Facet 2

           (b) LTVAE

Figure 13: (a) Input examples for clusters of MFCVAE with two-facets ($J = 2$) trained on MNIST. Rows and columns are sorted as in Fig. 4. (b) Input examples for clusters of LTVAE with two-facets, likewise trained on MNIST. Plot is taken as reported in [27], Fig. 5.

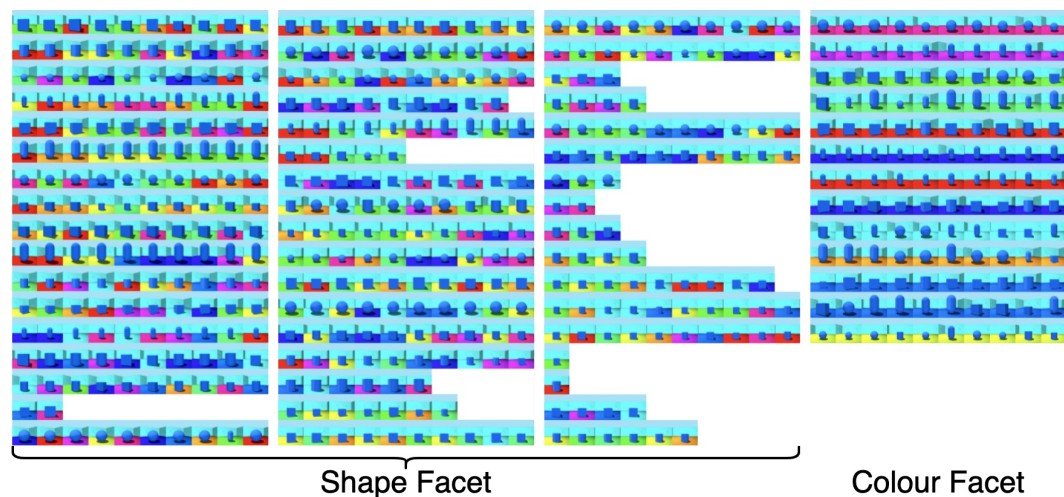

Shape Facet        Colour Facet

Figure 14: Input examples for clusters of MFCVAE with two-facets ($J = 2$) trained on 3DShapes (configuration 1). Rows and columns are sorted as in Fig. 4.

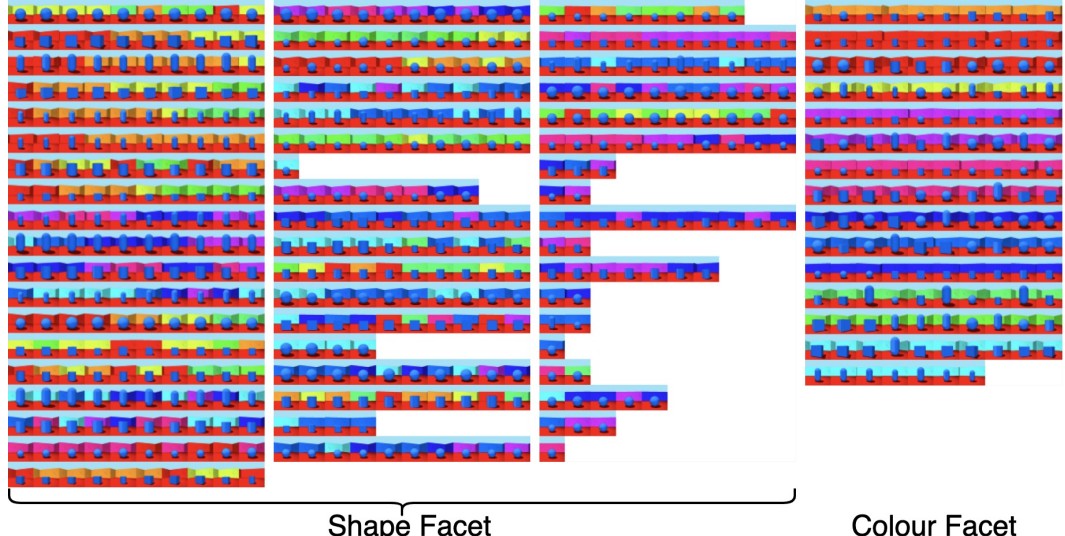

Shape Facet          Colour Facet

Figure 15: Input examples for clusters of MFCVAE with two-facets ($J = 2$) trained on 3DShapes (configuration 2). Rows and columns are sorted as in Fig. 4.

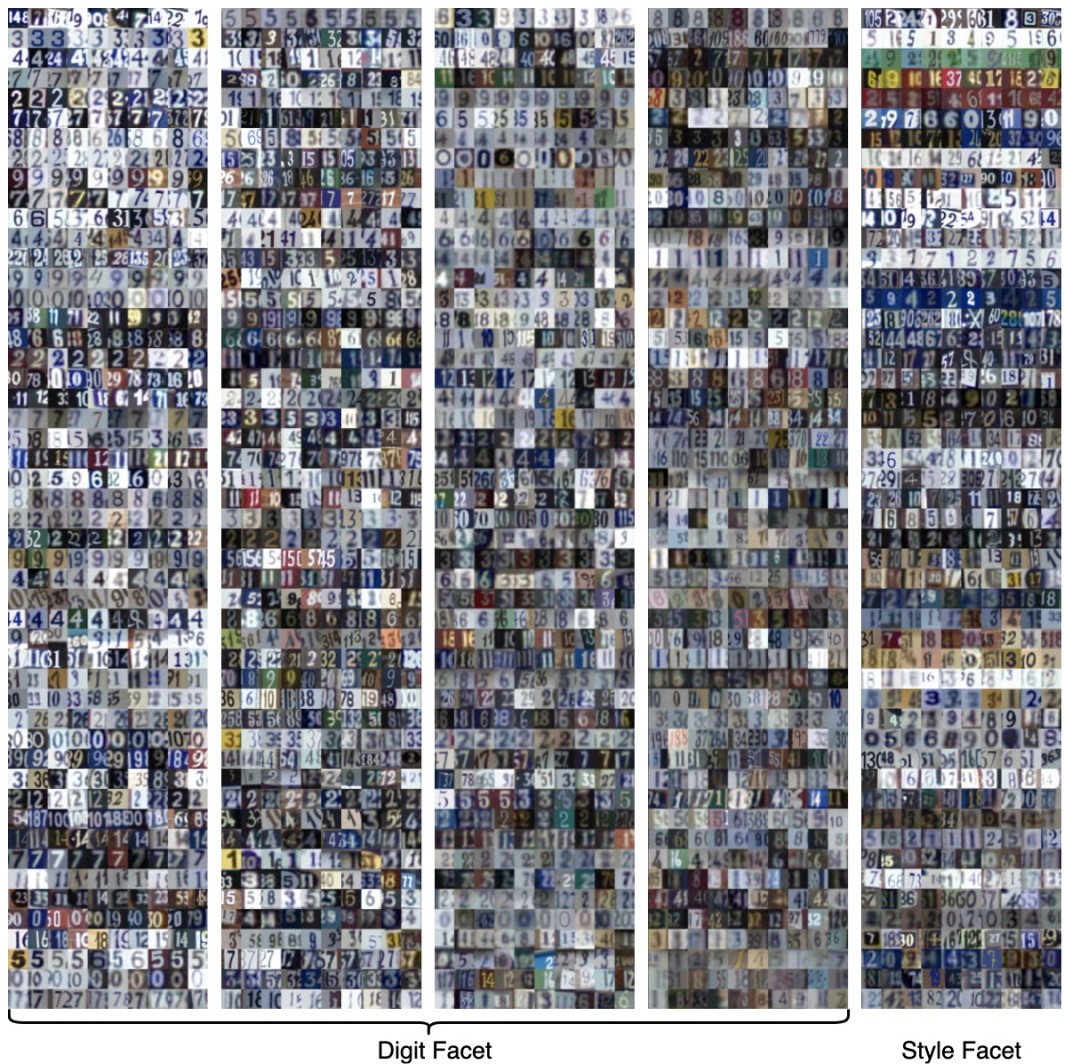

Digit Facet                                    Style Facet

Figure 16: Input examples for clusters of MFCVAE with two-facets ($J = 2$) trained on SVHN. Rows and columns are sorted as in Fig. 4.

### E.4 Compositionality of facets

In this appendix, we provide further combinations of clusters of which the style facet is swapped and give a more rigorous explanation of the swapping procedure applied.

Let us have two input examples $\mathbf{x}^{(1)}$ and $\mathbf{x}^{(2)}$ assigned to two different style clusters according to Eq. (12) (and typically two different digit clusters), i.e. $c_j^{(i)} = \mathrm{argmax}_{c_j}\,\boldsymbol{\pi}_j(c_j|q_\phi(\mathbf{z}_j|\mathbf{x}^{(i)}))$ and $c_j^{(1)} \neq c_j^{(2)}$, where $j = 1$ for MNIST and $j = 2$ for 3DShapes and SVHN. We can obtain the latent representations $\tilde{\mathbf{z}}_j^{(i)}$ of input examples for both facets, taking the mode of $q_\phi(\mathbf{z}_j|\mathbf{x}^{(i)})$, respectively, which is parameterised via a forward pass. Now, we swap the style/colour facet's latent representation ($\tilde{\mathbf{z}}_1^{(i)}$ for MNIST, and $\tilde{\mathbf{z}}_2^{(i)}$ for 3DShapes and SVHN) between the two inputs, while fixing the digit/shape facet's latent representation ($\tilde{\mathbf{z}}_2^{(i)}$ for MNIST, and $\tilde{\mathbf{z}}_1^{(i)}$ for 3DShapes and SVHN). Once the swapping is complete, we pass these latent representations through the decoder of our model to obtain reconstructions of our model from "swapped style" latent representations. Note that this swapping operation is symmetric in the two-facet case, in the sense that the resulting two reconstructions are the same regardless of whether we swap $\tilde{\mathbf{z}}_1$ or $\tilde{\mathbf{z}}_2$. Formally, we obtain reconstructions $\hat{\mathbf{x}}^{(1)} = f(\{\tilde{\mathbf{z}}_1^{(1)}, \tilde{\mathbf{z}}_2^{(2)}\}; \theta)$ and $\hat{\mathbf{x}}^{(2)} = f(\{\tilde{\mathbf{z}}_1^{(2)}, \tilde{\mathbf{z}}_2^{(1)}\}; \theta)$. It is such 4-tuples of inputs $\mathbf{x}^{(1)}$ and $\mathbf{x}^{(2)}$ and reconstructions $\hat{\mathbf{x}}^{(1)}$ and $\hat{\mathbf{x}}^{(2)}$ that we visualise.

In Fig. 17, we visualise further 4-tuples from all datasets (and configurations), in addition to the results presented in Section 4.2. For each dataset, the first 4-tuple consists of the first element of both input rows (left and right) and both reconstruction rows (left and right). Note that we limit ourselves here to few digit/shape and colour/style cluster combinations. However, we note that for MNIST and 3DShapes, many other combinations could be found where a similar "style swapping effect" can be observed. For the much richer dataset SVHN on which our model is less well fitted, while we can find cluster combinations where compositionality of facets works somewhat well (see Fig. 17 (d)), we can also find failure cases (see e.g. Fig. 17 (e)). Here, the style, represented as a white background, shall be composed with a white digit. We hypothesise that as this combination is particularly rare in the real world, the model struggles to reconstruct these latent combinations and as a consequence introduces undesired artefacts into the reconstructions.

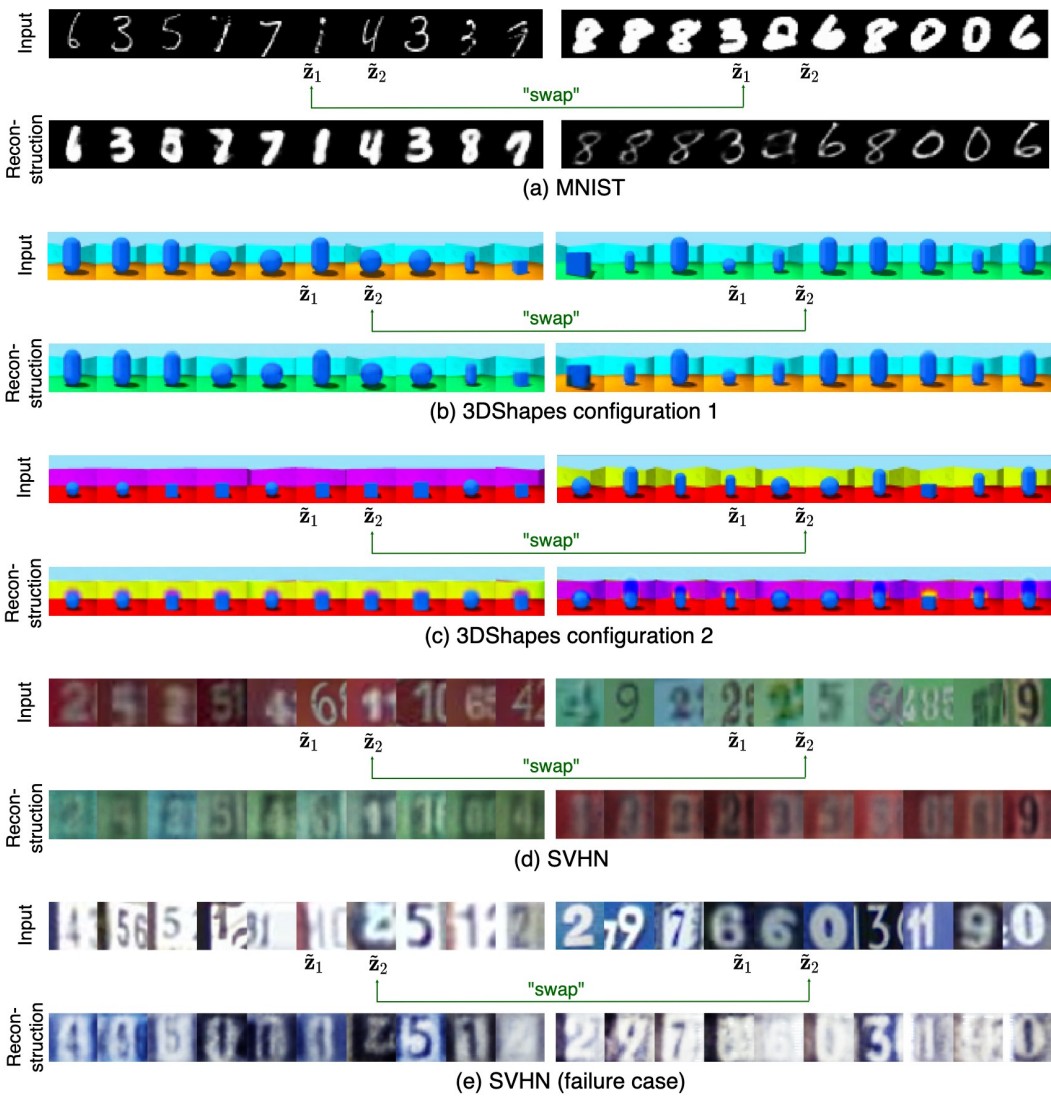

Figure 17: Input examples and reconstructions when swapping their style/colour facet's latent representation.

## E.5 Generative, unsupervised classification

To compare against assumed ground-truth clusterings imposed by the supervised class structure in our datasets, we report generative classification performance in terms of *unsupervised clustering accuracy* on the test set. When the number of clusters in a facet is equal to the number of ground-truth classes compared against, one can use the Hungarian algorithm to find the optimal 1-to-1 mapping between clusters and classes [10, 63]. When the number of clusters in a facet is greater than the number of ground-truth classes compared against, as is common, one can simply assign each cluster in a facet to the most frequent ground-truth class found within that cluster [28].

In Figs. 18 to 20, we plot generative classification performance on the test set measured in terms of unsupervised clustering accuracy over the training epochs. The blue shade is bounded by the mean accuracy plus and minus one standard deviation over the ten runs. The sudden jumps of accuracy after $\approx 100$ epochs are caused by the progressive training algorithm which loops in a new facet at that point.

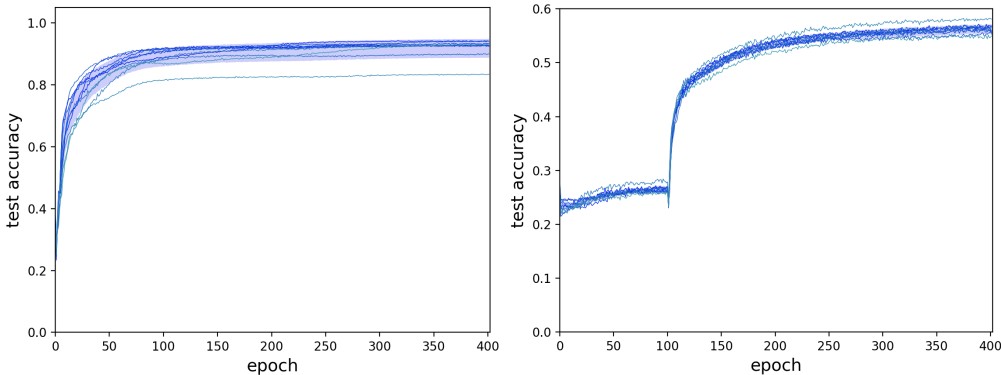

Figure 18: Unsupervised clustering accuracy on the test set w.r.t. the supervised label, when trained on [Left] MNIST, and [Right] SVHN.

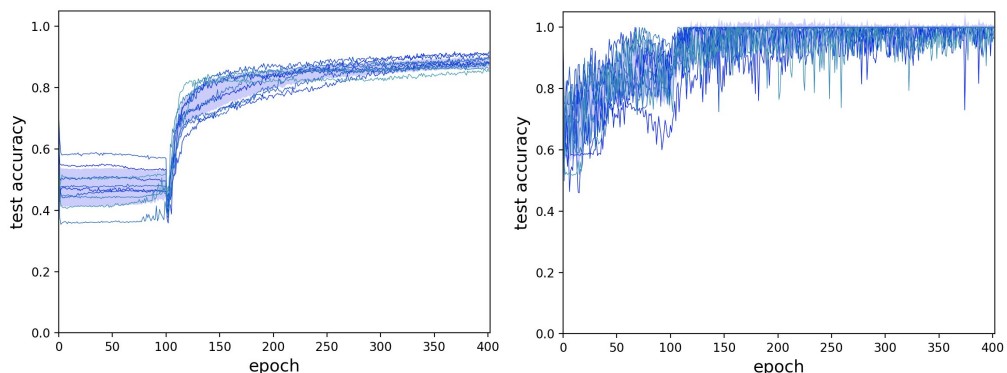

Figure 19: Unsupervised clustering accuracy on the test set for [Left] object shape and [Right] floor colour, when trained on 3DShapes (config. 1).

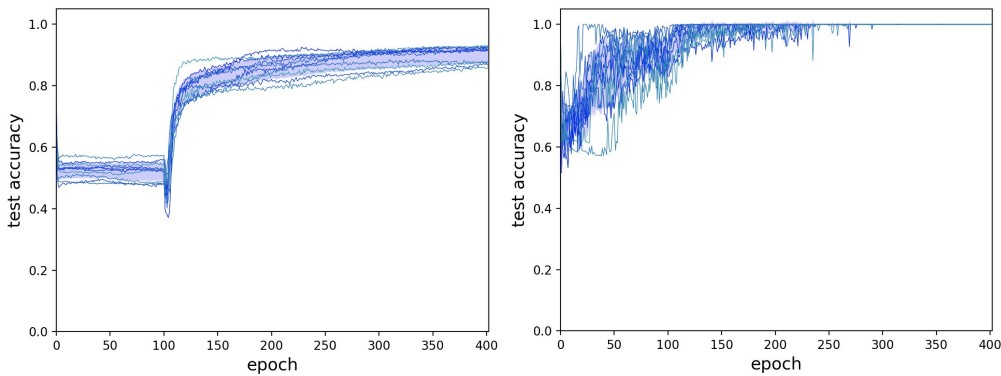

Figure 20: Unsupervised clustering accuracy on the test set for [Left] object shape and [Right] wall colour, when trained on 3DShapes (config. 2).

## E.6 Diversity of generated samples

First, we compare sample generation performance on MNIST between MFCVAE (a) and LTVAE (b) in Fig. 21. We sample MFCVAE as discussed in Section 4.4. For LTVAE, while not fully clear, LTVAE samples from one Mixture-of-Gaussian, i.e. first samples from a categorical, then from the chosen component, to obtain $\mathbf{z}$ ([27], Section 4.5). This sampling procedure resembles MFCVAE's reconstructions that illustrate its digit facet (e.g. Fig. 21 (a.1)). When comparing (a.1) and (b), it can be observed that generation performance is comparable between the two models, with each row representing a certain digit identity cluster. However, as shown in (a.2) and not demonstrated by LTVAE, our model additionally allows sample generation conditional on style clusters. This demonstrates the advantage of MFCVAE in its intervention capability for each facet during sample generation, allowing a rich set of options for potential downstream tasks.

To quantitatively compare the sampling diversity of MFCVAE against VaDE, we compute the Learnt Perceptual Image Patch Similarity (LPIPS) [64] of 60,000 samples generated from models trained on MNIST and SVHN. We also report LPIPS computed on on real (i.e. non-synthetic) images, where we use the data from both training set and test set. For LPIPS, higher is better, while it must be ensured that the true data distribution is modelled (and not just a distribution that artificially maximises the metric). As shown in Table 7, our method generates samples with similar diversity to VaDE on both datasets. Further, MFCVAE is close to the "real image diversity" for MNIST, yet is somewhat lower for SVHN.

Table 7: LPIPS of real images and 60,000 samples generated from VaDE and MFCVAE for MNIST and SVHN.

|  | MNIST | SVHN |
|---|---|---|
| Real Images | 0.112 | 0.227 |
| VaDE | 0.111 | 0.187 |
| MFCVAE (ours) | 0.116 | 0.182 |

In addition, Figs. 22 to 24 show the complete plots of synthetic samples generated from MFCVAE where all clusters are visualised compared to the main text. Again, we refer to Section 4.4 for an explanation on the procedure of how these samples are generated, but provide here additional details: During sample generation, the variance of the distributions over latent variables $\mathbf{z}_1$ and $\mathbf{z}_2$ are scaled by a "temperature" factor $\tau > 0$. This is a common technique in likelihood-based deep generative models to improve the quality of generated samples [65, 66]. To formalise this, at sampling time, the covariance matrix $\tau\Sigma_{c_j}$ is used for $p(\mathbf{z}_j|c_j)$, instead of $\Sigma_{c_j}$. In this set of experiments, temperature scaling is used for 3DShapes and SVHN, where we choose $\tau = 0.3$. For MNIST, we do not use temperature scaling, i.e. $\tau = 1.0$.

For MNIST and 3DShapes, it can be observed that for clusters with a lower average assignment probability, synthetic samples remain homogeneous w.r.t. their characteristic value in each facet. For SVHN, the sample reconstruction quality drops for clusters with lower average assignment probabilities which we can attribute to a smaller separation of facets on this dataset as observed in Section 4.3.

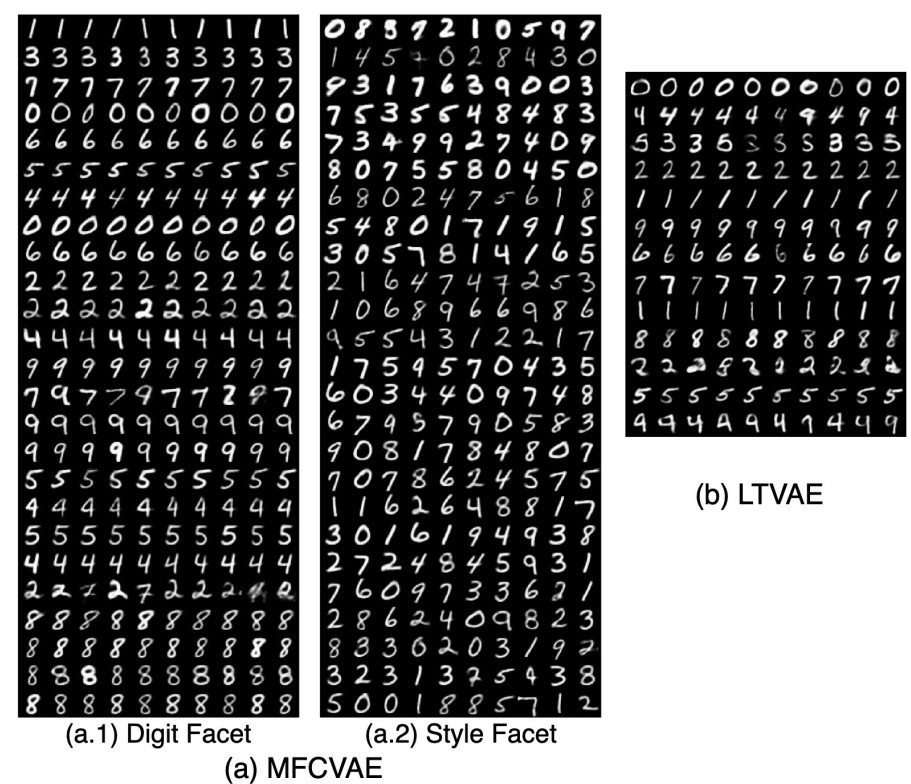

(a.1) Digit Facet     (a.2) Style Facet

(a) MFCVAE

(b) LTVAE

Figure 21: (a) Synthetic samples generated from MFCVAE with two facets ($J = 2$) trained on MNIST, with all clusters visualised. Rows are sorted as in Fig. 6. (b) Synthetic samples generated from LTVAE trained on MNIST. Plot is taken as reported in [27], Fig. 7.

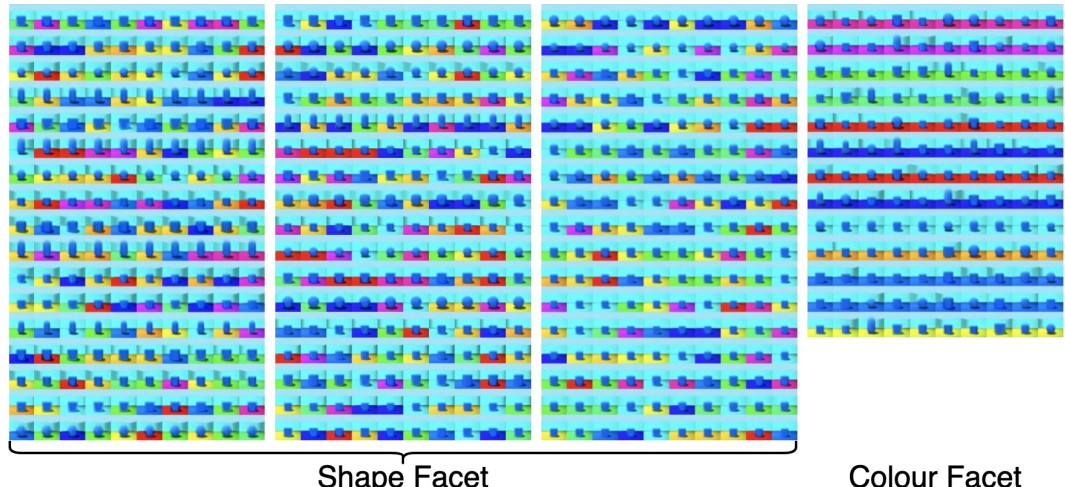

Shape Facet         Colour Facet

Figure 22: Synthetic samples generated from MFCVAE with two facets ($J = 2$) trained on 3DShapes (configuration 1), with all clusters visualised. Rows are sorted as in Fig. 6.

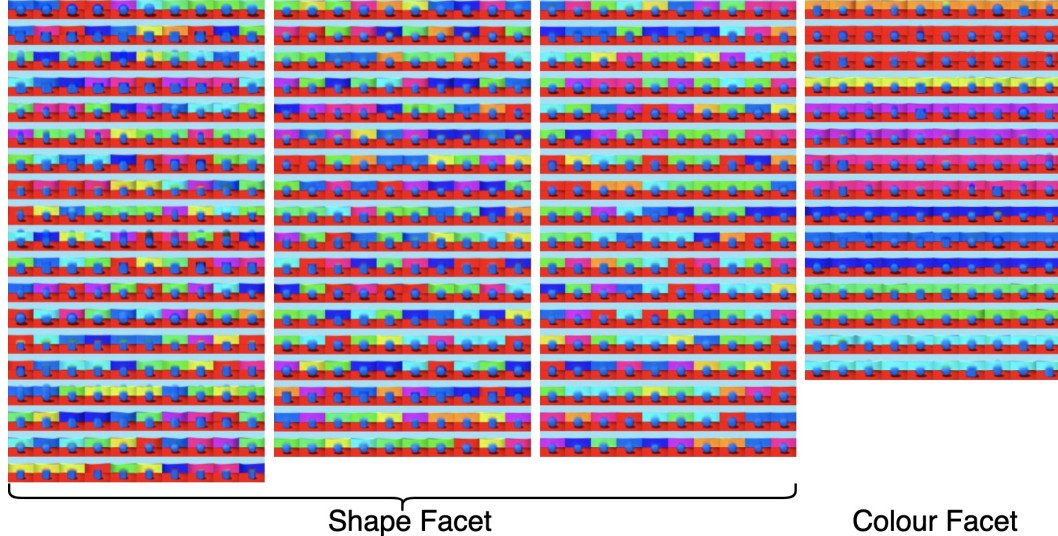

Shape Facet         Colour Facet

Figure 23: Synthetic samples generated from MFCVAE with two facets ($J = 2$) trained on 3DShapes (configuration 2), with all clusters visualised. Rows are sorted as in Fig. 6.

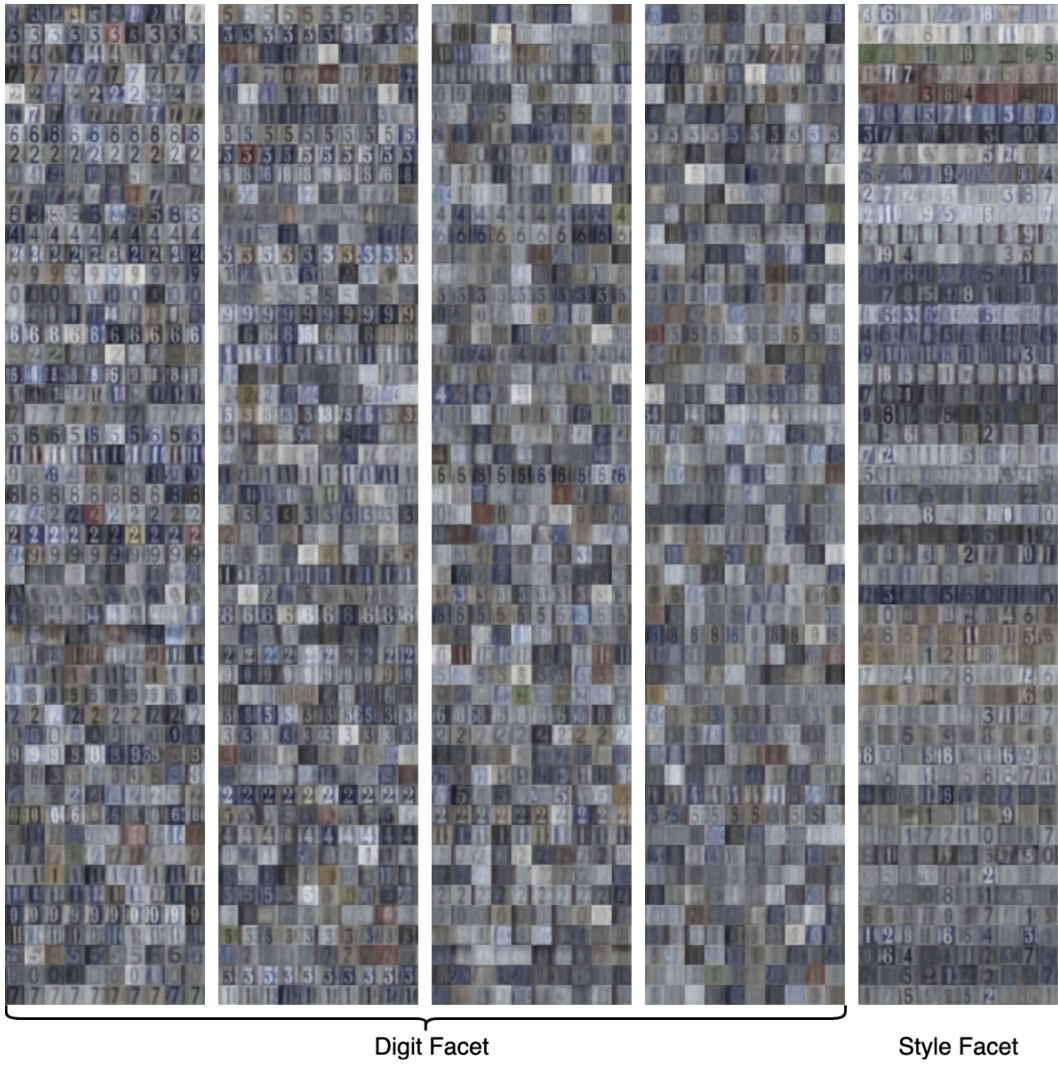

Digit Facet                                    Style Facet

Figure 24: Synthetic samples generated from MFCVAE with two facets ($J = 2$) trained on SVHN, with all clusters visualised. Rows are sorted as in Fig. 6.