# OpenReview forum: "Multi-Facet Clustering Variational Autoencoders"
_NeurIPS.cc/2021/Conference — NeurIPS 2021 Poster_

### Official Review · Reviewer_NacR · 2021-07-16

**Rating:** 6
**Confidence:** 3

**Summary:**

There are multiple interesting aspects of high-dimensional data.
This paper proposes a Multi-Facet Clustering Variational Autoencoders (MFCVAE) to cluster the data in multi-aspects.
MFCVAE utilizes and extends the trick of VaDE, and provides the methods to learn multi-aspect clustering in an end-to-end fashion.
This paper provides various qualitative results to validate the multi-aspect clustering performance.

**Ethical Concerns:**

There are no ethical concerns.

**Limitations And Societal Impact:**

The authors adequately addressed the limitations of their work.

**Main Review:**

< originality >
1. MFCVAE utilizes a mixture of Gaussian prior for multiple clustering simultaneously, and the task is important and novel.
2. For simultaneous clustering, MFCVAE utilizes and extends the VaDE tricks. The extension is somewhat novel.
3. MFCVAE is related to disentangled representation, but this paper does not discuss the relationship between MFCVAE and various disentanglement models, [1,2,3]. Discussion on the relationship between MFCVAE and various disentanglement methods will be interesting

< quality >
1. The technical part is sound
2. The paper does not provide the quantitative methods to measure the disentanglement and sample diversity. The lack of quantitative measure makes the reader not ensure the effectiveness of MFCVAE

< clarity >
1. The paper is well-written.
2. It is hard to clarify the quantitative experimental results. There is no quantitative results table.
3. There exist only J=2 experiments, and additional experimental results for J>2 will be more interesting.

< significance >
1. The experimental results of MFCVAE are not significant. On MNIST, MFCVAE shows 92.02%, but IMSAT achieves 98.4%.
2. There is no quantitative diversity metric, and it is hard to ensure that the MFCVAE generates better diverse samples.

[1] Kim, Hyunjik, and Andriy Mnih. "Disentangling by factorising." International Conference on Machine Learning. PMLR, 2018.
[2] Mathieu, Emile, et al. "Disentangling disentanglement in variational autoencoders." International Conference on Machine Learning. PMLR, 2019.
[3] Sha, Lei, and Thomas Lukasiewicz. "Multi-type Disentanglement without Adversarial Training." Proceedings of the 35th AAAI Conference on Artificial Intelligence. 2021.

**Time Spent Reviewing:**

6

---

> ### Author Response · Authors · 2021-08-10
> **Response to Reviewer NacR (Part 1 of 2)**
>
> Thank you so much for dedicating your valuable time to reviewing our research. Your astute comments have inspired us to add extra quantitative experimental results on accuracy, disentanglement and sample diversity, as well as extra discussion around the novel contributions of MFCVAE - we hope you like these enhancements!
>
> ---
>
> > “2. MFCVAE is related to disentangled representation, but this paper does not discuss the relationship between MFCVAE and various disentanglement models, [1,2,3]. Discussion on the relationship between MFCVAE and various disentanglement methods will be interesting”
>
> Thank you for this great suggestion which we have followed up on by adding text in the Related Work section, summarizing the similarities and differences between MFCVAE and disentangling methods. We give a flavour of our discussion here:
> Disentangling methods, e.g. in [12-14], provide models and adapt the loss function (often with a regularisation term) to learn a disentangled representation in a latent space. This gives disentangling methods a high degree of flexibility, as they can in principle be applied to a wide variety of models and datasets. MFCVAE is similar in this regard: it also seeks separation in its latent space, representing different data aspects/characteristics in the various facets, with each facet tractably (via VaDE tricks) incorporating not only continuous but also discrete latent variables in an MoG model.
>
> Turning to a few more ways in which MFCVAE differs from disentanglement methods:
>
> * MFCVAE has a clustered latent space via the multiple Mixture-of-Gaussians (MoG) prior, which allows to compute the “clustering probability” $\pi_j( c_j | q_\phi(\mathbf{z_j} | \mathbf{x})) := \frac{\exp(\mathbb{E}_{q_\phi(\mathbf{z_j} | \mathbf{x})} \log p_\theta(c_j|  \mathbf{z_j}))}{Z_j(q_\phi(\mathbf{z_j} | \mathbf{x}))}  \text{, for}\ c_j =1, \ldots, K_j$ for every facet j (compare Eq. (12)). In particular, in addition to disentangling data characteristic in the latent space, a clustered latent space makes clustered inputs within one cluster homogenous w.r.t. a specific characteristic value, and heterogeneous between clusters (compare Fig. 3). [12-14] do not possess this property.
> * Furthermore, we think it is somewhat surprising that by just choosing a multiple MoG prior (and the appropriate architecture and training algorithm), one can achieve a *clustered*, “disentangled” representation of the latent space learned in a fully unsupervised manner and with stable performance. This was not demonstrated before, and it is a novel contribution. We do not require disentangling regularisation to achieve this, rendering our method (in terms of just the model definition) particularly simple and elegant. In addition, our model is directly using the ELBO for training (besides applying the “VaDE tricks”) without adding a regularisation term, which better maintains its interpretability as a lower bound to the likelihood, and which we think is desirable. Lastly, we would like to note that one can in principle combine disentangling methods and “clustered representations” [15], but we opted not to do so in the present work.
> * What are the consequences of the property of a *clustered* latent space? Besides the obvious advantage of making our method applicable to multi-facet clustering ([12-14] couldn’t be directly used for this purpose), which is the task we aim at solving, this yields further advantages: For example, one has a greater level of control when sampling from the model, as one can sample from a conditional $p(\mathbf{z_j} | c_j)$ in one facet and a marginal $p(\mathbf{z}_{j'})$ in all other facets, and therefore generate images which all possess the characteristic of the chosen cluster $c_j$ (compare Fig. 5).
>
> ---
>
> “2. The paper does not provide the quantitative methods to measure the disentanglement and sample diversity. The lack of quantitative measure makes the reader not ensure the effectiveness of MFCVAE”
>
> Yes, thank you for pointing this out! Your insight here has motivated us to perform and present further experimental work quantifying the disentanglement attained by our model (we postpone additional quantifiable results of sample diversity to your related comment below). This is a valuable and interesting addition to the paper, particularly as disentanglement metrics are uncommon in both the single- and multi-facet clustering literature.
>
> To quantitatively assess disentanglement of our method, we use a similar setup to [14] (Section 3.2 and presented in Table 1; the paper you shared) which is as follows: For all three datasets (MNIST, 3DShapes, SVHN), we use exactly those trained models (with the same hyperparameters) that we report results throughout all experimental sections. For every train and test image, we estimate the distributions $q_\phi(\mathbf{z}_j | \mathbf{x})$ for both facets j given our trained model and sample once per facet and data point. This yields new training and test datasets of two *latent* vectors for each original image, which - as we claim - represent style/colour and digit/shape, respectively.
>
> We demonstrate this in a supervised experiment as follows: We add a supervision label (e.g. digit class in MNIST) to the latent vectors of the training set. Then, explained for the example of MNIST, we train a (multiclass) logistic regression (as [14]) and a multi-layer perceptron with 1 hidden layer of 100 hidden units and softmax (default settings in sklearn) on the latent vectors and labels of the training set of a chosen subset (either all “style/colour” latent vectors only, or all “digit/shape” latent vectors, or all latent vectors (i.e. “style/colour” and “digit/shape”) concatenated). We train 3 different classifiers per dataset (and label type), one for each subset of latent vectors. As 3DShapes has two labels, we train 6 different classifiers. Additionally, each classifier is repeated 3 times with different random seeds to receive error bars (sample standard deviations). Our reported metric is prediction accuracy (in %) on the test set with error bars (one sample standard deviation).
>
> Table 2: Quantitative assessment of the disentanglement of facets in MFCVAE on all three datasets. Values report supervised classification accuracy in %, error bars are the sample standard deviation across 3 runs.
>
> | Method                            	| Dataset                               	|      MNIST     	| 3DShapes config 1 	|                 	| 3DShapes config 2 	|                 	|      SVHN      	|
> |-----------------------------------	|---------------------------------------	|:--------------:	|:-----------------:	|:---------------:	|:-----------------:	|:---------------:	|:--------------:	|
> |                                   	|                                       	|   digit class  	|    object shape   	|   floor colour  	|    object shape   	|   wall colour   	|   digit class  	|
> | Random guess 	|                                       	|             10 	|                25 	|              10 	|                25 	|              10 	|             10 	|
> |   Multiclass logistic regression  	| latent vector = z_1          	| 92.62 +/- 0.03 	| 92.85 +/- 0.16    	| 14.31 +/- 0.70	| 95.59 +/- 0.53    	| 37.98 +/- 0.43  	| 25.27 +/- 0.10 	|
> |                                   	| latent vector = z_2         	| 14.54 +/- 0.09 	| 23.92 +/- 0.24    	|  100.00 +/- 0.00 		| 21.77 +/- 0.42    	| 100.00 +/- 0.00 	| 18.99 +/- 0.86 	|
> |                                   	| latent vector = (z_1, z_2)^T   	| 92.88 +/- 0.10 	| 93.09 +/- 0.39    	| 99.97 +/- 0.06  	| 95.73 +/- 0.54    	| 99.38 +/- 0.00  	| 25.11 +/- 0.63 	|
> |       Multilayer perceptron       	| latent vector = z_1       	| 94.95 +/- 0.04 	| 95.00 +/- 0.45    	| 20.00 +/- 0.68 	| 98.26 +/- 0.16    	| 73.40 +/- 1.48 	| 69.46 +/- 0.36 	|
> |                                   	| latent vector = z_2        	| 17.34 +/- 0.24 	| 32.43 +/- 1.38    	| 100.00 +/- 0.00 	| 24.41 +/- 1.34    	| 100.00 +/- 0.00 	| 22.30 +/- 0.16 	|
> |                                   	| latent vector = (z_1, z_2)^T    | 95.27 +/- 0.07 	| 95.18 +/- 0.42    	| 100.00 +/- 0.00 	| 98.19 +/- 0.30    	| 99.97 +/- 0.06  	| 70.39 +/- 0.29 	|
>
> Just like in [14], we find that our supervised models can learn to predict the respective class label that corresponds to the “representing” facet with high accuracy if trained on latent representations of that facet only, and matches the performance when concatenating both latent representations. However, performance of the supervised classifier significantly drops when trained on the “non-label” facet, as is expected. To give a concrete example on MNIST, when our supervised classifier is trained on latent vectors of a facet that (as we qualitatively found) correspond to digit class, the classifier can learn from these latent representations to predict the digit class label with high accuracy (92.62 +/- 0.03 with multiclass logistic regression). However, when trained on the stroke width latent vectors, it can intuitively not predict the digit class label with high accuracy if both facets are properly disentangled, and this is exactly what we observe on MNIST (14.54 +/- 0.09), and across all other datasets. This experiment demonstrates the good between-facet “disentanglement” of our method *without* using an explicit disentangling method regularising our loss, which further demonstrates the competitiveness of our model.
>
> We thank you for this very valuable comment which improves our paper a lot. We will include the above results in the camera-ready version of our paper.
>
> ---

---

> > ### Author Response · Authors · 2021-08-10
> > **Response to Reviewer NacR (Part 2 of 2)**
> >
> > (continued)
> >
> > > “2. It is hard to clarify the quantitative experimental results. There is no quantitative results table.”
> >
> > Thank you for providing this vital feedback, which has really highlighted for us the benefits of including a table of quantitative results. We have now replaced paragraph 3 in Section 4.3 with a table comparing MFCVAE with other single- and multi-facet clustering models w.r.t. unsupervised clustering accuracy on MNIST and SVHN, also presenting further experiments with two architectures for VaDE that we discuss in the Appendix.
> >
> > <See Table 1 in Official Comment above for the table>
> >
> > When we compare (deep) generative models alone, on MNIST, MFCVAE has comparable performance with VaDE, and is outperforming LTVAE by 6 percentage points. On SVHN, MFCVAE outperforms both VLAC [11] by ~ 18 and VaDE by ~ 26 percentage points (we have not attempted running LTVAE on SVHN as we could not get it to learn a multi-facet configuration on MNIST, see also second bullet point). If we compare MFCVAE to non-generative models, which neither have the capability to synthesise images (Section 4.4) nor have a compositional latent space similar to presented in Section 4.2, MFCVAE is outperformed on MNIST by IMSAT [5] and ACOL-GAR [9] by ~ 6 percentage points, and outperformed on SVHN by ~ 1 percentage point and ~ 20 percentage points, respectively. Note that both of these models are single-facet, but MFCVAE is performing multi-facet clustering in the reported configuration (and this qualitatively works as we demonstrate). Unfortunately, we can only compare against the supervised label on MNIST, and this is why we are restricted to this comparison. Seen another way, single-facet clustering models like IMSAT and ACOL-GAR assume a single partition on MNIST, even though we both agree that there is more than one partition. While we have an unquantifiable, but good qualitative performance on clustering style in a meaningful way, single-facet models have no performance here.
> >
> > Lastly, we would like to refer to our other comments in this response regarding additional quantitative tables we included, both regarding sample diversity and disentanglement of our model.
> >
> > ---
> >
> > > “There exist only J=2 experiments, and additional experimental results for J>2 will be more interesting.”
> >
> > Yes, we thoroughly agree. In particular it is in this J>2 case where the MFCVAE linear vs others’ exponential scaling of Section 2a will come into its own. While outside the scope of the current paper, we hope to progress to stable model fitting in the J>2 case in future research, and that our current paper helps motivate this. Please see more discussion on J>2 in our Official comment above.
> >
> > ---
> >
> > > “1. The experimental results of MFCVAE are not significant. On MNIST, MFCVAE shows 92.02%, but IMSAT achieves 98.4%.”
> >
> > We thank you for your correct observation on classification performance and pointing this out, noting that IMSAT is on par with MFCVAE on SVHN ($57.3 \pm 3.9$ vs. $56.25 \pm 0.88$), however, we kindly cannot agree with your overall judgement of the significance of our work. We would like to discuss our experimental results with you and would be keen to find a consensus on their interpretation. We will give this a start momentarily, but before we begin, we would like to acknowledge that we should have presented our qualitative and quantitative results also in comparison to other methods clearer, and will discuss improvements to this performance comparison we have made (we already discussed the new classification table you suggested) as we go along:
> >
> > Please indulge our enthusiasm - we are keen to present to you a bit more on our perspective on the importance of embracing the fundamental difference between single- and multi-facet models. Our connection with your sharp comment above is that it is important, for progress to be made in the deep clustering field, that experiments can be designed quantitatively to  compare like with like, and in particular multifaceted with multifaceted model, to avoid the field becoming stuck in a single-faceted rut (which will be the case if we insist upon benchmarking multi-facet models directly against single-facet models on single-facet “ground truths”). IMSAT aims to find a *single* partition of the data, being fundamentally a single-facet clustering model. IMSAT cannot yield multiple clusterings at the same time. When we compare accuracy w.r.t. the digit class label in MNIST for example, MFCVAE in fact clusters by two aspects: digit class (which we measure accuracy for) and stroke width (which we have no label for). Simply assuming the digit class to be the de-facto “ground truth clustering” is really oversimplifying the analysis of this high-dimensional data set. A dataset like MNIST can be clustered in multiple meaningful ways, at least two, and probably (as you pointed out in another comment) many more. IMSAT (and many other single-facet clustering models which were applied to MNIST [2]-[9]) make this oversimplifying assumption that there is one and only one ground-truth clustering. With our work, we clearly demonstrate the rich structure that can and should be explored in high dimensional data beyond the boundaries of a single facet. We have set up a methodological base camp with a stable implementation for J=2, and hope to inspire others to work on multi-facet clustering models for high-dimensional data going forwards.
> >
> > We would also like to present to you in more detail our qualitative results in  Fig. 12 in the Appendix, comparing MFCVAE with LTVAE [1], the model closest to ours in its attempt to learn a clustered latent space of multiple facets. The figure visualises input examples assigned to clusters (=rows), comparing the empirical results of the two models on MNIST side-by-side. LTVAE in sub-figure (b) struggles to separate data characteristics into separate facets (note that in LTVAE, both facets learn clusters homogeneous w.r.t. digit class, i.e. this property is not properly disentangled between facets). Now compare this with the performance of MFCVAE, our model, in sub-figure, where both facets learn distinct characteristics: one facet learns digit class, the other one stroke width. This Figure unequivocally shows that we are strongly outperforming LTVAE, arguably our closest competitor. Our experimental results go beyond MNIST, and we show similar results for 3DShapes and SVHN (compare Fig. 3). In contrast, LTVAE presents results on STL-10, however, we are not convinced that their trained model has truly disentangled meaningful aspects in both facets, while the clusters appear homogeneous (compare LTVAE [1], Fig. 6). - We want to acknowledge that while we briefly discussed it, we should have been clearer about this comparison in our main paper. To address this, we pull Fig. 12 (in slightly condensed form) from the Appendix into the main paper as part of our 1 additional page for the camera-ready version.
> >
> > We hope we could explain our understanding and intuitions of the task of clustering a high-dimensional dataset, and also hope this argumentation makes sense to you. If you have any further questions or concerns, we are happy to discuss this further.
> >
> >
> > ---
> >
> >
> > > “2. There is no quantitative diversity metric, and it is hard to ensure that the MFCVAE generates better diverse samples.”
> >
> > Yes, we agree we haven’t used a quantitative diversity metric, and thank you for pointing out this interesting addition to our paper!
> >
> > While diversity metrics play an important role in quantifying the performance of generative models, they are not often used in the Deep Clustering literature, and while this does not mean they cannot add a useful perspective on the experimental performance of the model, it is not something a reader of a Deep Clustering paper would necessarily expect.
> >
> > We would also like to point out that we did not claim MFCVAE would generate “more diverse samples” compared to other generative Deep Clustering models such as VaDE [3], VLAC [11] or LTVAE [1]. The goal of the experimental results presented in Section 4.4 was to demonstrate how images with (one or multiple) specific abstract characteristics (e.g. ‘red background’) can be generated thanks to the clustered latent space, achieved by sampling from the conditional distribution for these facets and the marginal distribution in all other facets (see Fig. 5). We show synthetic reconstructions to demonstrate they qualitatively look reasonable (similar to how VaDE, VLAC and LTVAE presents them) and possess the desired aspects when sampling with this procedure.
> >
> > However, we agree it would be informative to compare the sampling diversity of our model against VaDE. We have chosen Learnt Perceptual Image Patch Similarity (LPIPS) [16], a common evaluation metric to measure diversity of image generations (higher is better). We compare our sampling diversity to VaDE, the most relevant method to us, and to the diversity of the real images themselves (i.e. training data and test data). 60,000 sample images are generated for each method and each dataset for an accurate estimation of LPIPS. We present the table in markdown format:
> >
> > Table 3: Quantitative assessment of the diversity of samples generated from MFCVAE trained on MNIST and SVHN, using the LPIPS metric, with comparison to samples generated from VaDE and real images from the dataset.
> >
> > |             |        MNIST        |         SVHN        |
> > |-------------|:-------------------:|:-------------------:|
> > | Real Images | 0.112 | 0.227 |
> > |     VaDE    | 0.111 | 0.187 |
> > |    MFCVAE (ours)   | 0.116 | 0.182 |
> >
> > As shown in the table, our method generates samples with similar diversity to VaDE on both datasets. Compared with the ground truth diversity of real images (training and test data combined), our method is close in MNIST and slightly lower in SVHN. To conclude, our method is competitive in generating diverse samples compared to established methods such as VaDE.

---

> > > ### Comment · Reviewer_NacR · 2021-08-18
> > > **Response to Authors**
> > >
> > > Thank you for your valuable response.
> > > Most of my concerns are resolved, and I increase my score.

---

> > > > ### Author Response · Authors · 2021-08-19
> > > > **Thank you for your work and increasing your score!**
> > > >
> > > > We thank you very much for so generously spending your time in reviewing our work and your valuable insights and suggestions which improved it. We are sincerely grateful to you for giving us the opportunity and advice to address your perceptive concerns - thanks so much for checking our response and revisions and increasing your score!

---

### Official Review · Reviewer_vHkY · 2021-07-16

**Rating:** 6
**Confidence:** 4

**Summary:**

The paper presents Multi-Facet Clustering Variational Autoencoder which is a probabilistic model base on multiple mixtures of Gaussians that find multiple characteristics of the data simultaneously. Moreover, the paper shows theoretical results for the optimisation of ELBO analytically with respect to the categorical variational posterior distribution.

**Limitations And Societal Impact:**

yes

**Main Review:**

Originality:
The authors of this paper deal with a known problem of clustering a multi-facet that is less common than single-facet models. In the proposed method, they use well-known techniques such as Monte Carlo, progressive training, a mixture of Gaussians, or estimating the ELBO but it is a novel combination of them. The novel theoretical results for optimising its ELBO is extending an influential related paper for the single-facet case.

Quality and clarity:
In general, the paper is a complete piece of work but section Experiments needs some refinement. The authors show the results of the proposed method MFCVAE and severals samples generated from MFCVAE trained on MNIST, 3DShapes, and SVHN. In my opinion, quantitative results are lacking compared to other methods.
The paper was presented classification performance in terms of unsupervised clustering accuracy on the test set, comparing commonly used single-facet and multi-facet deep clustering approaches. These results show the average effectiveness of the proposed method, the authors emphasize that high classification accuracy was not their goal. A table would be a better form of presenting these results. It is not clear why actors do not use the same comparative methods for different datasets.

Significance:
The paper presents a new approach to a less-known problem of clustering a multi-facet. An important result of this work is novel theoretical results for optimising its ELBO analytically concerning the categorical variational posterior distribution. This result may be used by other researchers. Unfortunately, the evaluation of the proposed method requires additional work (I have given some of the comments above). The paper presents only the results for J = 2, while for most of the data sets J is much greater than 2. The positive fact is that the authors know about it, they indicate that this problem will be solved in the future.


**Time Spent Reviewing:**

5

---

> ### Author Response · Authors · 2021-08-10
> **Response to Reviewer vHkY**
>
> Thank you very much for dedicating your valuable time to reviewing our research. Your insightful comments are greatly appreciated, and you have inspired a number of enhancements to the discussion and experimental results that we outline below.
>
> ---
> > “The authors of this paper deal with a known problem of clustering a multi-facet that is less common than single-facet models.”
>
> Yes, we totally agree,  in fact we are only aware of two previous (also generative) works that have tackled this problem in high dimensions: VLAC [11] and LTVAE [1]. We strongly outperform them and we now clarify this like-for-like superiority in the text.
>
> It is important to us to stress that we find exactly this focus on single-facet clustering of the majority of current Deep Clustering papers highly limiting, and this in fact motivated our work in the first place: These papers benchmark their models on high-dimensional datasets like MNIST and SVHN and compare performance only in terms of accuracy against the supervised label (digit class), which is assumed as the de-facto “ground truth” partition. However, it is clear and we would both agree that there exist more than one reasonable partition of a high-dimensional dataset like MNIST or SVHN, depending on which abstract characteristic you are interested in to cluster by. To say it bluntly, the assumption that “the partition imposed by digit class is the only correct one” is wrong, it is limiting the view to *one* possible solution to the clustering problem, but many exist. This is exactly the problem that MFCVAE qualitatively solves (we demonstrate this below in more detail).
>
> ---
>
> > “In general, the paper is a complete piece of work but section Experiments needs some refinement.”
>
> Yes, we are completely with you, and hope you will enjoy the paper much more having fully revamped the Experiments section, including adding benchmarking tables, disentanglement experiments and extra clarification of MCFVAE’s novel contribution, for a potential camera ready version:
> * New classification benchmarking table comparing unsupervised clustering accuracy based on additional benchmark experiments we have now run. This hopefully adds much clarity by replacing paragraph three in Section 4.3 (please see further discussion below).
> * New disentanglement  table of additional experimental results, quantitatively demonstrating high quality disentanglement of facets in MFCVAE (for further context, please see our response to reviewer NacR).
> * We include (a condensed version of) Fig. 12 from the Appendix, which demonstrates that we qualitatively very strongly outperform LTVAE [1], the model closest to ours in its attempt to learn a clustered latent space of multiple facets (please see further discussion below).
> * We have included further details on the swapping procedure from Appendix E.4 in Section 4.2, as well as on the definition of unsupervised clustering accuracy from Appendix E.5 in Section 4.3.
> Related to the Experiments, in Section 3.2, we also included the graphical model of Ladder-MFCVAE (Fig. 7) from the Appendix in our main paper.
>
> ---
>
> > “In my opinion, quantitative results are lacking compared to other methods. The paper was presented classification performance in terms of unsupervised clustering accuracy on the test set, comparing commonly used single-facet and multi-facet deep clustering approaches. These results show the average effectiveness of the proposed method, the authors emphasize that high classification accuracy was not their goal. A table would be a better form of presenting these results. It is not clear why actors do not use the same comparative methods for different datasets.”
>
> Thank you for your perceptive remarks, which have stimulated us to make several improvements to the manuscript, clarifying, extending, and discussing experimental results.  In following your suggestions, we hope that we have now clearly demonstrated the quantitative performance of our method.
>
> “A table would be a better form of presenting these results”: We fully agree, and have now added this table for the datasets MNIST and SVHN in Section 4.3, replacing paragraph three. The table also shows additional experiments we ran for VaDE with two architectures that we briefly discuss in the Appendix. Here, we show the table in markdown:
>
> <See Table 1 in Official Comment above for the table>
>
> The table includes several single-facet models and also a few multi-facet models. Further, some of the models are generative, and some are not. In particular, MFCVAE is generative and multi-facet, and IMSAT [5] and ACOL-GAR [9], which we compare with momentarily, are single-facet and *not* generative. While the full table is interesting, it is most illuminating to benchmark on comparable methods, such as to compare multi-facet with multi-facet methods and/or generative with generative. A related point is that datasets like MNIST and SVHN possess multiple facets/partitions but come with just a single supervised label. Almost all previous work framed the task as single-facet clustering and developed single-facet models. We follow this benchmarking framework, even though it places multi-facet models at a disadvantage, as they are being judged on single-facet performance, having been designed with multiple facet problems in mind.
>
> Turning to the table: Compared against other generative models, MFCVAE is on par with VaDE, outperforms LTVAE [1] by ~ 6 percentage points on MNIST, and outperforms VLAC [11] by ~ 18 percentage points on SVHN. Note that we attempted LTVAE on SVHN, but could not have LTVAE learn a multi-facet structure on this dataset (see also our comment below). However, when we compare MFCVAE with non-generative models (IMSAT [5] and ACOL-GAR [9]), MFCVAE is outperformed by them by roughly ~ 6 percentage points on MNIST. Further, MFCVAE is clearly outperformed on SVHN (IMSAT: by ~ 1 percentage point and ACOL-GAR: ~ 20 percentage points, respectively). - Can we conclude from this comparison that our method “show[s] average effectiveness”? We do not think so: While yes, there are significantly better methods for clustering SVHN w.r.t. digit class, these are *single-facet, non-generative* models: They cannot come up with multiple partitions of the data as we do when reporting this performance (which is the main task), and they can neither produce synthetic images (Section 4.4) or compose latent facets (Section 4.2). When we say, as you paraphrased, that “high classification accuracy was not [our] goal”, we mean that we have achieved on both MNIST and SVHN higher accuracies than the ones reported during our experiments. These trained models, however, had slightly weaker sampling quality, and since we report *the same model* across all our experimental sections (Sections 4.1-4.4) for a transparent, clear comparison, we decided for the overall better model and have not optimized accuracy with a separate configuration specifically.
>
> We stress this point so much because we have doubts that LTVAE, the model closest to ours in its attempt to learn a clustered latent space of multiple facets, did the same: While it is somewhat unclear, LTVAE seems to report its classification performance on MNIST when trained in a configuration where it learned only a single-facet, not when performing multi-facet clustering, however when generating samples it is in a configuration where it has multiple samples (see [1], Section 4.4 “With z dimension of small value like 10, LTVAE usually discovers only one facet”, as opposed to our Fig. 5).
>
> Considering only little work has been done in the space of multi-facet clustering, and the issues outlined for quantitative performance comparisons (in particular not having the required labels to compute quantitative metrics), our conclusion was to focus on qualitative performance.
>
> We acknowledge that we were not clear enough about the performance comparison to LTVAE in our main paper, and therefore decided to include Fig. 12 and the corresponding paragraph in Appendix E.3 discussing it in the main text (in condensed form). Fig. 12 visualises input examples assigned to clusters, comparing MFCVAE’s and LTVAE’s side-by-side. It is strikingly clear that LTVAE (sub-figure (b)) does not disentangle two meaningful abstract characteristics, because digit class is learned in both facets. Compare this Figure with the performance of MFCVAE in sub-figure (a): here, both facets learn distinct abstract characteristics, namely digit class and stroke width, which are (as we show in an additional experiment in response to reviewer NacR) almost perfectly disentangled, and also homogenous within each cluster w.r.t. specific values of the characteristic (e.g. ‘bold’).
>
> This comparison makes very clear that we are very strongly outperforming LTVAE, arguably the (multi-facet) model closest to us, and frankly in fact *qualitatively* demonstrate a solution to the multi-facet clustering problem in high-dimensions. We believe these results are very interesting to the research community as a whole, particularly considering the reproducibility of our results (which is often a problem in Deep Clustering), and we hope they motivate further research on multi-facet clustering in the future.
>
> Please feel free to ask additional questions or leave comments on our thoughts. We find it important to agree on how to evaluate our results compared to other models, and are happy to discuss this further.
>
> ---
>
>
> > “The paper presents only the results for J = 2, while for most of the data sets J is much greater than 2. The positive fact is that the authors know about it, they indicate that this problem will be solved in the future.”
>
> Thank you for this important point, which we respond to in the Official Comment above.

---

> > ### Author Response · Authors · 2021-08-24
> > **Please let us know if we can clarify or enhance anything at all!**
> >
> > Dear reviewer vHkY,
> >
> > Thank you very much again for dedicating your valuable time to reviewing our work!
> >
> > As the meta-review due date approaches, we just want to check in with you to make sure that we have addressed your insightful comments as fully as possible, and that we have provided all the information we possibly can to help you make your final decision. We’d be delighted to engage with any further comments or queries you have about our work.
> >
> > Thanks so much again for all your efforts!

---

> > > ### Comment · Reviewer_vHkY · 2021-08-25
> > > **Response Authors**
> > >
> > > Thanks for your response. Most of my concerns are resolved or commented, therefore I am increasing my rating.

---

> > > > ### Author Response · Authors · 2021-08-26
> > > > **Thank you for examining our response and for increasing your score!**
> > > >
> > > > Thank you very much again for spending your precious time reviewing our research -- your insights and suggestions significantly improved our work! We are sincerely grateful that you gave us the opportunity to respond to your perceptive concerns, and we are thrilled that we managed to resolve most of them. Finally, many thanks for increasing your score!

---

### Official Review · Reviewer_VMVb · 2021-07-17

**Rating:** 6
**Confidence:** 3

**Summary:**

The paper proposed Multi-Facet Clustering Variational Autoencoders, which is an unsupervised model and can learn multiple latent characteristics of the data. They showed how to optimize the ELBO in this setting and implemented the model via Variational Ladder Autoencoder architecture with progressive training. Experimental results demonstrate good empirical performance.


**Limitations And Societal Impact:**

Yes.

**Main Review:**

Real-world data usually can be clustered in multiple ways. However, most existing deep learning based clustering algorithms are designed to learn only one such partition. This paper aims to learn multiple levels of abstract characteristics that in combination can form different partitions of data. They assume Mixtures of Gaussians prior on the latent space. The paper extends variational deep embedding to this multi-facet setting and shows how to optimize the ELBO. The experiments are done on datasets of MNIST, 3DShapes, and SVHN. While the experiments seem thorough, the performance comparison to other methods was not clearly presented.

While the concept seems to be meaningful, it is unclear to me how practical these (independent) facets are. Not every data set clearly has distinct characteristics like handwritten numbers. During implementation, it’s hypothesized that each layer of the VLAE captures distinct aspects of the data. If so, how many layers are needed? Are more layers better? Is it possible that some layers learn combinations of characteristics? More discussions on these aspects will be appreciated.

The paper is in general well written and the flow is easy to follow. Much of the experiments are postponed to the appendix, making that section difficult to read.


**Time Spent Reviewing:**

5 hours

---

> ### Author Response · Authors · 2021-08-10
> **Response to Reviewer VMVb**
>
> Thank you for your thoughtful and insightful feedback! We are glad that you appreciate our paper, and would like to respond to your stated questions and remarks.
>
> ---
>
> > “While the experiments seem thorough, the performance comparison to other methods was not clearly presented.”
>
> We agree with you that our performance comparison to other methods should have been presented clearer in our submission, and we appreciate that you pointed this out. We are addressing this issue by including three pieces of additional content to our potential camera-ready version, particularly comparing to LTVAE [1], the model closest to ours in its attempt to learn a clustered latent space of multiple facets:
>
> 1) We include Fig. 12 and the paragraph discussing it from Appendix E.3 to the main paper (in a slightly shortened form). Fig. 12 visualises input examples assigned to clusters, comparing MFCVAE and LTVAE side-by-side. As is strikingly visible (sub-figure (b)), LTVAE struggles to separate data characteristics into separate facets (in particular, in LTVAE, both facets learn digit class, i.e. this characteristic is not properly disentangled between facets). Compare this with MFCVAE in sub-figure (a), where we show that both facets clearly learn distinct characteristics: digit class and stroke width. This comparison makes very clear that we are strongly outperforming LTVAE. Furthermore, our qualitative demonstration goes beyond MNIST, and shows similar results for 3DShapes and the much more difficult SVHN dataset (compare Fig. 3). LTVAE in comparison presents further results on STL-10, however, we are not convinced that their trained model has learned any meaningful aspect in each facet, even though images within clusters are homogenous (compare LTVAE [1], Fig. 6).
>
> 2) We include a table comparing MFCVAE with other single- and multi-facet clustering models in terms of *unsupervised clustering accuracy* on MNIST and SVHN (also in response to reviewers vHkY and NacR), replacing paragraph three in section 4.3. In particular, we ran additional experiments for VaDE with two architectures which we discuss in the Appendix. We present the table here in markdown format:
>
> <See Table 1 in Official Comment above for the table>
>
> On this table, we would like to note the following points:
> 1) When comparing MFCVAE with other (deep) generative models, on MNIST, MFCVAE is on par with VaDE, and outperforming LTVAE by 6 percentage points. On SVHN, MFCVAE is outperforming VLAC [11] by ~ 18 percentage points and VaDE by ~ 26 percentage points (we have not attempted running LTVAE on SVHN as we could not get it to learn a multi-facet configuration on MNIST, see also second bullet point). When comparing MFCVAE to non-generative models, which can neither produce synthetic images (Section 4.4) nor have a compositional latent space (Section 4.2), the performance of MFCVAE compared to IMSAT [5] and ACOL-GAR [9] is inferior on MNIST by ~ 6 percentage, and MFCVAE is clearly outperformed on SVHN (by ~ 1 percentage point and ~ 20 percentage points, respectively). Further, note that IMSAT and ACOL-GAR are both single-facet clustering models, while MFCVAE is performing multi-facet clustering (but we can only compare against the available supervised label).
>
> 2) We note that while it is somewhat unclear, LTVAE seems to report its clustering performance on MNIST when LTVAE has only a single facet, not when performing multi-facet clustering to obtain classification results. When rerunning LTVAE’s code with reported hyperparameters, we also experienced “collapsing” behavior. However, when generating samples, LTVAE is in a different, multi-facet configuration [1]. In contrast, we report MFCVAE results in a multi-facet configuration from a single trained model across all experiments, i.e. for Sections 4.1-4.4 for reasons of transparency and clear comparison (with error bars).
>
> 3) In Appendix E.6, we add a Figure comparing images generated from MFCVAE and LTVAE on MNIST. For MFCVAE (see Fig. 5), we observe synthetic images that are homogeneous w.r.t. the characteristic value of a cluster in the chosen facet, but heterogeneous and diverse w.r.t. all other facets (as we sample all other continuous latent variables from their marginal distribution).
> Compare this with images generated from LTVAE (see [1], Fig. 7). Here, only one facet is presented, which has homogeneous clusters (as for MFCVAE) and a comparable sample quality (with the exception of few clusters). It is unclear whether the other facet generates images homogeneous w.r.t. characteristic values of another data aspect (e.g. “style”) or not. Note we demonstrate diverse sample generations also for 3DShapes and SVHN (see Fig. 5), while LTVAE only shows MNIST images.
>
> ---
>
> > “While the concept seems to be meaningful, it is unclear to me how practical these (independent) facets are. Not every data set clearly has distinct characteristics like handwritten numbers.”
>
> We agree that there exist datasets where no meaningful multi-facet structure is present, and our proposed multi-facet clustering method would not be adequate for such instances. Yet, we were motivated by a ubiquitous number of datasets in biology and medicine that do have a multi-facet structure (e.g. Electronic Health Records, clinical and trial surveys, pan-cancer gene expression data). We believe that a method relevant even only to these datasets is of sufficient interest to a wide audience.
>
> Note, however, that it is remarkable that almost all work in Deep Clustering developed *single-facet* clustering models [2]-[9] and applied them to datasets like MNIST or SVHN, clustering and comparing performance only against the supervised digit class label, even though we would both agree that more than one ground truth partition exists on these datasets. Our work argues that doing so is not appropriate, and is missing the full picture: While the class label certainly imposes the partition with the largest statistical variation, in an exploratory scenario, one is interested in finding *all* useful groupings of the data, as one might use these other ones for downstream tasks.
>
> To illustrate the *practical* relevance of our method further, consider yourself in an exploratory setting in which one is given a complex, high-dimensional dataset and one is interested in finding meaningful groups in it. If one now goes ahead and applies any single-facet clustering method to this dataset, the assumption is that the dataset contains exactly *one* meaningful partition. But how can one know this a-priori? - We argue that especially in high-dimensional data, we typically have several partitions present which all split the data into meaningful and potentially useful groups (depending on your downstream task) [10]. In other words, there is not one “correct” partition in high-dimensional data (even though assumed by the majority of previous work), and also not just one partition you are interested in for your exploratory analysis.
>
> ---
>
> > “During implementation, it’s hypothesized that each layer of the VLAE captures distinct aspects of the data. If so, how many layers are needed? Are more layers better? Is it possible that some layers learn combinations of characteristics? More discussions on these aspects will be appreciated.”
>
> Thank you for raising these questions, we are happy to provide detailed answers. Your question is related to the question of other reviewers on the J>2 case, which we have responded to in the Official Comment, but it goes beyond it:
>
> The number of required layers (=facets) depends on the number of underlying distinct/independent characteristics which we assume in the data. For example, the two configurations of 3DShapes have two (simulated) latent factors with strong statistical variation, and here, two layers are a reasonable choice. In MNIST and SVHN, probably more aspects can be reasonably assumed. While using more layers and the question how to choose the number of layers is beyond the scope of this work, using the likelihood lower bound to guide this choice might be an interesting idea to try next, but we acknowledge that optimisation might become more difficult.
>
> It is possible that some layers learn combinations of characteristics. For example, MNIST has at least three abstract characteristics (digit identity, digit slant and stroke width), and when we trained a 2-layer model, two of these three characteristics are combined into one layer (see Fig. 3 and 5 where digit identity and slant are combined). We argue that single-facet models learn exactly such cross-combinations all the time, and are theoretically suffering from combinatorial explosion (see Appendix A for details).
>
> ---
>
> > “The paper is in general well written and the flow is easy to follow. Much of the experiments are postponed to the appendix, making that section difficult to read.”
>
> We agree that the experiments section postpones important details to the Appendix, without which some details cannot be fully understood from the main paper, and thank you for pointing this out. Based on your remarks, we have made the following adjustments of the Experiments section in the main text:
> 1): We included the Ladder-MFCVAE graphical model (Fig. 7).
> 2): We included a condensed paragraph from Appendix E.4 on the swapping procedure.
> 3): We give more details on the definition of unsupervised clustering accuracy from Appendix E.5.
> We think 1) is particularly important to discuss in detail, as we spent major efforts on fixing stability issues of our model which are widely known in the research area of Deep Clustering, but often not sufficiently addressed by previous work [1]-[3] (see Appendix E.1).
>
> Lastly, we would like to point out that Figures 13-16 and 20-23 merely illustrate the complete figures (all clusters=rows visualised) of their more condensed versions in Section 4, not “new” experiments.

---

### Official Review · Reviewer_bBmr · 2021-07-18

**Rating:** 7
**Confidence:** 3

**Summary:**

The authors extend the VaDE model (Jiang et al.) to allow for multiple, independent facets i.e. Mixtures of Gaussians. The authors first point out and resolve a technical oversight in derivations related to estimation of $q(c \mid x)$ of Jiang et al. (VaDE). The authors implement their formulation in an architecture based on ladder VAE to extract multiple clusterings of the data simultaneously. The authors demonstrate their method on 3 different benchmark datasets.

**Limitations And Societal Impact:**

The paper and appendices include thoughtful notes for each dataset. I also appreciate the transparency in discussing limitations and experimental details!

**Main Review:**

Note: I'm not very familiar with other comparable models learning hierarchical structure in latent variables, and cannot judge the novelty of the contribution too well. I'm willing to revise my score in case I have missed something.

Pros
 - The paper is clearly written, and the theoretical portion is well laid out and pleasant to read. The ideas are uncluttered, and the appendices provide enough context for an outsider to the field to follow.

Cons:
1. The authors do not learn more than a handful of clusters in each facet, and do not practically demonstrate the feasibility of the approach in a setting where the number of independent clusters blows up. It might be helpful to elaborate more on the shortcomings of Gumbel-softmax based approaches or even compare this approach with one where the discrete variable is learned through the re-parametrization trick.

2. Many details have been offloaded to the appendices. This is not a major issue since the appendices are well written. For example, it might might help to move schematics and related description for the ladder MFCVAE to the main text since this is the version used for all 3 datasets/ to demonstrate the key points in Sec 4.1-4.4.

Other comments:
 - It's not clear how much do results depend on various hyperparameter choices. In particular, number of components $K$ for each facet, dimensionality of $z$, priors on the distribution across components, are likely central to the interpretability/ apparent disentaglement of the models. It would be great if the authors can address this point.

 - Majority of the demonstrations rely on qualitative assessment of the reconstructions, for visual datasets where the factors have some semantic meaning (e.g. color, stroke styles, digit identity etc.), and convolutional neural network layers are known to compose features into images. It would be great if the authors could demonstrate these methods for exploratory settings, for example to make predictions on publicly available tabular, biological datasets that come with rich metadata and often a hierarchical structure.

**Time Spent Reviewing:**

8

---

> ### Author Response · Authors · 2021-08-10
> **Response to Reviewer bBmr (1 of 2)**
>
> Thank you for your thoughtful review. We are glad you like our paper and thank you for highlighting the clarity and transparency of our work. Below, we would like to address the questions and concerns stated in your review:
>
> ---
>
>
>
> > “The authors do not learn more than a handful of clusters in each facet, and do not practically demonstrate the feasibility of the approach in a setting where the number of independent clusters blows up.”
>
> Yes, we agree: it is a limitation of our work that the number of “ground truth” clusters per facet in the chosen datasets is small. It would be exactly such datasets with a large number of clusters per facet where the theoretical advantage of linear complexity in terms of number of clusters vs. exponentially exploding number of clusters in single-facet models would become particularly useful, and demonstrating this empirically is an interesting next step.
>
> We would like to note that the datasets we used are very common within the research area of Deep Clustering (see these papers [1-9] which also use one or multiple of them). We used such “standard datasets” to qualitatively demonstrate that we can learn a previously known, but unsolved and overlooked problem in high-dimensional data: finding *multiple* useful partitions of data in a fully unsupervised fashion. LTVAE [1], the model closest to ours in its attempt to learn a clustered latent space of multiple facets, struggles to separate data characteristics into separate facets (c.f. [1] Fig. 5; in particular, both facets learn digit class, i.e. this characteristic is not properly disentangled between facets). Compare this Figure with the performance of MFCVAE in Fig. 3, which is demonstrated on MNIST, 3DShapes and SVHN (not just MNIST). To see a side-by-side comparison of MFCVAE and LTVAE on MNIST, compare Fig. 12 in our Appendix E.3, which we decided to pull to the main paper (in condensed form) in the potential camera-ready version to make this comparison clearer. Our point is: This work was centred on showing that multi-facet clustering can work very well, and we achieved this goal, strongly outperforming related work. We focussed our efforts on this qualitative demonstration and the associated technical difficulties around stability which we discuss below in more detail.
>
> ---
>
> > “It might be helpful to elaborate more on the shortcomings of Gumbel-softmax based approaches or even compare this approach with one where the discrete variable is learned through the re-parametrization trick.”
>
> Yes, we agree that the benefits of using VaDE-like approaches over Gumbel-Softmax (GS) approaches deserve to be expanded on.
>
> The first problem we faced is that we wish to have relatively high-dimensional discrete latents, which we have found GS does not tend to perform well on. And of course, in this case, exact marginalisation is not appealing either. And, as you know, GS introduces bias into the optimisation.
>
> There is also the related choice of whether to have an explicitly-trained variational posterior for a given discrete variable, $q_\phi(c|\cdot)$, (implemented as its own neural network) or instead to define such a posterior as needed. Recall that in our VaDE-like approach, we construct the Bayes-optimal factorised-posterior for c as needed.
>
> While the former approach of having an explicitly-represented neural posterior for c does mean that a full factorisation of the variational posterior is possible - $q_\phi(c,\mathbf{z}|\mathbf{x})=q_\phi(c|\mathbf{x})q_\phi(\mathbf{z}|c,\mathbf{x})$ or $q_\phi(c|\mathbf{z},\mathbf{x})q_\phi(\mathbf{z}|\mathbf{x})$, say - it does have two downsides.
> Firstly, exact marginalisation is particularly costly here (very much like M2 in Kingma et al., NeurIPS 2014), so we really would have to use GS or similar to sidestep this, which as we have said we want to avoid due to bias and instability.
>
> Secondly though, even if cheap and performative reparameterization tricks for (multiple) discrete latent variables were a solved problem, we would still be keen to use a VaDE-like approach in our problem to avoid introducing additional neural networks.
>
> Recall that in the Variational-Ladder-like approach of MFCVAEs we make different layers of latents in the model be ‘about’ different aspects of the data from having increasingly-powerful neural mappings between the input data and each successive layer of latents. We are using the inductive biases that come from the neural parameterisation of the model to obtain the kinds of representations we want.
>
> If we were to introduce extra $q_\phi(c|\cdot)$ networks, they would themselves introduce further inductive biases into the overall model. Their neural architecture, too, would have to be considered, and it is not clear to us how best to go about designing them. We note that prior work [11] tried this kind of approach. Our VaDE-like approach gets better performance, and is conceptually simpler.
>
> Finally, note that if we had an autoregressive factorisation over each facet’s c variables in the posterior, we would have further complications. See Lievin et al. (AABI 2019) for discussion of how GS bias compounds in the hierarchical case, while methods like REBAR (Tucker et al., NeurIPS 2017) and backpropagation-through-the-void (Grathwohl et al., ICLR 2018) scale very poorly to the hierarchical case (see Pervez et al. (ICML 2020) for detailed discussion of this).
>
> We agree that these are important questions, and will extend our discussion of how part of the motivation of the structure of our model comes from trying to sidestep these issues, giving us a mathematically-simple and high-performance approach.
>
>
> ---
>
> > “1. Many details have been offloaded to the appendices. This is not a major issue since the appendices are well written. For example, it might might help to move schematics and related description for the ladder MFCVAE to the main text [...].”
>
> We agree with you and thank you for pointing this out. In our submission, the discussion on our neural implementation (VLAE architecture) and training algorithm (adapted progressive training) in Section 3.2 was too short, in particular considering that we spent major efforts on fixing the stability issues (e.g. w.r.t. random seed) of our Deep Clustering model which are widely known, but often overlooked in previous work [1]-[3] (compare Appendix E.1 and specifically Fig. 9 for a discussion and demonstration on their importance).
>
> To address your feedback, we have frontloaded content taken from Appendices D.2 (VLAE architecture) and D.3 (Progressive training algorithm) to the main part of our potential camera-ready paper, including the schematics (graphical model) of Ladder-MFCVAE (Fig. 7) (in slightly condensed form). We do this as part of our additional 1 page of content granted for the camera-ready version. We hope this addresses your feedback adequately and makes the main paper easier to understand.
>
> ---
>
> > “It's not clear how much do results depend on various hyperparameter choices. In particular, number of components K for each facet, dimensionality of z, priors on the distribution across components, [...]. It would be great if the authors can address this point.”
>
> Thank you for pointing this out. We have discussed hyperparameter tuning, various design choices to be made and detailed hyperparameters of our architecture in Appendix D.4, but we agree that further details would be useful, and are happy to discuss them here:
>
> Overall, the VLAE architecture and progressive training algorithm made our results very stable w.r.t. most hyperparameters, including batch size, learning rate, the number of components K in each facet, and the number of batches used during fade-in of the facets. As a result, we did not perform extensive grid search on these hyperparameters, but hand-picked some values instead, as shown in Table 1 in Appendix D.4. Both quantitative results (in terms of unsupervised clustering accuracy) and qualitative results (in terms of patterns shown in Fig. 3, 4 and 5) do not vary much with these mentioned hyperparameters, at least when remaining in reasonable ranges around our chosen values.
>
> However, we noticed that the dimensionality of $\mathbf{z_j}$ for the style/colour facet must be rather carefully set within some range to achieve strong disentanglement of facets. In particular, its value needs to be rather small, which makes intuitive sense to us as it avoids all characteristics of the data being solely learnt in the same facet. For example, for MNIST and SVHN, the $dim(\mathbf{z_j})$ needs to be around 5 for a strong disentanglement. For 3DShapes, $dim(\mathbf{z_1})$ needs to be around 2.
>
> For priors on the distribution across components, we selected the discrete uniform distribution for each $p(c_j)$ to encourage clusters with similar sizes to be learnt. We defined each $p(\mathbf{z_j} | c_j)$ to be a Gaussian, either with a diagonal or a full covariance matrix. On SVHN, a significant improvement in the unsupervised clustering accuracy is observed when a full covariance matrix is used for each $p(\mathbf{z_j} | c_j)$, while on the other two datasets, we can’t observe a significant difference. We did not try other priors, mainly focussing on a qualitative demonstration rather than improving quantitative performance to the maximum, but agree that trying more complex priors would be an interesting idea for future work.
>
> ---

---

> > ### Author Response · Authors · 2021-08-10
> > **Response to Reviewer bBmr (2 of 2)**
> >
> > (continued)
> >
> > > “[...] It would be great if the authors could demonstrate these methods for exploratory settings, for example to make predictions on publicly available tabular, biological datasets that come with rich metadata and often a hierarchical structure.”
> >
> > We agree that this is an interesting next step: Deep Clustering methods have mostly been applied to image datasets, but other datasets such as tabular data in biology and medicine (such as clinical or trial surveys) are likewise very high-dimensional and as you say often have a very interesting, hierarchical structure. In fact, this work was initially motivated by the question of how to cluster an Electronic Health Records (EHR) dataset in the first place.
> >
> > While we agree that demonstrating our method on such a much more complex dataset “in the wild” would be interesting, our goal in this project was to qualitatively demonstrate (as discussed above in more detail) that multi-facet clustering of high-dimensional data is possible with good performance. We even started off attempting this on a specific, non-public EHR dataset which we had access to that has a hierarchical multi-facet structure (ICD-10 codes), but soon realised that such a dataset would require lots of preprocessing and is much too complex for this qualitative and methodological work, and thus reverted to simpler “standard” datasets, which also make our work comparable to other single- and multi-facet Deep Clustering work. Furthermore, publicly available multi-facet clustering benchmark datasets with multiple ground-truth clustering labels  that can be readily used for this task are scarce, and this itself is a task of future work. If we would have used the EHR dataset available to us, this would have compromised reproducibility which was very important to us in this methodological project (compare also our provided code and shell scripts to easily reproduce all our experiments attached in the Supplementary Material of this submission). - We hope these are understandable reasons why we decided to limit our work in this way.

---

> > > ### Comment · Reviewer_bBmr · 2021-08-26
> > > **Response to authors**
> > >
> > > I appreciate the detailed response by authors to all reviewers' comments. This paper develops a clear idea, and provide proof-of-concept empirical experiments. The clarity of presentation is also a plus, and that is something that multiple reviewers noted. The additional comparisons, benchmarks, and explanations make the paper stronger. I continue to stand by my initial score, and am happy to advocate for this paper if continues to remain borderline.

---

> > > > ### Author Response · Authors · 2021-08-30
> > > > **Thank you for assessing our response and for advocating for our paper!**
> > > >
> > > > Thank you so much again for your brilliant feedback! Your insights and suggestions have been extremely helpful. Thank you for your kind words of support for our methodological idea, proof-of-concept experiments and clarity of presentation. We are delighted that you feel our additional experiments and explanations have strengthened the paper. We are most grateful and excited that you are happy to advocate for our paper and thank you for your high score!

---

### Author Response · Authors · 2021-08-10
**Response to all reviewers**

### Introduction

We are extremely grateful to you, our reviewers, for dedicating your valuable time to considering our work, and for providing your astute and constructive feedback. Your comments and suggestions provide insights and encouragement that have motivated us to enhance the manuscript, perform further experiments, and clarify how we contribute to the existing methodological landscape of this diverse and important field. While we respond to each of you individually, we would like to stress the following two points which we will discuss in detail in our responses below:


* Our work is a qualitative step forward, in that we are thinking outside the mainstream methodological toolbox, in which high dimensional data sets are modelled with a *single* “ground truth” partition/facet. The overwhelming majority of Deep Clustering papers make this single-facet modelling assumption, and also base experimental evaluation on it. In contrast, our work is a successful demonstration of a *multi-facet* (generative and probabilistic) model in high dimensions, strongly outperforming previous attempts of the multi-facet clustering task. We have now added extra clarification of this contribution in the paper, and sincerely thank you for your comments and suggestions that motivated this enhancement.
* We strongly outperform LTVAE [1], the model closest to ours in its attempt to learn a clustered latent space of multiple facets. We provide clear evidence on this, and discuss this in detail below.

---

### Overview of changes to the manuscript

We provide a brief overview on all changes made compared to our submission towards a potential camera-ready version which allows 1 additional page of content:

* Fig. 12 (cluster examples assigned to clusters) from the Appendix is moved (in condensed form) to Section 4.1, qualitatively demonstrating that *MFCVAE strongly outperforms LTVAE* in terms of disentanglement of facets. This is further compared with an additional Figure in Appendix E.6 visualising images generated from MFCVAE and LTVAE on MNIST.
* A table and additional experimental results that quantitatively underline the *strong disentanglement of facets in MFCVAE* learned in all datasets.
* A table comparing *classification performance* in terms of unsupervised clustering accuracy of single-facet and multi-facet, generative and non-generative models.
* A table quantitatively comparing the *sample diversity* in terms of LPIPS [16] between MFCVAE and VaDE.
* Additional discussion on *Gumbel-Softmax* in comparison to the VaDE trick, and on *disentanglement methods* in the related work section.
* Additional *experimental details* in Section 3.2, in particular the Ladder-MFCVAE graphical model (Fig. 7) in condensed form, in Section 4.2 (from Appendix E.4) on the swapping procedure, and in Section 4.3 on the definition of unsupervised clustering accuracy.


---

### Other contributions

In addition, we would like to point out two contributions of our work that were not discussed in the review so far:

* We provide *stable* performance (in terms of random seed) across a wide range of hyperparmeters thanks to the VLAE architecture and the progressive training algorithm. The problem of stability (visible either in high error bars or even collapsing training runs) is often not sufficiently addressed in Deep Clustering research [1-3], yet, it is a commonly known problem here, and in particular for VAE based approaches. See paragraph 2 in Section 4.3 and Appendix E.1 for a detailed discussion.
* Partially due to the first point, *reproducibility* is a tremendous problem in Deep Clustering research. We provide our code, pretrained models, and shell scripts to reproduce our experimental results in the supplementary, and believe the Deep Clustering community will appreciate this material.

---

### Similar question by multiple reviewers

Furthermore, we would like to respond to one question/remark that was similarly raised by two reviewers:

> Reviewer vHkY: “The paper presents only the results for J = 2, while for most of the data sets J is much greater than 2. The positive fact is that the authors know about it, they indicate that this problem will be solved in the future.”

> Reviewer NacR: “There exist only J=2 experiments, and additional experimental results for J>2 will be more interesting.”

We thoroughly agree with you. The J>2 case is the one where the linear scaling of MFCVAE discussed in Section 2a and Appendix A would become strikingly useful compared to the exponential scaling of single-facet clustering models. While outside the scope of the current paper, we hope to present models with J>2 in future research.

---

### Additional quantitative results tables

We present three new tables (in markdown) on classification performance, disentanglement and diversity of generated samples based on new experimental results which we discuss in detail below and which will be part of our revised manuscript. The classification table is referred to in multiple reviews, so we show it here:

Table 1: Classification performance of single-facet (SF) and multi-facet (MF), generative (G) and non-generative (NG) models in terms of unsupervised clustering accuracy. Values are in % and taken as reported.

| Method            	|       MNIST      	|       SVHN       	|
|-------------------	|:----------------:	|:----------------:	|
| DEC (SF; NG)          	|       84.3       	|  11.9 $\pm$ 0.4  	|
| VaDE (MLP; SF; G)    	| 89.09 $\pm$ 3.32 	| 27.03 $\pm$ 1.53 	|
| VaDE (conv.; SF; G)  	|  92.65 $\pm$1.14 	| 30.80 $\pm$ 1.99 	|
| IMSAT (SF; NG)        	|  98.4 $\pm$ 0.4  	|  57.3 $\pm$ 3.9  	|
| ACOL-GAR (SF; NG)     	| 98.32 $\pm$ 0.08 	| 76.80 $\pm$ 1.30 	|
| VLAC (MF; G)         	|         -        	|  37.8 $\pm$ 2.2  	|
| LTVAE (MF; G)        	|       86.3       	|         -        	|
| MFCVAE (ours; MF; G) 	| 92.02 $\pm$ 3.18 	| 56.25 $\pm$ 0.93 	|


---

### References

Lastly, we provide a list of references which we use in our responses below:

[1] Li, X., Chen, Z., Poon, L.K. and Zhang, N.L., 2018. Learning latent superstructures in variational autoencoders for deep multidimensional clustering. In International Conference on Learning Representations.

[2] Xie, J., Girshick, R. and Farhadi, A., 2016. Unsupervised deep embedding for clustering analysis. In International conference on machine learning (pp. 478-487). PMLR.

[3] Jiang, Z., Zheng, Y., Tan, H., Tang, B. and Zhou, H., 2017. Variational deep embedding: An unsupervised and generative approach to clustering. In Proceedings of the 26th International Joint Conference on Artificial Intelligence (pp. 1965-1972).

[4] Yang, B., Fu, X., Sidiropoulos, N.D. and Hong, M., 2017. Towards k-means-friendly spaces: Simultaneous deep learning and clustering. In international conference on machine learning (pp. 3861-3870). PMLR.

[5] Hu, W., Miyato, T., Tokui, S., Matsumoto, E. and Sugiyama, M., 2017. Learning discrete representations via information maximizing self-augmented training. In International conference on machine learning (pp. 1558-1567). PMLR.

[6] Shaham, U., Stanton, K., Li, H., Nadler, B., Basri, R. and Kluger, Y., 2018. SpectralNet: Spectral clustering using deep neural networks. In International Conference on Learning Representations, 2018.

[7] Yang, X., Deng, C., Zheng, F., Yan, J. and Liu, W., 2019. Deep spectral clustering using dual autoencoder network. In Proceedings of the IEEE/CVF Conference on Computer Vision and Pattern Recognition (pp. 4066-4075).

[8] Mukherjee, S., Asnani, H., Lin, E. and Kannan, S., 2019. ClusterGAN: Latent space clustering in generative adversarial networks. In Proceedings of the AAAI Conference on Artificial Intelligence (Vol. 33, No. 01, pp. 4610-4617).

[9] Kilinc, O. and Uysal, I., 2018. Learning latent representations in neural networks for clustering through pseudo supervision and graph-based activity regularization. In International Conference on Learning Representations.

[10] Von Luxburg, U., Williamson, R.C. and Guyon, I., 2012. Clustering: Science or art?. In Proceedings of ICML workshop on unsupervised and transfer learning (pp. 65-79). JMLR Workshop and Conference Proceedings.

[11] Willetts, M., Roberts, S. and Holmes, C., 2019. Disentangling to cluster: Gaussian mixture variational ladder autoencoders. In 4th Workshop on Bayesian Deep Learning (NeurIPS 2019).

[12] Kim, Hyunjik, and Andriy Mnih. "Disentangling by factorising." International Conference on Machine Learning. PMLR, 2018.

[13] Mathieu, Emile, et al. "Disentangling disentanglement in variational autoencoders." International Conference on Machine Learning. PMLR, 2019.

[14] Sha, Lei, and Thomas Lukasiewicz. "Multi-type Disentanglement without Adversarial Training." Proceedings of the 35th AAAI Conference on Artificial Intelligence. 2021.

[15] Dupont, E., 2018. Learning disentangled joint continuous and discrete representations. In Proceedings of the 32nd International Conference on Neural Information Processing Systems (pp. 708-718).

[16] Zhang, R., Isola, P., Efros, A.A., Shechtman, E. and Wang, O., 2018. The unreasonable effectiveness of deep features as a perceptual metric. In Proceedings of the IEEE conference on computer vision and pattern recognition (pp. 586-595).

---

### Decision · Program_Chairs · 2021-09-27

**Decision:**

Accept (Poster)

**Comment:**

The paper is well-written and the proposed method shows good performance (quantitative results were shown in the rebuttal).  Correcting a flaw in previous work is a plus.  Reviewer's major concerns, including lack of quantitative results, dependence on hyperparameter choice, etc., have been addressed in the rebuttal.